# Designer artificial environments for membrane protein synthesis

Conary Meyer [1,6], Alessandra Arizzi [1,6], Tanner Henson[1,2,3], Sharon Aviran [1,4], Marjorie L. Longo [5], Aijun Wang [1,2,3] & Cheemeng Tan [1]✉

Protein synthesis in natural cells involves intricate interactions between chemical environments, protein-protein interactions, and protein machinery. Replicating such interactions in artificial and cell-free environments can control the precision of protein synthesis, elucidate complex cellular mechanisms, create synthetic cells, and discover new therapeutics. Yet, creating artificial synthesis environments, particularly for membrane proteins, is challenging due to the poorly defined chemical-protein-lipid interactions. Here, we introduce MEMPLEX (Membrane Protein Learning and Expression), which utilizes machine learning and a fluorescent reporter to rapidly design artificial synthesis environments of membrane proteins. MEMPLEX generates over 20,000 different artificial chemical-protein environments spanning 28 membrane proteins. It captures the interdependent impact of lipid types, chemical environments, chaperone proteins, and protein structures on membrane protein synthesis. As a result, MEMPLEX creates new artificial environments that successfully synthesize membrane proteins of broad interest but previously intractable. In addition, we identify a quantitative metric, based on the hydrophobicity of the membrane-contacting amino acids, that predicts membrane protein synthesis in artificial environments. Our work allows others to rapidly study and resolve the "dark" proteome using predictive generation of artificial chemical-protein environments. Furthermore, the results represent a new frontier in artificial intelligence-guided approaches to creating synthetic environments for protein synthesis.

Membrane proteins play crucial roles in cell biology and biotechnology, ranging from cell signaling to therapeutic discovery[1]. Yet, the study of membrane proteins is hindered by our inability to synthesize them in pure lipid environments. One main challenge is the seemingly complex interplay of chemical environment, lipid, and protein[2]. General rules dictating successful membrane protein synthesis are elusive and sometimes contradictory[3], resulting in tremendous and often failed attempts to synthesize them. Along this line, living cells have been extensively used for synthesizing membrane proteins due to their ability to provide a native cellular environment for protein expression and folding[4]. However, live cell-based methods must overcome the potential toxicity of overexpressed membrane proteins, difficulties in obtaining high yields of membrane proteins, and challenges in purifying membrane proteins from complex cellular extracts[5,6]. To overcome these limitations, cell-free systems have emerged as an alternate tool for synthesizing membrane proteins in

[1]Department of Biomedical Engineering, University of California, Davis, Davis, CA 95616, USA. [2]Center for Surgical Bioengineering, Department of Surgery, University of California Davis School of Medicine, Davis, USA. [3]Institute for Pediatric Regenerative Medicine (IPRM), Shriners Children's Northern, California, USA. [4]Genome Center, University of California, Davis, Davis, CA 95616, USA. [5]Department of Chemical Engineering, University of California, Davis, Davis, CA 95616, USA. [6]These authors contributed equally: Conary Meyer, Alessandra Arizzi. ✉e-mail: cmtan@ucdavis.edu

artificial environments. Cell-free systems offer advantages such as rapid protein synthesis, flexibility in protein engineering, and the ability to control the artificial synthesis environment[4,6–9]. Additionally, cell-free systems enable the synthesis of membrane proteins in pure lipid environments, specifically liposomes[5], allowing for the study of their structure, function, and interactions in a controlled environment that mimics their native lipid bilayer environment[10]. Liposomes provide a suitable environment for the reconstitution of membrane proteins, thereby enabling the study of their functions, such as interactions with other proteins or ligands[11], membrane fusion processes, and the morphological changes of membranes after fusion[12]. Furthermore, liposomes can be used to deliver therapeutic membrane-associated proteins to target cells, similar to extracellular vesicles[2].

Given the complexities of membrane protein synthesis and the limited understanding of protein-lipid-chemical interactions, a method for designing artificial synthesis conditions is essential. Until now, most efforts have favored brute force optimization, which exhaustively searches constrained search spaces. In Bruni et al., they attempted to synthesize 61 different eukaryotic proteins[13]. Though they found successful synthesis conditions for 57% of the proteins, only 15% of the proteins were synthesized using liposomes. Isaksson et al.[14] attempted to synthesize 38 different proteins, but only 7 were synthesized using lipids. An effective approach for predictive membrane protein synthesis is the combination of computational and high-throughput synthesis methods. However, with only hundreds of data points, no computational framework, and a lack of standardization in the literature, there are insufficient datasets to synthesize an accurate predictive framework for designing artificial synthesis environments. In addition, no platform allows flexible, reproducible, and high-throughput manipulation of chemical conditions in artificial membrane-protein synthesis. Finally, a quantitative and rapid reporter for measuring the performance of artificial membrane-protein synthesis is also lacking. Altogether, these factors hinder the rapid design of artificial synthesis environments for membrane proteins, resulting in significant gaps in the membrane proteome[15]. Without the ability to synthesize membrane proteins rapidly, the rate of functional studies has not kept pace with the accumulation of sequence data, resulting in a growing gap between known and unknown protein functions[16].

Here, we develop a custom droplet printing robot capable of generating hundreds of artificial and cell-free synthesis environments in a combinatorial array[17,18] (Fig. 1A). Each synthesis environment uses an *E. coli*-based whole cell extract produced in-house based on past work[19]. The membrane proteins are synthesized outside of supplied liposomes to allow for co-translational outside-in insertion into the membrane. Successful design of optimal artificial synthesis environments is determined by comparing the raw fluorescence of an engineered fluorescent solubilization reporter in the presence and absence of liposomes. The solubilization reporter informs the successful liposome insertion, but not functional outcomes, of the cell-free synthesized membrane proteins. We modulate various parameters, including lipid type, small molecule concentrations, and chaperone proteins, generating a large search space of artificial synthesis environments. A subset of these artificial environments is screened to generate initial training data. An ensemble of deep neural networks is then trained on the experimental data and used to predict artificial environments in the subsequent round of synthesis. We then develop a feature set to describe the structural characteristics of membrane proteins. Finally, we use these features to train a classifier to predict the solubilization success of membrane proteins in artificial environments.

## Results
### Established a high-throughput platform for designing artificial synthesis environments
Our platform constituted cell-free protein synthesis, nanoliter droplet printing, and a fluorescent solubilization reporter. The platform is termed MEMPLEX (Membrane Protein Learning and Expression) from here on. Many cell-free protein synthesis systems (CFPS) have been created in the literature by us[19] and others[20,21], but we opted to make CFPS using the common strain BL21(DE3) Star[22] (Methods 2 and 3), to allow ease of extending MEMPLEX and broadening its utility. Using CFPS for high-throughput study often encounters a non-trivial challenge: CFPS can have variable yield and composition between batches[23]. Before using each CFPS for membrane protein synthesis, we benchmarked it against the previous CFPS. Each CFPS batch was screened against a range of magnesium and potassium concentrations to identify any substantial deviation in responses (Supplementary Fig. 1, Method 2). In general, extracts showed about 5% variance between each extract and the previous preparation across all salt concentrations tested. The preparation and benchmarking protocol of our CFPS were used throughout the experiments.

The next major component of MEMPLEX is our custom nanoliter droplet printer. In past work[18], we have demonstrated robust droplet generation of the numerous reagents required to assemble cell-free reactions. This system utilized empirical mass measurements from printing to calibrate the required pressures to ensure accurate and robust printing of the target reagent. Here, we made several enhancements to enable robust high-throughput printing of the CFPS reagents (see details in Method 4). Specifically, we upgraded the robotic arm used to position the printer cartridge to improve the speed and range of motion of the system. We also implemented an anti-evaporation system that limits volume loss from wells to <0.1 μL over the course of printing by selectively uncovering only the target well while printing. Automated, on-deck calibrations using a micro-balance were also implemented so that every reagent could be calibrated in under 30 s immediately before printing. For this work, the droplet printer was used to assemble all unique cell-free reactions in 384-well plates. The system printed 60 nL droplets of various reaction components (i.e. plasmids, magnesium glutamate, poly-ethylene glycol 8000, etc.) at a frequency of 20 Hz with a droplet volume variance of <3% based on mass measurements. Each reaction required the addition of 16 droplets to reach the desired 2 μL reaction volume, which equates to >6000 droplets per 384-well plate (Method 5).

To enable MEMPLEX, another crucial component is a reporter that can detect if a membrane protein is successfully solubilized in the presence of liposomes. This detection is traditionally achieved with Western blots[4,13,24], but this detection method has a low throughput and is not amenable for MEMPLEX. Hence, we implemented a split GFP solubilization reporter based on the work by Waldo et al.[25]. The reporter utilized two components: a large GFP fragment comprising the first 10 of the 11 beta strands of GFP, and a small fragment containing the last beta strand of GFP. Both components were independently non-fluorescent, but when mixed in the same reaction, they bound to one another and reformed the fluorophore (Supplementary Fig. 2). Past work has utilized the same reporter to assess the solubility of numerous proteins in a 96-well plate format and demonstrated the principle that the small fragment is inaccessible for complementation when the attached protein aggregates[25–27]. Here, we hypothesized the same principle could be applied to membrane proteins. The small fragment was attached to the C-terminus of each membrane protein using a 12 amino acid flexible linker, while the large fragment was bulk-purified (Method 6) and supplemented after each cell-free synthesis reaction. Theoretically, when membrane proteins were not properly integrated into a lipidic environment, they would aggregate and produce minimal background fluorescence. When the membrane proteins were solubilized into the membrane, the tag would be accessible to bind the large fragment (Supplementary Fig. 2, Supplementary Notes 1 and 2, Method 9). The fluorescence was converted to the quantity of protein in the reaction using a calibration curve (Supplementary Fig. 3, Supplementary Note 3, Methods 7 and 8). The protein must be

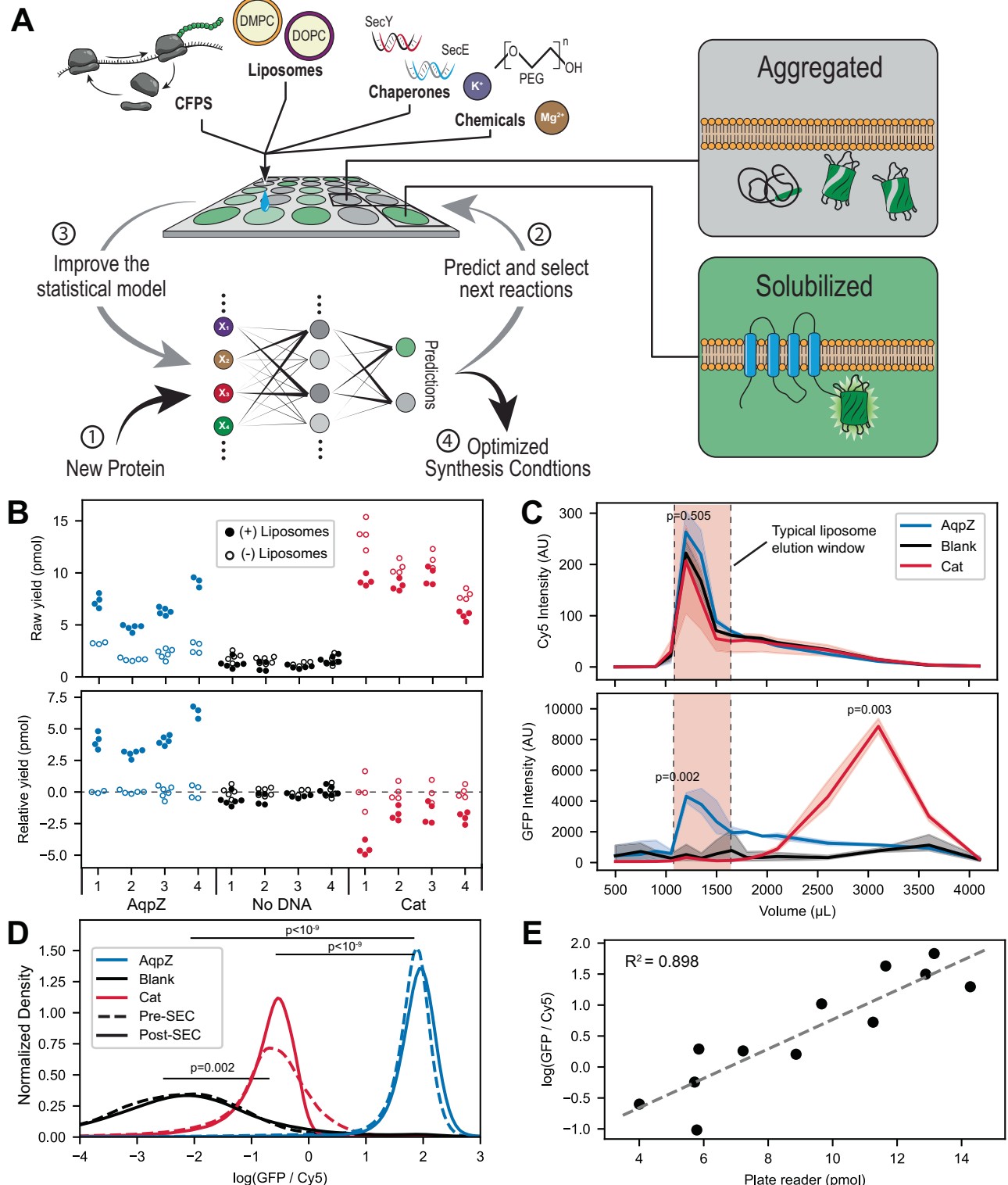

oriented with the tag facing the exterior of the membrane for the large fragment to bind (Supplementary Note 4).

Indeed, our results demonstrated that the GFP-solubilization reporter could reliably report the increased yield of solubilized membrane proteins in the presence of small unilamellar liposomes of ~100 nm diameter. The reported yield also correlated with the function of Bacteriorhodopsin. (see Method 10 for liposome preparation and Methods 11 and 12, Supplementary Figs. 4, 5, and 6 for characterization). First, the amount of the membrane protein AquaporinZ (AqpZ) incorporated into the membrane increases with the addition of

liposomes by 4.2 pmol on average across the evaluated reactions ($p$-value $< 10^{-15}$) (Fig. 1B and Supplementary Table 1). When the same reaction conditions were evaluated for the soluble protein Chloramphenicol acetyltransferase (Cat), the addition of liposomes showed a decrease of 2.48 pmol ($p$-value $< 10^{-7}$) (Fig. 1B). Second, the elutions from size-exclusion chromatography (SEC, Method 13) show that AqpZ-liposomes exhibit a 12.6-fold increase in GFP signal at the peak liposome elution volume, 1.2 mL, compared to samples producing Cat ($p$-value $= 0.002$) (Fig. 1C). Third, flow cytometry analysis (Method 14, Supplementary Fig. 6) of the SEC purified samples shows that AqpZ-

**Fig. 1 | Established MEMPLEX (Membrane Protein Learning and Expression) to design artificial environments for membrane protein synthesis. A** Schematic illustrating MEMPLEX. A custom droplet printer tests new proteins against varied conditions; solubilization is assessed by a split-GFP reporter. Each condition has a control lacking liposomes to track the background from misfolded protein. The resulting data is used to train deep neural networks to select the highest-yielding reactions for the next round of experiments. **B** Only the membrane protein (AqpZ) shows an increase in fluorescence with liposome addition. Raw yields (top) and yields were corrected by subtracting the no-liposome control (bottom) (Method 8). Filled circles: +liposomes; open circles: −liposomes. ($n = 4$–$6$). See Supplementary Table 1 for reaction conditions. **C** Size-Exclusion Chromatography (SEC) traces confirm only AqpZ co-elutes with liposomes (Cy5-labeled) (Method 13). Top panel: Liposome elution monitoring Cy5 signal from labeled lipids. Bottom panel: GFP elution monitoring GFP reporter bound to liposomes/membrane proteins. Red shaded area indicates the expected elution for liposomes. Blue line is AqpZ, Black line is no protein, and Red line is Cat. ($n = 3$ biological replicates, each with 3 technical replicates pooled for analysis. Data are shown as mean values +/- SEM. Overall differences were tested via one-way ANOVA. If ANOVA test was significant, pairwise comparisons were performed using Welch's two-sided $t$-tests with Bonferroni correction.). **D** Comparison of the solubilized membrane protein signal (GFP) normalized by the liposome size (Cy5) from flow cytometry analysis of single liposomes. AqpZ-liposomes show higher ratios than samples producing Cat or no protein (Method 14). Kernel density plots show the distribution of the log(GFP/Cy5) values taken from events recorded during flow cytometry (Supplementary Fig. 6) (flow cytometry samples $n = 3$ biological replicates, 100,000 events recorded for each). Dashed lines represent samples prior to SEC, solid lines represent samples after SEC. ($n = 3$, $p$-values from a Tukey's HSD post hoc test performed on median values per biological replicate, following a significant two-way ANOVA.). **E** Plate reader measurements show linear agreement with the normalized GFP signal observed in the flow cytometry sample (Method 14). ($n = 12$ biological replicates). Source data are provided as a Source Data file.

liposomes leads to a 60-fold increase in the median GFP signal when normalized by liposome size compared to samples not expressing any proteins, while the expression of Cat only increased the ratio by 4.6-fold (Fig. 1D). Furthermore, the measurement from the plate reader is highly correlated with the normalized GFP signal from flow cytometry ($R^2 = 0.898$, Fig. 1E). Altogether, the results corroborate that the GFP-solubilization reporter is a robust reporter for general membrane association in the presence of liposomes. We note that the reporter does not distinguish functional vs non-functional membrane proteins.

## Initial screening demonstrated variable artificial environments for membrane protein synthesis

The successful synthesis of each membrane protein depends on its chemical environment. We focused on 5 reaction parameters that are known to impact membrane protein synthesis: magnesium glutamate concentration[28], potassium glutamate concentration[28], the concentration of the molecular crowding agent Polyethylene glycol 8000[29], the lipid type used in the liposomes (DMPC, DPPC, or DOPC)[30], and the addition of plasmid DNA encoding the protein translocating channel SecYE[31] (Fig. 2A). A detailed description of the motivation behind the selection of these parameters is provided in Supplementary Note 5. The concentration ranges were defined from empirical data or literature (Supplementary Note 6, Supplementary Figs. 7 and 8). The successful expression of SecYE in the cell-free system was validated by tagging SecY with the small GFP tag and evaluating its expression across different conditions. It was found that equal amounts of SecY and SecE plasmids yield the highest results, and that successful synthesis was observed for all salt conditions evaluated (Supplementary Fig. 8C). The lipid type was converted into numerical values using the length of the carbon tails of each lipid, DMPC with 14 carbons as low, DOPC with 18 carbons as high, and DPPC with 16 carbons as in between. Every reaction composition was tested in quadruplicate with an additional quadruplicate of the equivalent reaction composition excluding liposomes (Supplementary Note 1). To reduce the number of reactions required for each protein while still covering the search space, we implemented a generalized subset design (Method 15) to decrease the number of reactions needed by 3-fold while still testing the most informative reactions[32].

We chose 28 membrane proteins (Supplementary Table 2, Method 1) with the following criteria: (1) taken from *E. coli*, *A. thaliana*, or *H. sapiens*. (2) ranged in size from 15–68 kDa and from 2–12 transmembrane domains. (3) all α-helical. (4) had transmembrane domain annotations[33]. (5) the C-terminus of the protein faced the interior of the cell. To further narrow the list, we selected a subset that was tested by Bruni et al. to compare their reaction success[13]. The remaining proteins were selected for their therapeutic applications (tetraspanins for drug delivery[34–36]), biotechnological utility (olfactory protein for real-time volatile chemical sensing[37–39]), or medical relevance (FDA approved drug targets with no experimentally verified structures[40,41]). All proteins were subject to the generalized subset design for the initial screening, generating a set of 8296 unique cell-free reactions across all proteins. A successful reaction composition was defined as having a two-sided $t$-test $p$-value <0.05 and a mean difference of >1 pmol of protein, when comparing the experimental replicates with the corresponding negative control reactions that excluded liposomes (Fig. 2B). A yield of 1 pmol of protein inside the 2 μL reaction corresponds to 20 μg/mL of protein, assuming a molecular weight of 40 kDa, which is the average size of the tested proteins. This yield also corresponds to an average of 8 proteins per liposome, assuming an average liposome diameter of 100 nm. The yield threshold was set where >80% of conditions that cleared the threshold were statistically significant, to minimize the likelihood of false positives.

Based on the data, consistent with the generic rule of protein synthesis, small membrane proteins were easier to produce than large membrane proteins (Fig. 2C). The three proteins that were <25 kDa showed a high average likelihood of successful membrane incorporation (75%). Proteins above 25 kDa had a much lower likelihood (2.9%), indicating the need for non-trivial optimization to identify their ideal synthesis condition (Fig. 2B). Next, our results challenge the generic claims of using "standard" CFPS (i.e., without tuning chemical compositions) for synthesizing membrane proteins. The "standard" CFPS composition was set as the tested concentration most similar to the average of the common reagent concentrations seen across other cell-free papers. We found that the optimal synthesis conditions of each protein are not consistent, indicating that there is no discernible optimal condition for all membrane proteins. The 3 top performing reaction compositions for each protein were selected, and the concentrations of each component in those reactions were averaged (Fig. 2D). This heat map illustrates the difference in the preferred reaction compositions across the tested membrane proteins. The preferred reaction compositions are defined as the average concentrations for each component in the top 3 performing reactions for each protein, not the exact concentration of the top-performing reaction. Proteins showed variability in preference with PEG concentration and lipid type, with a standard deviation of 0.35. For 39.3% of the proteins, all top 3 reactions used 2% PEG, while 25% of the proteins used 0% PEG in all of the top 3 reactions. A similar separation between proteins is seen for lipid type, with 39.3% exclusively using DOPC lipids in their top 3 reactions, while 14.3% only used DMPC lipids. The three other variables (i.e., K, Mg, and SecYE) showed less definitive preference toward one concentration or another. Across all proteins, there appears to be a bias for lower SecYE concentration (average normalized concentration of 0.33) and for DOPC lipids (average normalized value of 0.66). Mg, K, and PEG average closer to the middle concentration (0.44, 0.51, and 0.58). This data supports the common belief that each protein's synthesis conditions must be individually

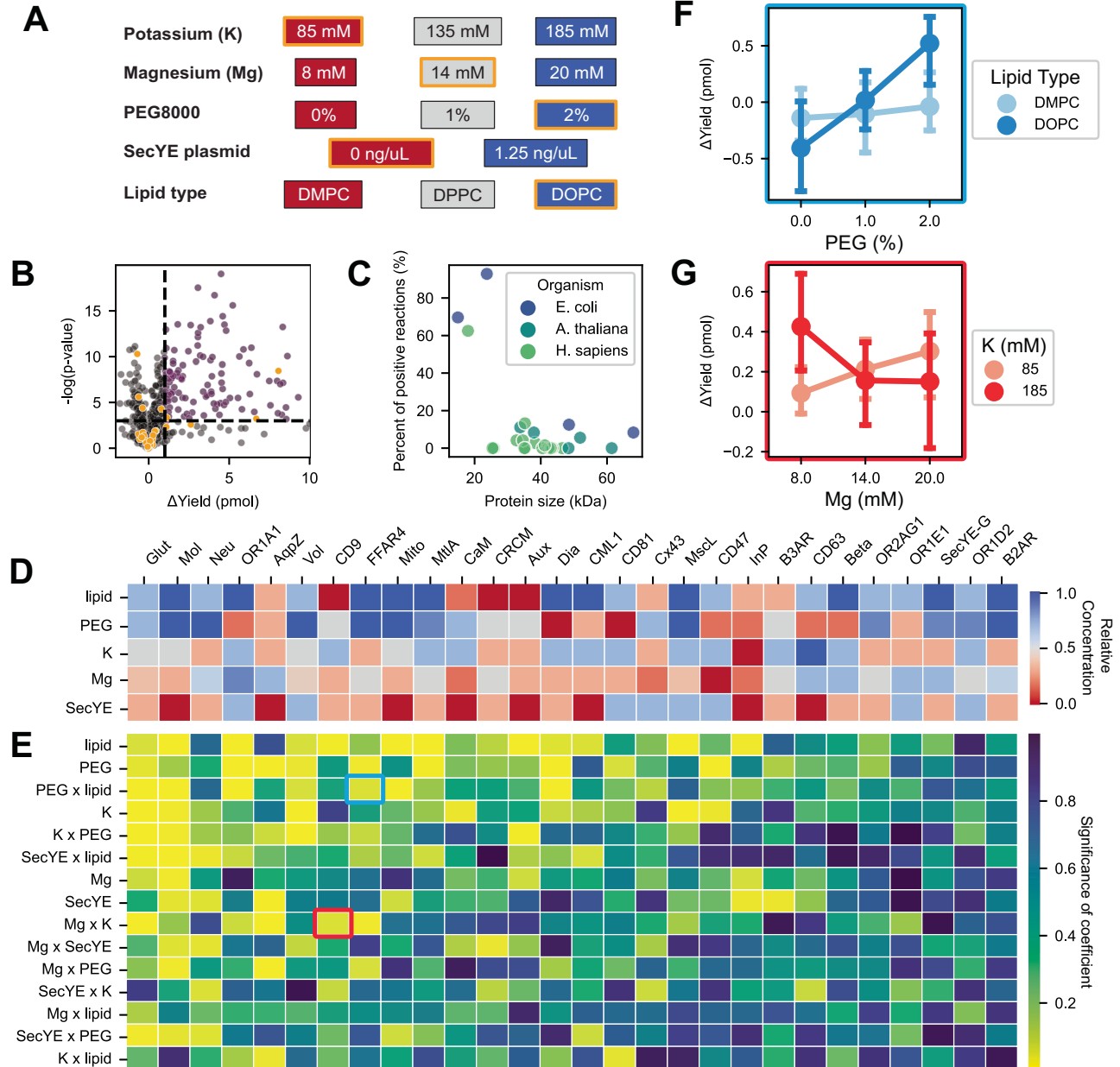

**Fig. 2 | Chemical, lipid, and protein factors interact to determine optimal artificial synthesis environments of membrane proteins. A** Schematic of the initial screen's search space. Lipid types were numerically encoded by carbon tail length, from DMPC (14 carbons, low) to DOPC (18 carbons, high). Orange-outlined boxes indicate the "standard" reaction composition. **B** Each dot compares the same reaction ± liposomes ($n = 914$). The vertical line ($\Delta$yield = 1 pmol) and horizontal line ($p = 0.05$ from two-sided $t$-tests) define successful synthesis. Orange points: standard condition; magenta points: yield $\geq 1$ pmol and $p < 0.05$. ($n = 914$, 4 with liposome and 4 without liposomes each). **C** The low frequency of identifying a successful reaction composition for proteins >25 kDa supports the need for high throughput screening. A successful reaction composition was defined as having a $p$-value <0.05 and a mean difference of >1 pmol of protein. The number of reactions

tested that were labeled as successful was divided by the total number tested for each protein ($n \geq 24$ per protein, 4 with liposome and 4 without liposomes each). 8,296 unique cell-free reactions. **D** The optimal reaction compositions differ widely across proteins. The average concentration for each component across the top 3 performing reaction conditions was calculated for each protein. The color scale corresponds to the colors used in (**A**). ($n = 8296$) **E** Heat map of $p$-values (ordinary least squares models) shows frequent interdependence among variables ($n = 8296$). Blue and red boxes correspond to panels (**F**) and (**G**). **F** FFAR4 requires specific lipid types (DMPC vs. DOPC) at varying PEG levels ($n = 24$). Error bars: 95% CI. **G** CD9 depends on potassium levels as magnesium changes ($n = 24$). Error bars: 95% CI. Source data are provided as a Source Data file.

optimized while cautioning against using universal CFPS conditions to produce membrane proteins.

In addition, we found significant interdependence between the reaction variables for each membrane protein. To evaluate the interdependence between these variables, we trained an ordinary least squares regressor using each variable concentration and interaction terms using the pairwise product of those variables on each protein.

Then, we evaluated the derived $p$-value of those interaction terms to assess whether they impacted the model accuracy (Fig. 2E, Method 16). Every term, except for "Mg x lipid" (i.e., interdependence between Mg and Lipid), was found to be significant (two-sided $t$-test $p$-value < 0.05) for at least one of the tested proteins. Nearly 50% of the terms were found to be significant for at least 5 proteins. Specific examples of this interdependence are shown in Fig. 2F, G. This interdependence of

reaction variables and protein-specific responses motivates the use of a more sophisticated model capable of capturing the convoluted, protein-specific trends that are observed in the screening data.

## Active learning optimized artificial environments for membrane-protein synthesis

While previous work generally posits that membrane-protein synthesis requires non-trivial optimization, our work is the first to systematically reveal the quantitative and nonlinear dependency of each membrane protein on its synthesis conditions. Here, we leverage MEMPLEX and the dataset to create a data-driven approach (i.e., Active Learning) to accelerate the synthesis of membrane proteins without resorting to brute-force optimization.

We implemented a deep neural network (NN) to model the relationship between the reaction variables and the solubilized-protein yield. To account for the differential response of each protein to the reaction variables, the identity of each protein was one-hot encoded, where a unique binary column for each protein was created to encode this categorical variable into numerical values. The solubilized-protein yield of each membrane protein was scaled using a unique z-score scaler to account for the difference in absolute yield across the proteins. Next, we implemented an ensemble-based active learning approach (Method 17), where models comprised 3 different complexity levels (4, 5, or 6 hidden fully connected layers), and where each architecture was trained using 3 different batch sizes (10, 50, and 200) and 5 different train-test data splits. This resulted in 45 different trained models (Fig. 3A). The full dataset gathered from the reaction screens (4148 liposome reactions) was split up by 80% for training and 20% for testing. After each model was trained, they were evaluated based on their performance on the test set (Fig. 3B). Each reaction prediction from each model (Fig. 3B, blue dots) exhibits a large scatter with an $R^2$ value of 0.47. After averaging the ensemble predictions (Fig. 3B, black dots), the scatter was reduced, resulting in an $R^2$ value of 0.64, indicating that ensemble-based predictions outperform single-model ones. In addition, an ensemble was trained with the encoded protein identifiers excluded from the input. The model performed far worse, with an $R^2$ value of 0.41, further emphasizing that there is no shared optimal condition across the proteins (Supplementary Fig. 9).

Next, we used the NN ensemble to optimize membrane protein synthesis. The ensemble predicted the resulting yield of new candidate reaction conditions (Fig. 3A). The mean-predicted-values were used to select reactions that were predicted to perform well. The standard deviation of those predictions was used to assess the confidence in the mean predicted values. These two metrics were used to calculate the upper confidence bound metric (Method 17)[42,43]. The 12 reactions with the highest score were selected for testing in the subsequent experiments. Combining the previous screening data and the additional active learning reactions, we generated a dataset of >11,000 unique reactions (including each replicate as a unique observation). From this full set, 21 membrane proteins (Fig. 3C) showed statistically significant improvement in synthesis yield (two-sided $t$-test $p$-value < 0.05) between the active learning selected (Fig. 3C, blue points) and screening reactions (Fig. 3C, red points). The remaining 7 proteins showed insignificant improvement from active learning (Supplementary Fig. 10). The Benjamini–Hochberg method was used to correct the $p$-values to avoid type I errors in assessing the successful synthesis of a given protein when considering all tested reactions. With the multiple hypotheses testing correction, our full platform successfully synthesized 3 proteins (OR1A1, CD47, and CD9) not synthesized in the literature (Fig. 3C, bottom first row, gray box), 6 proteins (Glut, Vol, Aux, InP, Dia, and CaM) that were failed to be produced by others (Fig. 3C, bottom first row, black box), and 7 proteins previously synthesized by others (AqpZ, Beta, Mito, Mol, MscL, MtlA, and SecYE-G). Several proteins were tested using the low-throughput size-exclusion chromatography and flow cytometry analysis to verify that the proteins

were successfully incorporated into the liposomes (Supplementary Fig. 15). In addition, the successful expression of the leukocyte surface antigen CD47 and its association with liposomes was verified by Western Blot (Method 23, Supplementary Fig. 18A). We further verified the successful production of CD9 and CD81 using a CD9/CD81-specific antibody to tag the protein and measure its distribution using flow cytometry (Supplementary Fig. 18B-C). In addition, we verified that the selected reaction compositions perform similarly across an 8-fold volume change and in tubes compared to well plates (Supplementary Fig. 19).

Furthermore, we investigated if any variable provided a higher likelihood of improving the solubilization yield of a new membrane protein. Briefly, we analyzed the trajectory of yield improvement. We used the "standard" reaction condition (Fig. 2A) and the best-performing reaction condition for each protein (Fig. 3C). We calculated the Euclidean norm between each reaction component concentration in the standard and the optimal concentration and the resulting yield in the standard and optimal concentration. The resulting distances were plotted for each component and each protein (Fig. 3D). PEG and potassium showed the highest average distances of 0.96 and 0.91, respectively, across all proteins. SecYE showed the lowest average distance of 0.67. The results show that no one variable can predictively optimize membrane-protein incorporation, corroborating the common challenges encountered in empirical methods of synthesizing new membrane proteins. Our work provides the first data and algorithm-guided method to overcome the common barrier in membrane-protein synthesis.

## Synthesis yields across various reaction conditions can be leveraged to improve predictions of optimal artificial synthesis environments

Leveraging the new dataset (Fig. 3C, >11,000 data points), we explored the possibility of transferring the knowledge from some proteins to make better predictions on the ideal synthesis conditions of a new membrane protein. We generated a t-distributed Stochastic Neighbor Embedding (t-SNE) to capture the similarities between proteins in responding to different reaction conditions (Method 18). t-SNE maps multi-dimensional data to a 2D plane, termed embedded space, where the proximity of the points in the embedded space captures their similarity in the original, higher dimensional dataset. Each protein was characterized by its predicted yields for all combinations of reaction compositions (Method 18). We then assessed the similarity between proteins using these yield profiles and visualized their relationship to one another on a 2D plane using t-SNE. Proteins that are derived from the same organism do not group well in this embedded space (Fig. 4A, top panel). There is also no clear clustering of proteins based on their length (Fig. 4A, bottom panel).

We next sought to evaluate if the protein's location in this embedded space could predict the solubilization yield of a specific reaction composition on a protein previously unseen by the model. We replaced the protein IDs with the embedding values in the input and re-trained the NN ensemble using different train-test splits of all the data, excluding one protein each. This resulted in 28 different ensembles. We then used each ensemble to predict the response for the protein it never saw during training. We calculated the $R^2$ value of the predicted versus observed z-scored values for each protein. We compared the performance of 3 types of inputs with respect to protein information: (1) No protein info, (2) protein identifiers, and (3) t-SNE embedding. The inclusion of unique identifiers slightly decreases the model's accuracy compared to having only the reaction features (Fig. 4B), likely because it cannot generalize the knowledge it learns from the identified proteins to new ones. When the embedded features are included, there is a 68% increase in the average $R^2$ compared to the models using the protein identifiers (Fig. 4B, two-sided $t$-test with Bonferroni correction $p$-value = 0.001). This improvement indicates that if the

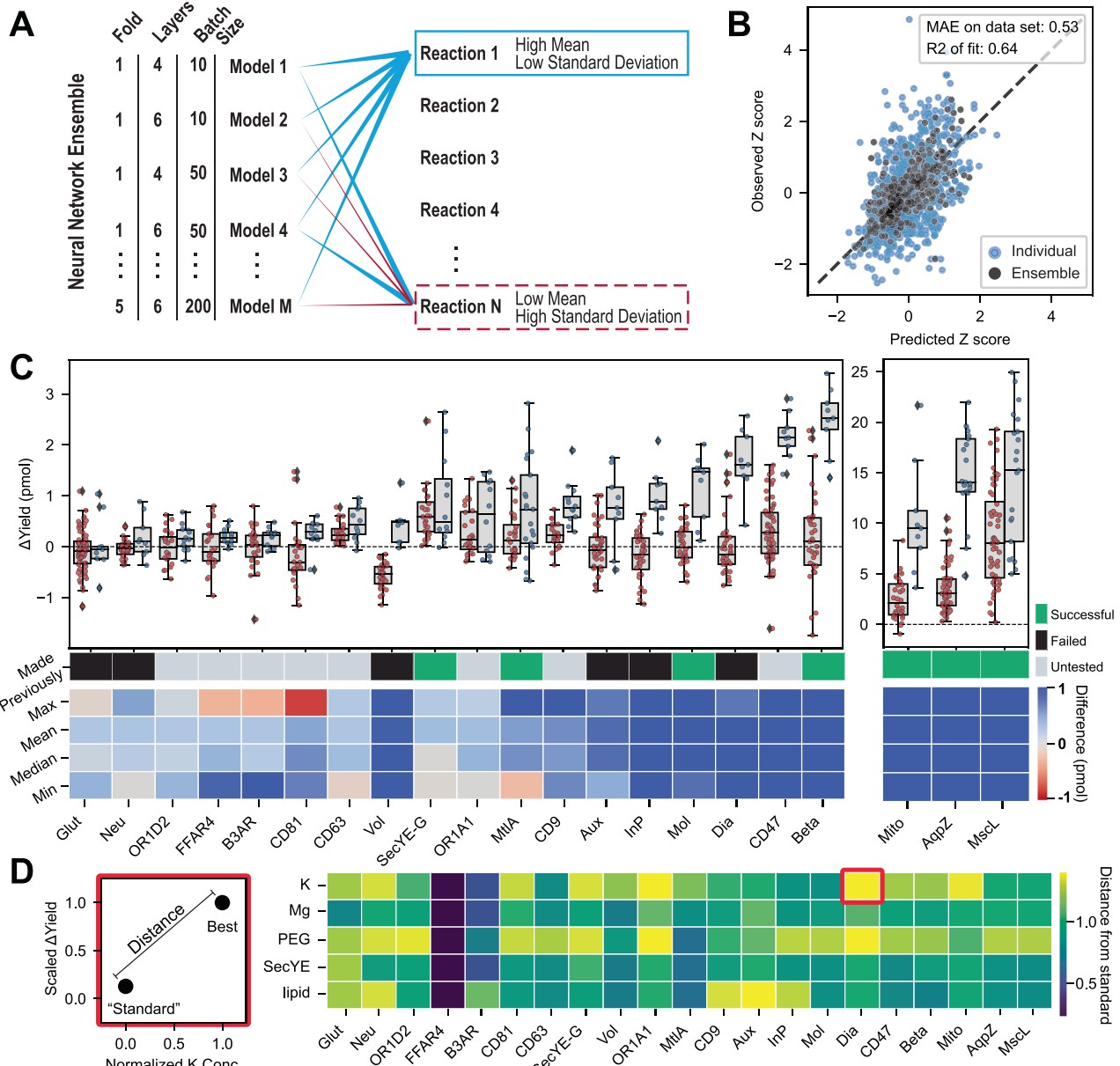

**Fig. 3 | Machine learning enables the rapid design of artificial environments to synthesize previously unattainable membrane proteins. A** Illustration of the ensemble-based active learning strategy. 45 different models were trained using: (1) different levels of model complexity (4, 5, or 6 hidden layers), (2) different batch sizes during training (10, 50, and 200) and (3) different train-test splits in the data to capture the data differently. These diverse models are then used to predict the yield of possible reaction conditions. The reactions with consistently high predicted values are selected over the reactions with low predicted values or high variance among predictions (Method 17). **B** The ensemble outperforms the individual model predictions when comparing the predicted to the observed Z-scored values (Method 17). Blue: individual model predictions, Black: average of all model predictions versus average of all observed values from a specific reaction composition. **C** Active learning results in higher synthesis yield of membrane proteins. The points represent the average of all biological replicates ($n = 4$) for a given reaction composition. All proteins in this plot show a statistically significant (two-sided $t$-test $p$-

value < 0.05) difference between the screening (red dots) and active learning (light blue dots) reactions. Box plots illustrate the interquartile range (25th to 75th percentile), with the center line indicating the median, and the whiskers extending to minimum and maximum values. 7 proteins do not show a statistically significant change from the active learning (Supplementary Fig. 10). The bar immediately below the box plots indicates whether the protein has been reportedly attempted in previous works. Gray indicates it has not been reported, Black indicates it has been tried but was unsuccessful in cell-free protein synthesis using liposomes, Green indicates it was shown to be successful. The heat map indicates the changes (max, mean, min, and median) between the screening and active learning populations. >11,000 data points. **D** PEG and potassium result in the largest average increase in membrane-protein yield when comparing the standard vs. optimal synthesis conditions. The calculation of the Euclidean norm, $\sqrt{(\triangle Yield)^2 + (\triangle Concentration)^2}$, is shown in the left panel. Source data are provided as a Source Data file.

similarities between a protein's response to reaction conditions can be captured, the reaction condition preference for a new protein could be extrapolated from known information about other proteins. It also indicates that the location of a protein in the embedded space (Fig. 4A) predicts ideal synthesis conditions of a new membrane protein.

## Inner-protein hydrophobicity metrics predicted the synthesis success in artificial environments

None of the simple protein descriptions we included in the input, including organism, the number of transmembrane passes, and protein length, could predict if proteins could or could not be solubilized

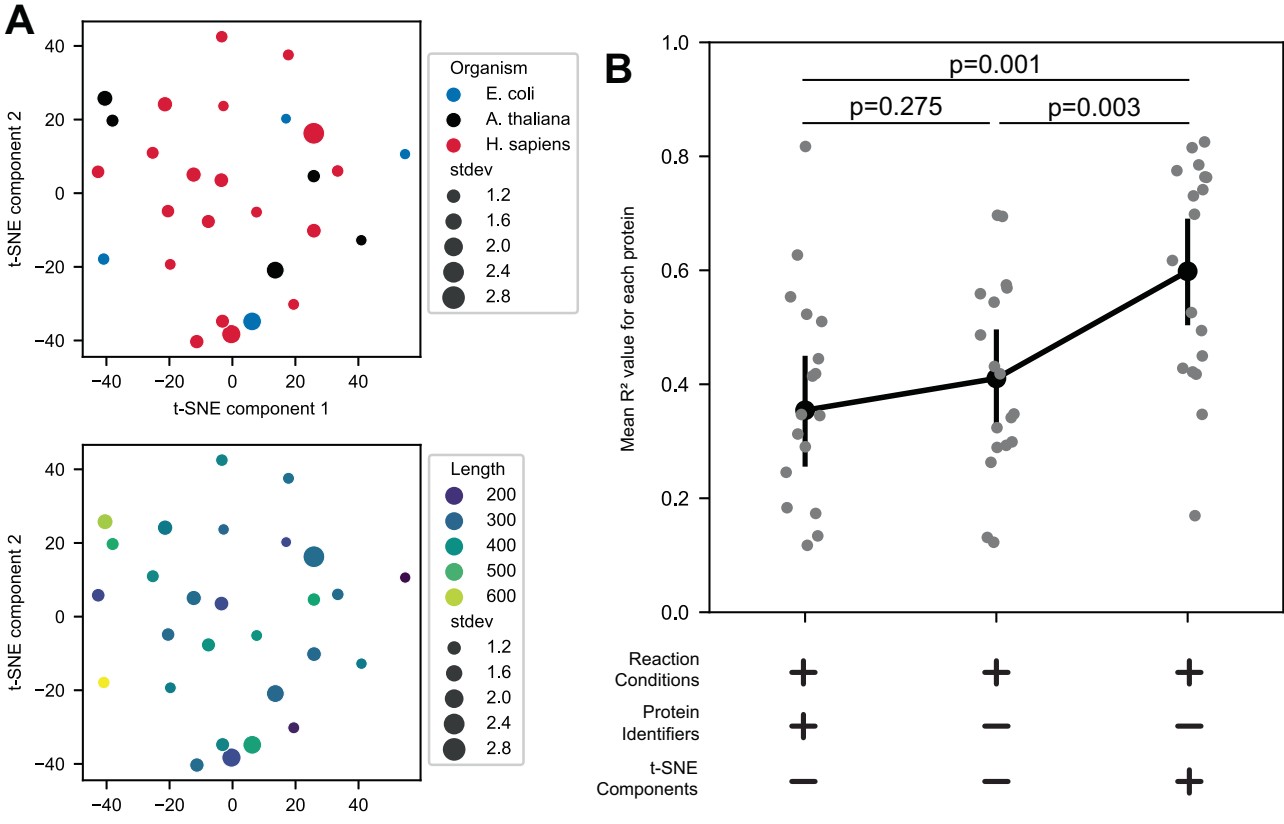

**Fig. 4 | Cross-protein learning improves the prediction of optimal artificial environments for membrane-protein synthesis. A** t-SNE embedding capturing the proteins' differential response to reaction conditions reveals no clear clustering of proteins. Each point represents the centroid of all individual model's predictions for how a protein will respond to all assessed reaction conditions (Method 18). The size of the point represents the standard deviation between the model's predictions for each protein. Top panel: colored by organism of origin. Bottom panel: colored by the length of the protein. **B** The inclusion of a protein's location in the reaction condition embedding space yields improved prediction accuracy. Individual ensembles are trained using the specified input data, one where each protein is held out of the dataset and then predicted afterward. Proteins with no successful synthesis conditions were excluded. $n = 16$ proteins. Each data point represents the $R^2$ obtained by a given model on a particular protein. Statistical significance was evaluated using repeated-measures ANOVA (with 'Protein' as the subject factor and 'Model' as the within-subject factor). Where overall significance was detected, pairwise comparisons between models were conducted using two-sided paired $t$-tests with a Bonferroni correction. Error bars represent mean ± SEM. Source data are provided as a Source Data file.

in the presence of liposomes. Hence, we computed protein structure features that would capture how the protein interacts with its environments and how energetically favorable the folded protein is. We generated a dataset comprised of structures predicted by AlphaFold[44,45]. The structures were oriented in a simulated membrane using transmembrane helix annotations obtained from UNIPROT[46] (Method 19) and subsequently used to generate numerous quantitative metrics to compare across proteins. The dataset includes all alpha-helical transmembrane proteins from *E. coli*, *A. thaliana*, and *H. sapiens* with more than one transmembrane domain and that passed all quality checks. This analysis generated a dataset of 4,612 annotated membrane proteins with 32 features each (Method 20). Briefly, every amino acid was assigned to a bin that would capture a unique physiochemical environment that the amino acid could exist in ref. [47]. These bins were defined by two criteria: the layer and the shell (Fig. 5A). The layer describes the height of an amino acid relative to the center of the simulated membrane. The shell indicates whether the amino acid was located inside or outside the membrane, and if inside, whether it was internal to the protein or in contact with the surrounding phospholipids in the membrane. Each amino acid was then given different quantitative metrics to describe how it might interact with that environment, which include 1) charge, 2) hydrophobicity, 3) depth, indicating how many amino acids are nearby, and 4) probability, indicating the probability of finding that specific amino acid in that

specific layer, shell, and depth across the full set of protein structures. All the values of all amino acids in each bin for each protein were averaged to calculate the final value used as a feature in the model. An example of a complete feature is the mean hydrophobicity of all amino acids in the inner hydrophobic layer of the membrane in direct contact with the surrounding lipids.

Next, we separated the membrane proteins into two classes. Proteins were labeled as successfully synthesized if one reaction composition cleared these two criteria: (1) the difference between the liposome and no liposome reactions was >1 pmol and (2) the Benjamini–Hochberg corrected $p$-value from a $t$-test comparing the liposome and no liposome reactions was <0.05 (Method 21). Looking across all screening and active learning reactions, 16 of the 28 proteins were labeled as successfully synthesized (Supplementary Fig. 11). To visualize the similarities between the two classes of proteins among the full set of proteins in the dataset, we used t-SNE to generate 2D projections of the protein structure feature data. We found that the separation of the two classes of proteins in the embedded space changed depending on the subset of structural features that were included before generating the embedding (Supplementary Fig. 12).

Utilizing the insight from t-SNE projections, we generated ~3000 different embedded spaces using different combinations of structural features and then trained Support Vector Machine (SVM) classifiers on each embedding to evaluate the separation of the two protein classes

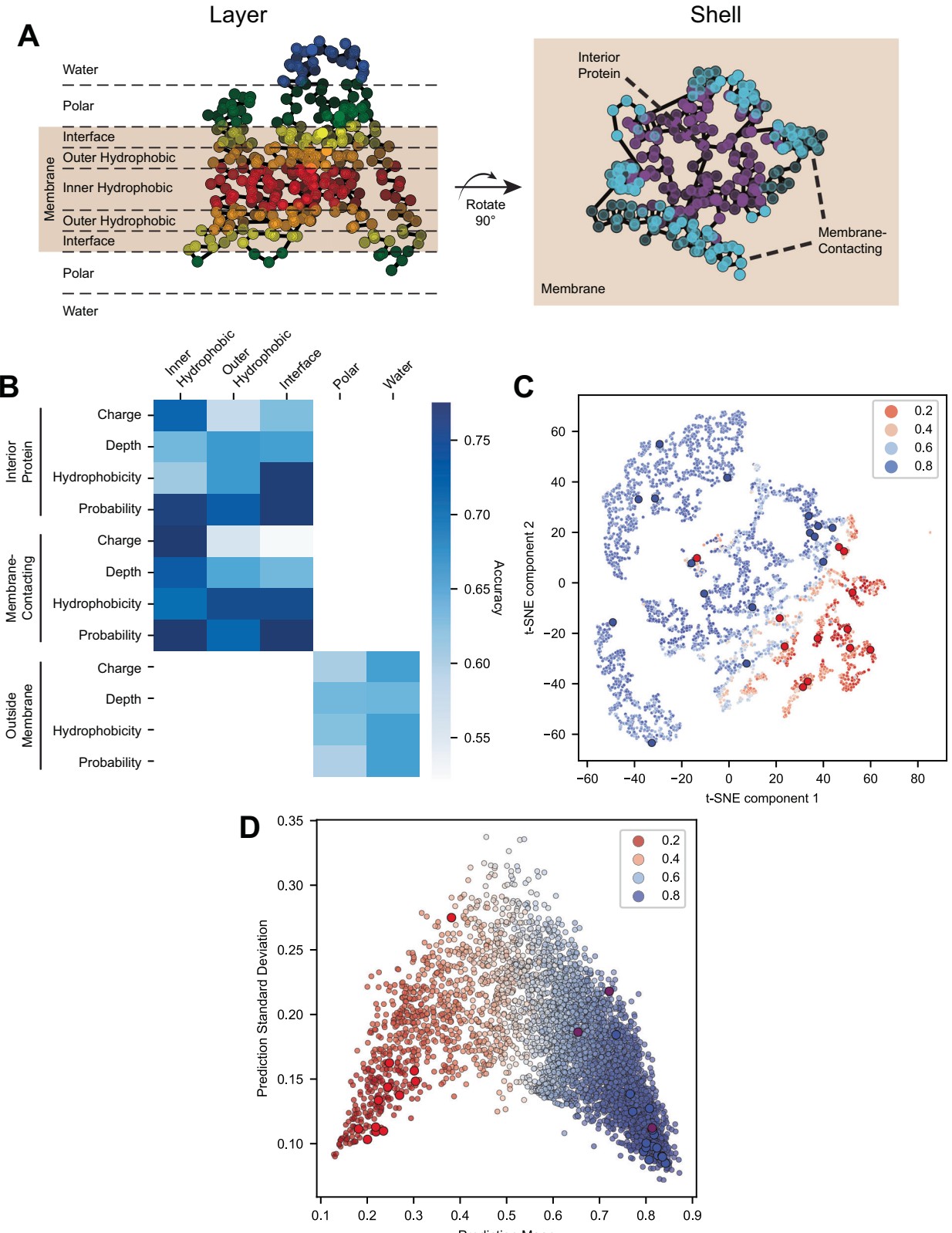

(Method 21). For every embedding, 50 classifiers were trained using different subsets of the labeled proteins to make ensembles of classifiers, termed Ensemble Classifier. The accuracy of each Ensemble Classifier is based on the average prediction accuracy of all classifiers in the ensemble, predicting only on proteins in each classifier's test set. Using the accuracy of the Ensemble Classifier as our comparative metric, we found that the hydrophobicity and probability values of the

intra-membrane layers (Fig. 5A, left panel) and the membrane-contacting shell (Fig. 5A, right panel) were most informative for separating the classes (Fig. 5B, Supplementary Fig. 16). The best-performing Ensemble Classifier had an accuracy of 84.6%.

Furthermore, we combined the Ensemble Classifiers to predict the likelihood of successful synthesis of membrane proteins that have not been tested (Method 22). An example of the predictions from one

**Fig. 5 | Structural features predict successful membrane protein synthesis in artificial environments. A** Protein structures are decomposed into bins based on amino acid location relative to a simulated membrane (Method 20). Classification is based on the vertical position and whether an amino acid contacts lipids or is buried within the protein. Left panel: All possible layers are shown, with the shaded region indicating the membrane interior. Right panel: The protein is rotated 90° to display only amino acids within membrane layers; those in polar or water layers are labeled as part of the external shell. **B** Classifiers trained on tSNE embeddings of membrane-contacting amino acids achieved the highest accuracy (Method 21). Embeddings were generated using various feature combinations, and the maximum accuracy of each Ensemble Classifier is shown. Although all combinations were evaluated, only pairs of features are displayed to highlight the specific contributions of each feature. **C** The tested proteins are well dispersed among the other proteins in the dataset when plotted in one of the embedded spaces used in the top classifier. Red points indicate the proteins that were not successfully made. Blue indicates that they were made. The hue of the remaining points indicates the predicted probability of synthesis success using the top classifier. 4,612 membrane proteins. **D** All 10 paired classifiers with an accuracy >83%, show strong prediction agreement for proteins that are predicted to have a high likelihood of successful synthesis. The mean of the predictions is plotted versus the standard deviation of predictions. Hue indicates the average predicted label for each protein. Red points indicate the proteins that were not successfully made. Blue indicates that they were made. Purple indicates 3 additional proteins that were selected based on the predicted outcome, and all 3 passed the threshold to be considered successfully produced. 4612 membrane proteins were included in the full predicted set shown in this plot. Source data are provided as a Source Data file.

combined Ensemble Classifier illustrates that the tested proteins cover much of the embedded plane (Fig. 5C). Using the combined Ensemble Classifier approach (Method 22, Supplementary Fig. 13 and 14), we find that 39.9% of the proteins in the dataset are labeled as likely synthesizable (prediction standard deviation < 0.2 and prediction mean > 0.7) while only 5.9% are likely not synthesizable (prediction standard deviation < 0.15 and prediction mean < 0.3) (Fig. 5D, Supplementary Data File 1). To further assess the validity of the predictions, we attempted to produce 3 proteins, AuxT1, FITM2, and PPT1, that were predicted to be successful. One has a low standard deviation among predictions (AuxT1), and two have a higher standard deviation (FITM2 and PPT1). All 3 proteins passed the threshold for successful solubilization (Fig. 5D, purple markers). Our results produce the first model to predict successful CFPS synthesis of membrane proteins.

## Discussion

This study introduces MEMPLEX, a high-throughput and AI-guided approach for rapidly designing artificial synthesis environments. MEMPLEX leverages active learning for targeted optimization of artificial synthesis environments. Our data showcases how insights gained from the synthesis of one protein inform the optimization of others, showing preliminary evidence of cross-protein learning. We also provide a method to describe membrane proteins based on their structure and use these features to predict successful protein synthesis in artificial environments.

Our platform lays the groundwork for future exploration: while we investigated a limited set of five variables, other conditions may influence the optimality of artificial synthesis environments. Investigating other membrane compositions that closely mimic native membranes, by including cholesterol, alternative lipid head groups such as phosphatidylethanolamine or phosphatidylserine, and acyl-chain lengths with varying levels of saturation, could broaden the scope and yield of our platform[48,49]. Additionally, other small molecules, such as spermidine, have the potential to impact the synthesis and insertion process. Functioning similarly to magnesium, spermidine acts as a stabilizer for RNA secondary structure and likely interacts with the negatively charged phosphates of the membrane, influencing the synthesis and insertion process[50]. Modulating plasmid concentration could provide another avenue to tune translation rates by limiting the abundance of transcription templates[6]. Prepositioning the Sec translocon in the system emerges as a promising strategy. Co-expression of the translocon during target membrane protein synthesis diverts resources and is likely constrained by its limited concentration. Bulk purifying the translocon and incorporating it into the membrane ahead of time could enhance synthesis efficiency[31,51]. Additional parameters, such as the addition of glutathione and disulfide isomerase, could enable the proper formation of disulfide bonds[52]. In addition, replacing liposomes with polymer nanodiscs and hybrid vesicles, could enhance the synthesis of membrane proteins[34–36]. Polymer nanodiscs have advantages over traditional

liposomes, including better size control and stability[53,54]. Beyond the chemical environment, modification of the DNA template to increase the translation initiation rate could lead to successful membrane protein synthesis[55]. Further expansion of MEMPLEX to encompass more proteins may generate insights into their respective response to given synthesis conditions. Exploration of additional protein features could shed light on the intricate interplay between proteins and reaction conditions, enabling the extension of the model to previously unexplored proteins.

While this work focused on the solubilization of membrane proteins into lipidic environments, the next step is to evaluate protein functions. As we sought to produce proteins that had not previously been synthesized, there were no available functional tests for most of the proteins. Our low throughput Western Blot and flow cytometry data further corroborate that the proteins were successfully inserted into the membrane. We also showed the presence of the specific functional motifs of CD9 and CD81 for antibody binding. To further demonstrate and quantify protein function, we synthesized the light-responsive ion channel bacteriorhodopsin using the MEMPLEX system, and quantified its light response in an ion flux test (Supplementary Fig. 6D and 6E, Method 24). However, future work will need to develop tailored assays to validate the function of these proteins.

In our recent efforts, we expanded the number of parameters from 5–8 and tested an additional 12,000 reactions, equating to a similar number of conditions tested per protein as in our previous screen. While including these new parameters leads to increased yields in some proteins, the active learning model struggled to capture the data, likely because the search space expanded dramatically, but the number of tested reactions remained the same (Supplementary Fig. 17). Expansion of the search space will likely require a larger sampling of the search space or multiple rounds of learning to capture all the variable interactions. In addition, while cell-free protein synthesis provides several advantages over cell-based synthesis, there are key limitations in its ability to make membrane proteins. The expression of some functionally folded proteins can require specific post-translational modifications such as disulfide bonds and glycosylation. While systems to implement these modifications have been demonstrated in cell-free systems, integrating these modifications with membrane protein synthesis is potentially nontrivial. The difference in the cost and yield obtained when producing proteins using cells versus cell-free systems is substantial when scaling up production. It is worth noting that while the reaction volumes produced in this work are small, protein concentrations of 10–100 μg/mL are sufficient for subsequent cell-culture tests[56], which require 0.05-5 μM (equivalent to 0.1–10 pmol reported in this study).

By offering an accelerated, high-throughput, and scalable approach, MEMPLEX tackles the decades-old challenge of designing artificial environments for producing membrane-incorporated membrane proteins. Our extensive dataset yields a wealth of information for understanding the influence of environmental factors on

membrane protein synthesis, specifically the first critical step of solubilization in the presence of liposomes. Our machine-learning algorithms also make that data actionable for designing artificial synthesis environments. MEMPLEX can be further refined with broader parameter search spaces, additional proteins, and larger reaction screens. The possibility of accurately predicting artificial synthesis environments for any given membrane protein will open unprecedented avenues for identifying new therapeutic targets and propelling drug innovation. Our work may also catalyze advancements in understanding, designing, and controlling intricate protein-chemical-lipid interactions within both natural and artificial synthesis environments.

## Methods

### M1: Construction of plasmids

All membrane protein plasmids were cloned using the EcoFlex MoClo Golden Gate cloning standard[57,58]. All membrane proteins were cloned into a standard level 1 transcription unit plasmid (pTU1-A-RFP) with a fragment encoding the T7 promoter and a strong ribosome binding site (EXPpro_RBS) and another encoding the GFP11 tag and the T7 terminator (G11_T7term). The membrane protein sequences were either PCR amplified from the plasmids graciously provided by Bruni et al. or synthesized as gene fragments prior to assembly into the final plasmids. Membrane proteins that were synthesized were codon optimized by the Twist Biosciences codon optimizer using the *E. coli* codon usage table. The plasmids used for purification of the GFP1-10 fragment and the standard soluble protein Chloramphenicol acetyltransferase were made from the standard pTU1-A-RFP plasmid, but they were modified using Gibson Assembly to encode for the lacI gene under a lactose inducible promoter and termed pTU1-L-RFP. The Chloramphenicol acetyltransferase expression plasmid used the Pro_RBS_6xHis sequence to add an N-terminal Histidine tag to the protein. The SecY and SecE plasmids used to co-express the translocon were cloned into the pTU1-A-RFP plasmid using the EXPpro_RBS promoter and the standard T7 terminator (T7_term). All sequences are listed in Supplementary Data 1.

### M2: Preparation of whole-cell extract

The strain BL21(DE3) Star was struck out onto a plate of 2xYT (Millipore Sigma Y2377) agar from a glycerol stock and allowed to grow overnight at 37 °C in a static incubator. Fresh colonies were used to inoculate 3 mL cultures of 2xYT media at 30 °C with shaking at 250 rpm overnight. The saturated overnight culture was then used to inoculate 300 mL of 2xYT at a starting concentration of 0.01 OD600. The culture was incubated at 30 °C, 250 rpm, until the OD reached 0.05-0.1. The culture is then induced with 0.4 mM IPTG (Millipore Sigma 11411446001) and grown until an OD600 of 1.0. Once the target cell density was reached, bacteria cells were harvested by centrifugation in 300 mL bottles at 4000 g for 20 min at 4 °C. The supernatant was then removed, and the pellet was resuspended with 20 mL of Buffer A (10 mM Tris-acetate (Thomas Scientific C993H97) pH 7.6, 14 mM Magnesium acetate (Millipore Sigma M0631), and 60 mM Potassium gluconate (Fisher Scientific AC229322500) and the centrifugation was repeated. This wash step was repeated to ensure that the cell pellet is clean. After the final wash and centrifugation, the pelleted cells were weighed and suspended in 1 mL of Buffer A supplemented with 2 mM DTT (VWR 97061-340) per 1 g of wet cell mass. To lyse cells by sonication, freshly suspended cells were transferred into 1.5 mL microtube and placed in an ice-water bath to minimize heat damage during sonication. The cells were lysed in an ice-water slurry using a Q125 Sonicator with a 2 mm diameter probe at a frequency of 20 kHz and 50% amplitude. Sonication was continued for about 27 cycles 10 s ON/ 10 s OFF. For each 0.5 mL sample, the input energy was ~1000 J. Cell lysates were centrifuged at 12,000 g for 20 min at 4 °C. The supernatant was collected and incubated at 30 °C for 30 min. The resulting WCE was aliquoted and stored at −80 °C.

All extracts were benchmarked against the previously used extract by expressing deGFP across several different magnesium glutamate (Millipore Sigma S595179) (8–16 mM) and potassium glutamate (Millipore Sigma G1501) (85–185 mM) concentrations. Each unique reaction composition was assembled in triplicate. The coefficient of variance (standard deviation / mean) between the replicates of each extract at each unique salt concentration was calculated to assess their similarity. Extracts with an average coefficient of variance of >10% were remade to ensure consistency of results.

### M3: Preparation of cell-free protein synthesis reaction supplement

The concentrations of the reaction conditions being screened in bulk cell-free protein synthesis reaction supplement were intentionally decreased to the lowest value found to still permit protein synthesis so that each of those concentrations could be increased during the droplet printing reaction assembly process. The base concentration of all components in the reaction supplement were as follows: 1.2 mM each of ATP (Promega E6011) and GTP (Promega E6031); 0.85 mM each of UTP (Promega E6021), and CTP (Promega E6041); 34 µg mL$^{-1}$ folinic acid (Millipore Sigma F7878); 170 µg mL$^{-1}$ of *E. coli* tRNA mixture from *E. coli* MRE600 (Roche 10109541001); 2 mM for each of the 20 standard amino acids (Millipore Sigma LAA21); 0.33 mM NAD (Roche NAD100-RO); 0.27 mM CoA (Sigma Aldrich C4282); 4 mM spermidine (Sigma-Aldrich S0266); 2 mM Dithiothreitol (DTT, Cleland's reagent) (VWR 97061-340): 85 mM potassium glutamate (Millipore Sigma S595179); 8 mM magnesium glutamate (Millipore Sigma S595179); 50 mM HEPES pH 7.6 (Neta Scientific RPI-H75030); 67 mM creatine phosphate (Roche 10621714001); 80 µg mL$^{-1}$ Creatine Kinase (Roche 10127566001); 0.64 mM cAMP (Sigma-Aldrich A6885); 0% PEG8000 (Sigma-Aldrich P2139).

### M4: Establishing the droplet printing robot

The core of the droplet printing system is the same as the previously documented version[18]. The major changes are described here: 1) we upgraded the robotic arm from the Dobot Magician to the Dobot M1 (Dobot). 2) Added a custom anti-evaporation system that is comprised of an acrylic sheet with a hole cut through the center which guides that interface with the robot arm to position the hole directly below the printer. The plate moves around with the arm to expose only the well below the printer head. 3) The printing pressure calibration was simplified. The printer initially sets the pressure to the average required pressure and then prints numerous test droplets onto the on-deck microbalance. The mass measurement from the droplets are used to calculate the droplet volume. The pressure is changed in proportion to the difference in the observed volume and the desired volume. This process is repeated until the desired droplet volume is achieved. 4) Additional modifications to the code are documented in the GitHub repository: https://github.com/ccmeyer/printing-platform.

### M5: Droplet printing to assemble reactions

This work utilized the custom droplet printing robot detailed in our earlier works[17,18]. The system utilized microfluidic adaptive printing for ultra-low volume liquid handling. Reagents were loaded into a disposable multilayer microfluidic container. These containers consisted of a 3D-printed body, containing a reservoir and a printing chamber, with two laser-cut layers attached to the bottom. The first attached layer connected the reservoir to the printing chamber with a thin, high-resistance channel. The bottom layer contained a single 65-µm diameter hole positioned below the printing channel. The reagent was stored in the reservoir and the application of pressure to this side of the container pushed the reagent through the high-resistance channel into the printing chamber. A short burst of pressurized air applied to

the printing chamber caused the reagent to push its way through the small hole in the bottom layer and is ejected as a single droplet. Pressure applied to the reservoir side replenished the volume in the printing chamber that was lost during the print, allowing the process to repeat. The pressure was generated using peristaltic pumps, and both the refueling and printing pressures were adjusted to match the reagent. The droplet volume was calculated using the mass of 100 droplets recorded by the on-deck microbalance and the known density of the reagent. The printing pressure was then modulated to achieve the desired droplet volume. The refueling pressure was then modified to ensure a consistent volume in the printing chamber. Tuning of the pressures allowed for the printing of accurate droplet volumes for a variety of reagents ranging from water to concentrated liposome solutions or PEG8000. To dispense into specific wells of a 384-well plate, these cartridges were manipulated in space using a robotic positioning system. The target volume for each well was achieved by adding the required number of droplets to reach the target volume.

### M6: Preparation of GFP1-10 fragment

The protocol for the purification of the GFP1-10 fragment was derived from the work by Waldo et al.[25]. The strain BL21(DE3) harboring the pTU1-L-GFP1-10 plasmid was struck out onto a plate of LB agar + 100 μg/mL carbenicillin (Millipore Sigma C3416) from a glycerol stock and allowed to grow overnight at 37 °C in a static incubator. Fresh colonies were used to inoculate a 3 mL culture of LB media+100 μg/mL carbenicillin and incubated at 37 °C with shaking at 250 rpm overnight. The full 3 mL overnight culture was used to inoculate a 2 L flask with 500 mL of LB + 100 μg/mL carbenicillin. The flask was incubated at 37 °C with 250 rpm shaking. After 2 h, 1 M IPTG (Millipore Sigma 11411446001) stock solution was added to reach a final concentration of 1 mM. The flask was incubated in the same conditions for an additional 5 h. Cells were then collected by centrifugation at 4000 × g for 30 min. The supernatant was discarded, and the pellet was resuspended in 15 mL of TNG buffer (TNG buffer contains 100 mM Tris-HCl (Sigma Aldrich10708976001) (pH 7.4), 150 mM NaCl (Sigma Aldrich S9888), and 10% (v/v) Glycerol (Sigma Aldrich G7893)). The cells were sonicated for 2 min in an ice-water slurry using a Q125 Sonicator with a 4 mm diameter sonication probe at a frequency of 20 kHz and 50% amplitude. This sonication was repeated 2 additional times with a 2 min break in between. The cell debris were pelleted by centrifugation at 25,000 × g for 30 min at 4 °C. The supernatant was discarded again, and the pellet was resuspended in 5 mL of BugBuster Protein Extraction Reagent (Novagen 70584-M). The suspension was then centrifuged for 10 min at 10,000 × g, and the supernatant was discarded. This wash process was repeated 3 additional time using BugBuster. After the last BugBuster wash, the wash was repeated using TNG buffer 2 times. The final pellet mass was measured, and the pellet was resuspended to a concentration of 75 mg/mL in TNG buffer. The suspension was aliquoted into 1.7 mL microfuge tubes. The tubes were then centrifuged at 16,000 × g for 10 min at 4 °C to pellet the protein, and the supernatant was discarded. The pellets were stored at -80 °C for up to 6 months.

The working stock of GFP1-10 used to bind GFP11 tagged proteins was prepared as follows: One GFP1-10 inclusion body pellet was dissolved in 250 μL of 9 M urea (Sigma Aldrich U5378) and 1.25 μL of 1 M DTT (VWR 97061-340). The pellet was broken up using a pipet tip. Incubation of the pellet in a 37 °C water bath helped dissolve the inclusion bodies. The suspension was centrifuged for 1 min at 16,000 × g to remove aggregated material. 200 μL of the resuspension was gently transferred to 10.2 mL of pre-warmed TNG buffer. The suspension was mixed by slowly inverting the tube. The solution was filtered using a 0.22 μm syringe filter (VWR 76479-024). For the membrane protein binding assays conducted in this paper, the stock solution was diluted 4-fold prior to use. Working stocks were used on the day of preparation, as it was found that the fluorescence diminished substantially over subsequent days.

### M7: Preparation of purified chloramphenicol acetyltransferase

The protocol for the purification of the soluble His-tagged protein is also derived from the work by Waldo et al.[25]. The strain BL21(DE3) harboring the pTU1-L-6xHis-Cat-GFP11 plasmid was struck out onto a plate of LB agar+100 μg/mL carbenicillin from a glycerol stock and allowed to grow overnight at 37 °C in a static incubator. Fresh colonies were used to inoculate a 3 mL culture of LB media+100 μg/mL carbenicillin at 37 °C with shaking at 250 rpm overnight. A 2 L flask was inoculated with 300 mL of LB + 100 μg/mL carbenicillin with the full 3 mL overnight culture and incubated at 37 °C with 250 rpm shaking. Once the OD600 reached 0.5, IPTG was added to reach a final concentration of 1 mM. The culture was incubated for an additional 4 h. Cells were collected by centrifugation at 4000 × g for 30 min. The supernatant was discarded, and the pellet was resuspended in 15 mL of TNG buffer and transferred to a pre-weighed 50mL conical tube. The pellet was spun down for another 20min at 4000 × g, and the supernatant was discarded. The remaining pellet was stored overnight at −80 °C. The next day, the pellet was resuspended in 1 mL of Equilibration buffer (TNG buffer with 10 mM of Imidizole (Sigma Aldrich I202) and 300 mM of NaCl) per 1 g of cell mass. The cells were aliquoted into 1.5 mL tubes and lysed in an ice-water slurry using a Q125 Sonicator with a 2 mm diameter probe at a frequency of 20 kHz and 50% amplitude. Sonication was continued for about 27 cycles 10 s ON/ 10 s OFF. For each 0.5 mL sample, the input energy was ~1000 J. The cell debris was pelleted by centrifugation at 20,000 × g for 20 min at 4 °C.

To purify the histidine-tagged protein from the lysate, 1 mL of HisPur Ni-NTA Resin (ThermoFisher 88221) for 1 mL of sample was equilibrated. The resin was spun down at 700 × g for 2 min, and the supernatant was discarded. The resin was resuspended using 2 resin bed volumes of Equilibration buffer. This spin and wash step was repeated. The cell lysate was mixed with an equal volume of Equilibration Buffer prior to addition to the resin. The resin and lysate were incubated together on a rotary shaker for 30 min at room temperature. The resin was then pelleted at 700 × g for 2 min, and the supernatant was removed. Two resin bed volumes of Wash Buffer (TNG Buffer with 10 mM of Imidizole) were used to wash the resin. This wash step was repeated 3–4 times. Elution of the protein was performed by adding one resin bed volume of Elution Buffer (TNG Buffer with 150 mM Imidizole). The eluted protein was then dialyzed overnight in 1 L of TNG buffer.

### M8: Membrane protein yield quantification

The fluorescence values from the GFP solubilization reporter were measured with the Synergy H1 Multimode Microplate Reader (BioTek) using 485 nm excitation and 515 nm emission. These values were converted to an amount of protein using a calibration curve. This calibration curve was made from multiple dilutions of a soluble protein with the GFP11 tag (Supplementary Fig. 3). The soluble protein used in these experiments is Chloramphenicol acetyltransferase (Method 7). It was bulk purified and quantified using the Pierce BCA protein assay and a Bovine Serum Albumin calibration curve (Thermo Scientific 23227). The soluble protein was added to a 384-well plate at known dilutions. The volume added was 2 μL to match the volume of the cell-free reactions. The large GFP fragment was added to the calibration curve wells at the same time that they were added to the experimental wells. The calibration curve was generated for each experiment as it was observed that fluorescence varied between working stocks of the large fragment (Supplementary Note 3). The calibration curve, as shown in Supplementary Fig. 3, follows a sigmoidal trend. Hence, the following function was used to define the curve that converted the fluorescence values to protein concentration. The following equation was used to generate the calibration curve: $y = \frac{L}{1+e^{-k(x-x_0)}} + b$. The

tunable parameters L, k, $x_0$, and b were fit to the data using the Scipy optimize curve_fit function[59].

## M9: Complete protein synthesis protocol

All high-throughput experiments utilized small volume, flat bottom, round well, 384-well plates (Greiner). Prior to addition of any reagents to the well plates, 20 μL of 20 mg/mL Bovine Serum Albumin (BSA) (Sigma Aldrich A4503) was added to each well and allowed to block the wells for 30 min at room temperature. The BSA was then aspirated out of all wells. The base cell-free protein synthesis master mix was made by mixing the reaction supplement at 2.74x the working concentration with the whole cell extract, which was at 3.7x the working concentration. Once these reagents were gently but thoroughly mixed, they were aliquoted into all required wells using a multi-channel pipet. Once the cell-free mix was added, the plate was loaded onto the droplet printing robot. The reagents that were to be added to the reaction were sequentially loaded into the robot and printed into the desired wells. The maximum final reaction volume was set to 2 μL. To avoid any bias due to the printing, the location of all reactions was randomized so that any variability introduced during the reaction assembly process was captured in the variance of the replicates.

Once all reagents were added, the plate was spun down at 200 × g for 2 min to ensure that all reagents were collected at the bottom of the well. The plate was then sealed using optical adhesive film (Applied Biosystems 4311971) and placed in a shaking incubator set to 30 °C with 300 rpm shaking. The reaction was run overnight. The following day, the large GFP fragment was added to the reaction to evaluate the yield of solubilized protein. To do this, the working stock of the GFP1-10 fragment was prepared fresh (Method 6) and then added to each reaction. The GFP1-10 stock was added to the 2 μL reaction to reach a final volume of 20 μL. The 18 μL of the stock solution were added using an adapter for the droplet printer, which used the same pneumatic control to dispense large volume droplets from a pipet tip. Once added to all reactions, the plate was once again sealed with the optical film and allowed to incubate overnight at 30 °C with 300 rpm shaking. The following day, the fluorescence was measured, and the amount of protein synthesized was quantified as described in Method 8.

## M10: Liposome preparation

Liposomes were prepared using a standard sonication method. Lipids (DMPC (Avanti Polar Lipids 850345), DPPC (Avanti Polar Lipids 850355), and DOPC (Avanti Polar Lipids 850375), from Avanti Polar Lipids) suspended in Chloroform (Millipore Sigma 1024441000) at a concentration of 25 mg/mL, were transferred to a glass vial using a glass syringe. When preparing liposomes for size exclusion chromatography or flow cytometry, 18:1-PC Cy5 labeled lipids (Avanti Polar Lipids 810335) were added to the lipid:chloroform suspension to reach a final concentration of 0.05 mol%. The lipid suspension was then dried under a stream of nitrogen to create a thin layer of lipids on the glass surface. The glass vial was placed in a desiccator overnight to ensure the removal of all residual chloroform. The next day, lipids were rehydrated using Liposome Buffer (50 mM HEPES-KOH (Neta Scientific RPI-H75030) pH 7.6, 8 mM Magnesium Glutamate (Millipore Sigma S595179) 85 mM Potassium Glutamate (Millipore Sigma G1501), 158 mM Potassium Phosphate Monobasic (VWR BDH9268)) to reach 25 mg/mL. The dried lipid was slowly resuspended by pipetting and light vortexing until all the lipids were removed from the glass. Liposomes were formed by sonicating the resuspended lipids in a water bath using a Q125 Sonicator with a 4 mm diameter sonication probe at a frequency of 20 kHz and 50% amplitude. Sonication was continued for about 34 cycles 10 s ON/ 10 s OFF, to reach an input energy of ~1250 J. The liposome solution was then centrifuged at 16,000 × g for 3 min. The clear liposome solution was then transferred to a fresh tube. Each liposome solution was used the same day they were prepared. Liposomes were analyzed using Nanoparticle Tracking Analysis

(Method 10) and Cryo-electron microscopy (Method 11) to verify particle count, size distribution, and lamellarity.

## M11: Nanoparticle Tracking Analysis (NTA) of liposomes

The size and concentration of the liposomes were measured with nanoparticle tracking analysis (NTA) using the ZetaView (Particle Metrix). Samples were prediluted in water to an optimal concentration to allow for about 150 particles/frames (ZetaView v.8.05.12). For each measurement, 11 positions were scanned for two cycles using the following parameters: camera sensitivity: 92, shutter: 200, frame rate: 30, and cell temperature: 25 °C.

## M12: Cryo-EM imaging

Freshly prepared liposomes were diluted from 25 mg/mL to 1 mg/mL prior to imaging. 5 μL of sample was applied to a glow-discharged (25 mA, 30 sec) holey carbon grid (300 mesh Quantifoil 1.2/1.3) and allowed to incubate for 15 min. Then, the grid was loaded into the Leica EM GP2 plunger, where the grid was blotted from the back and then front for 4 s each. Another 5 μL of sample were applied to the grid and given 5 min to incubate. The grid was then blotted from the front for 4 s, followed by rapid plunge freezing into liquid nitrogen-cooled liquid ethane. Cryo-EM images were acquired at 200 kV on a Thermofisher Glacios electron microscope equipped with a Gatan K3 direct electron detector, available at the UC Davis BioEM Core Facility. Micrographs were recorded at 56,818x (0.88 Å/pixel) magnification using K3 with a dose of 40 e/A^2. Parallel beam illumination and coma-free alignment were applied using SerialEM.

## M13: Size exclusion chromatography

Size exclusion chromatography was performed using qEVsingle Gen2 columns (IZON Science ICS-70). The protocol follows the manufacturer's instructions for using the columns with gravity flow. Cell-free protein synthesis reactions were assembled in microfuge tubes and incubated at 30 °C with 300 rpm shaking overnight. The columns were equilibrated by allowing the storage buffer to exit the column. Two column volumes of 1x PBS (Grainger RPI-P32060) that had been freshly sterilized with a 0.22 μm filter (VWR 76479-024) were added to the column and allowed time to pass through the column. 50 μL of cell-free reactions were mixed with 100 μL of PBS to bring the sample volume to 150 μL. The cell-free mix was added to the column. Additional PBS was added to the column in steps to allow for separate fractions to be collected. Cy5-labeled liposomes were used in all chromatography runs so the elution of the liposomes could be monitored via plate reader measurements. In general, the liposomes eluted most between 1.1 and 1.6 mL, including the sample volume.

## M14: Single liposome flow cytometry

A CytoFLEX flow cytometer (Beckman Coulter) equipped with four lasers was used for all flow cytometry experiments. Prior to sample analysis, we employed a calibration procedure utilizing standard beads and FCMPASS software, as described recently by Welsh et al.[60,61]. As all liposomes were labeled with Cy5-lipids, the red laser with a 660 nm filter was used for triggering. The threshold for triggering was set above the background Cy5 signal seen in the diluent. Samples were diluted to reach an event rater of 2000 events s⁻¹ on the lowest flow rate of 10 μL min⁻¹. Generally, samples were diluted 50-fold using 0.2 μm-filtered PBS. The gain of the various channels was optimized by analyzing 8 Peak Rainbow Calibration Beads from Spherotech (Cat. RCP-30-5A). The gain was increased incrementally from 25–3000 in each channel. The gain that best separated the dimmest bead population from background, calculated by stain index ((MFIbeads-MFI-background)/Stdevbackground), was chosen for further studies. For all samples, 100,000 events were recorded and used for analysis. No gating was applied through the flow cytometer. During the analysis of the flow cytometry data, liposomes were identified using the ratio of R-

660-20-A (Cy5-area signal) and SSC-A (side scatter area). Events that had a ratio below 0.5 or above 5 were removed from the dataset.

The flow cytometry samples were analyzed using the area of the red laser with the 660 nm filter signal (R-660-20-A) to measure the liposome-Cy5 signal and the area of the blue laser with the 525 nm filter signal (B-525-40-A) to measure the protein-GFP signal. The normalization of the protein signal by the liposome signal was calculated using the following equation to calculate the normalized protein signal: log(B-525-40-A / R-660-20-A). This ratio was calculated for each observed event in each flow cytometry sample.

Flow cytometry data collected in Supplementary Fig. 18 was collected on Flow Cytometer Cytek® Aurora 5-laser spectral 40+ color (with nanoparticle detection upgrade). Samples were run as described above. For CD9 detection, a 1:10 dilution of obtained CD9-proteoliposomes obtained from cell free-reactions was performed. The antibody (FITC anti-human CD9 Antibody, Cat# 312103, LOT B392948, Clone HI9a, BioLegend) staining was done by incubating 8uL of diluted sample with 2 μL of 100 μg/mL antibody stock for 2 h on ice (1:5 antibody dilution, or final concentration 20 μg/mL). For CD81 detection, a 1:100 dilution of cell-free reactions was done. The antibody (FITC anti-human CD81 (TAPA-1) Antibody, Cat# 349503, LOT B342884, Clone 5A6, BioLegend) staining was done by incubating 10 μL of diluted sample with 0.5 μL of 200 μg/mL antibody stock for 2 h on ice (1:20 dilution, or final concentration of 10 μg/mL).

## M15: Reaction selection for condition screening

The 5 factors that were modulated in this study (Magnesium Glutamate (Millipore Sigma S595179), Potassium Glutamate (Millipore Sigma G1501), Polyethylene glycol 8000 (Sigma-Aldrich P2139), and SecYE plasmid concentrations and lipid type) were entered at their discretized concentrations (e.g., high, medium, low) into the pyDOE2 generalized subset design function[32] and used to reduce the required reactions by 3-fold.

## M16: Linear model fitting and variable interdependence calculation

All the values that represent the reaction conditions used in a specific reaction were scaled using a min-max scaler, defining the highest included concentration as 1 and the lowest as 0. The yields from the reactions are used as data labels. To account for the variability in the range of yields observed across different proteins, the yield seen for each protein was scaled using a unique Z-score scaler to bring the mean of all yields for a given protein to 0 and a standard deviation of 1. The model's pairwise interaction terms were calculated by multiplying the concentrations. The summation of all single and pairwise interaction terms was used as the linear model. We used Ordinary Least Squares (OLS) regression through the Statsmodels python package[62] to fit the data. A model was trained for each protein separately. The p-value for each term was calculated using a t-test comparing the model's error with and without the term included.

## M17: Data preparation, model fitting, and active learning reaction selection

Prior to model training, the dataset was cleaned up to eliminate outliers. Any reaction that had a yield that was at least two standard deviations away from the median of all replicates was eliminated from the set. All reactions were tested in at least quadruplicate so that if an outlier was removed there would still be at least three replicates for downstream statistical testing. Out of all reactions, 7.5% were flagged as outliers and removed from the data set. All the values that represented the reaction conditions used in a specific reaction were scaled with a min-max scaler. The protein used in each reaction was entered into the model as one hot encoded columns, one column for each unique protein. The yields from the reactions were used as the labels for the dataset. To account for the range of possible yields across

different proteins, the yield seen for each protein was scaled using a unique Z-score scaler to bring the mean of all yields for a given protein to 0 and a standard deviation of 1.

This dataset was then used to train an ensemble of deep neural networks. All neural networks were implemented using Keras from TensorFlow[63]. The NN could have one of three architectures. All used a sequential layer structure composed of an input layer, several densely connected layers with a ReLu activation function, and a final single node dense layer using a linear activation function. All architectures were compiled using the mean absolute error loss function and the Adam optimization method with a learning rate of 0.001. Architectures differed in the number of densely connected layers. The simplest model had 4 layers with 128, 64, 32, and 16 nodes. The next architecture used the same 4 layers but added a 256 node layer at the start. The final added an additional 512 node layer at the start, totaling 6 densely connected layers. Models were trained for 100 epochs each. Each unique reaction was treated as a data point, meaning replicates of the same reaction condition were entered as separate instances. The available data was split into 5 different train and test sets at an 80:20 ratio. All replicates of a single reaction composition were kept in either the training or the testing set when training the neural networks.

After training, the models were used to predict the reaction conditions that were included in their specific test set to assess the accuracy of the models. The models were then used to predict all possible combinations of reaction components (432 combinations). All the predictions for a given reaction condition were grouped together and used to calculate the mean predicted yield and the standard deviation of the predictions. The metric used to score the reactions was calculated by subtracting the mean predicted value from the standard deviation. The reactions with the highest score were chosen for testing.

## M18: Reaction condition embedding

Given that all proteins were not evaluated for all the same reaction conditions, we used the ensemble predicted responses for the reaction condition embedding. We generated the same 45 model ensemble (Method 13) but, this time, trained the models using the data from the screening and active learning reactions. These 45 models were then used to make predictions on the outcome of all proteins in the 432 possible reaction conditions that were available during active learning. All the predictions for a single protein using one model were regarded as a unique observation so the predicted outcome of each unique reaction was pivoted to be a new column. This resulted in a matrix that was 1260 (45 models x 28 proteins) rows by 432 columns.

Prior to feature embedding, all feature values were Z-scored. Once the values were standardized, they were transformed using Scikit Learn's T-distributed Stochastic Neighbor Embedding (t-SNE) function to the resulting embedding values[64]. All included variables were converted to 2 components, and the perplexity value for t-SNE algorithm was set to the standard value of 30. This resulted in the projection of each model's approximation of each protein. The location of all proteins was averaged to identify the centroid of the protein predictions, and the standard deviation between the positions of each model's projection was calculated.

## M19: Assembly of the membrane protein structure dataset

The data used in the protein structure dataset was assembled from two sources: Uniprot[46] and AlphaFold[44,45]. The sequence annotations, specifically for the transmembrane helices, were taken from Uniprot. Proteins were queried using the keyword "transmembrane" and limited to the reviewed Swiss-Prot entries. The proteins in this set were further limited to alpha-helical proteins that had more than one transmembrane domain and belonged to *E. coli*, *A. thaliana*, or *H. sapiens*. The corresponding AlphaFold entry number was pulled from the Uniprot entry and used to download every structure in the list.

Each structure then had to be oriented into a mock membrane environment. Proteins were oriented using the assumption that the transmembrane helices would have to span the membrane and adjacent transmembrane domains would have to be inverted (the end of transmembrane domain 1 would be on the same side of the membrane as the start of domain 2). The coordinates for all amino acids at the ends of each transmembrane helix were identified and averaged to find the center point of the helices on either side of the membrane. The middle of these two points was used to translate the structure to the origin of the coordinate plate. Then, the angle between the line connecting the center of the termini on either side of the membrane and the normal to the mid plane was used to create a rotation matrix to rotate the full structure. This resulted in a protein whose membrane-spanning region was centered at the origin, and the Z-plane cut through the middle of it[47].

To assess the accuracy of the protein orientation in the mock environment, the location label for all residues (inside or outside of the membrane) was compared to their coordinates in relation to the mock environment. Residues labeled as part of the transmembrane helices that were located >20 Å from the center line were labeled as incorrect. The overall percentage of correctly labeled residues and the percentage of residues labeled as outside the membrane but were inside the mock membrane were used to eliminate structures. The majority of structures were found to have >75% accurately labeled residues. The average z distance between termini of transmembrane domains was also investigated to eliminate annotations that are suspected to be impossible. Specifically, labels that would result in transmembrane domains shorter than 10 Å were removed.

## M20: Structure analysis and feature extraction

This section details how the complex protein structure was reduced to a set of features that seek to describe how these proteins interact with their surrounding environment. The first step was to define the locations where an amino acid would experience different interactions with its environment. These locations are defined by two different identifiers. One is the z-coordinate, which represents the location of the amino acid in the context of the membrane. The membrane is divided into 5 horizontal layers corresponding to water-exposed (>30 Å from the midline), polar, interface, outer hydrophobic, and inner hydrophobic (<6 Å from the midline) (Fig. 5A left panel). The other locational identifier is whether the amino acid is located in the interior of the protein, in contact with the membrane, or not in the membrane at all (Fig. 5A, right panel). A surface to represent the closest position of the surrounding lipids was calculated using a three-dimensional alpha shape[65]. This surface contours to the shape of the membrane protein by identifying the outermost amino acids based on their proximity to the surrounding amino acids. The outer amino acids are presumed to be in contact with the membrane. The alpha shape is dependent on the parameter α, which determines how tightly the surface fits to the protein. This value was scaled in relation to the size of the protein to ensure the surface matched the protein well. The bounding surface that was located outside of the membrane was extended far from the protein. The shortest distance between each amino acid and this surface was calculated. A distance of <4 Å was set as being in contact with the membrane, while the others were labeled as inside the protein. Every amino acid was identified as a part of both a layer and a shell.

Next, a value had to be associated with each amino acid. Four different metrics were used for this purpose. The first was the charge of the amino acid. Arg, Lys, and His were scored as +1. Glu and Asp were scored as -1. All other residues were scored as 0. The next metric was the hydrophobicity of the residue. The scale was taken from Stephen White's lab[66]. The third metric was defined as the depth, which is how buried the residue is within the protein. This was calculated by identifying the number of other residues whose centroids were within a 6 Å

sphere around a given amino acid[47]. The final metric represents the likelihood of observing a specific amino acid in a specific location. This is calculated by looking at the frequency of finding a specific amino acid in a specific location (layer, shell, and depth) divided by the frequency of observing that amino acid in all locations across all structures in the dataset[47].

Every amino acid that falls into each location bin must be summarized to reduce all the amino acids in that bin to a number. The statistics used here are the mean, minimum, maximum, and sum. An example of a final feature is the mean of all hydrophobicities of the amino acids in the inner hydrophobic layer and in direct contact with the membrane. This results in 32 features ((3 layers in the membrane x 2 shells) + (2 layers outside of the membrane x 1 shell) = 8 locations x 4 metrics = 32 features).

## M21: Feature embedding and classifier training

Prior to feature embedding, all feature values were Z-scored. Once the values were standardized, they were transformed using Scikit Learn's T-distributed Stochastic Neighbor Embedding (t-SNE) function to the resulting embedding values[64]. All included variables were converted to 2 components, and the perplexity value for t-SNE algorithm was set to the standard value of 30. Proteins were classified as successfully synthesized if one reaction composition yielded a mean difference between the liposome and corresponding no liposome condition of 1 pmol and a corrected p-value of <0.05. The Benjamini–Hochberg method was used to calculate the false discovery rate and correct the p-value such that all reaction conditions could be assessed while mitigating the risk of type I errors. For each embedding, the labeled set of proteins was randomly divided up into 50 different train-test sets and used to train separate scikit learn Support Vector Machine (SVM) classifiers[64,67]. The SVM classifiers used a Radial Basis Function (RBF) kernel, and the regularization parameter C was set to 100. The classifiers were then used to predict the label of only the proteins included in the test set. Scoring for an embedding was based on the accuracy of all replicate classifiers trained on that embedding.

## M22: Classifier ensemble training and predictions

To further improve the accuracy of the Ensemble Classifiers, we increased the number of input variables to be included during training. We paired all the tSNE components that resulted in Ensemble Classifiers with accuracies above 77% with one another to create 190 new Ensemble Classifiers. Each classifier used the two tSNE components from each of the two included embeddings as well as the length of the protein. This resulted in 5 input variables for each protein during classifier training. The 12 Ensemble Classifiers that achieved an accuracy above 85% were included in a comprehensive Ensemble Classifier (Supplementary Fig. 13). All 600 different classifiers in this ensemble were used to predict each protein in the dataset. Each classifier arrived at a different prediction for each protein. The consistency of the predictions from the set of ensembles indicates confidence in that prediction (Supplementary Fig. 14).

## M23: Western blot characterization

Samples were loaded onto 4–20% Mini-PROTEAN TGX Precast Protein Gels (Biorad) and run at a constant voltage of 200 V for 60 min in prechilled 1x Tris/Glycine/SDS buffer (Biorad). While the protein gel was running, the Immun-Blot PVDF Membrane (Biorad) was soaked in methanol for 1 min until it became translucent and then soaked in Trans-Blot Turbo Transfer Buffer (Biorad) for at least 3 min. The SDS-PAGE gel was washed and placed into an assembly with the presoaked transfer stack (Biorad). The assembly was then inserted into the Trans-Blot Turbo Transfer System (Biorad) to conduct the transfer of the protein into the PVDF membrane. The preprogrammed "mini TGX" program was used for the transfer. After the transfer, the membrane was rinsed thoroughly with distilled water, and the membrane was

blocked using 5% non-fat dry milk solution/TBST (Blotting Grade Blocker Non Fat Dry Milk (Biorad)) for 1 h. The blocking solution was poured off, and the primary antibody (CD47 Monoclonal Antibody (B6H12, LOT 2836880) (Invitrogen)) was added, and diluted 1:1000 in 5% non-fat dry milk solution/TBST. The primary antibody was incubated overnight at 4 °C with gentle shaking. The antibody solution was poured off, and the membrane was washed with TBST twice. The membrane was blocked again using the 5% non-fat dry milk solution/TBST for an additional hour at room temperature with gentle shaking. The membrane was then rinsed an additional 3 times. The secondary antibody (Goat anti-Mouse IgG (H + L) Secondary Antibody HRP (Invitrogen) at a 1:25,000 dilution) was added to the membrane and incubated for 1 h at room temperature with gentle shaking. The membrane was then rinsed with TBST. The presence of the secondary antibody was detected using the Clarity Western ECL Substrate chemiluminescence kit (Biorad), and the gel was imaged using the Syngene PXi gel imager.

**M24: Assessment of bR protein function using the ACMA fluorescence assay**
Flux inside and outside buffers were prepared with 20 mM HEPES (RPI Research Products, 7365-45-9), 1 mM EGTA (Millipore Sigma, 324626-25GM), and either 450 mM KCl (Millipore Sigma, P3911) (inside buffer, pH 7.4) or 450 mM N-Methyl-D-glucamine (Millipore Sigma, M2004-100G) (outside buffer, pH 7.4). For liposome preparation, bR proteoliposomes were generated using cell-free synthesis reactions with varying plasmid concentrations (low, medium, high) in 10 μL reaction volumes, as described in Method 9. Reactions were incubated overnight at 37 °C with 250 rpm shaking, pooled, and purified by size-exclusion chromatography (Method 13). Vesicles were concentrated using 100 kDa Amicon filters (Millipore Sigma, UFC510008) and normalized by Cy5 fluorescence to ensure consistent liposome quantities. Control DMPC liposomes were prepared as outlined in Method 10, rehydrated with flux inside buffer, and purple membrane (PM) liposomes were prepared by incorporating 0.3 mg of commercial PM (Purple Membrane from *Halobacterium Salinarium*, Sigma Aldrich, 53026-44-1) into the same buffer for light-responsive positive controls. Proteoliposomes and control liposomes were incubated with 200 μM all-trans retinal (Millipore Sigma, R2500-100MG) for 2 h on ice in the dark before the ACMA flux assay. The assay was performed in a black 96-well plate with 2.5 μL of liposome solution and 47.5 μL of flux outside buffer containing 2 μM ACMA (Invitrogen, A1324) (excitation: 419 nm, emission: 490 nm). Following a 10-min equilibration, ion flux was initiated with 1 μL of 400 μM valinomycin (Thermo Fisher, V1644), and fluorescence was recorded during alternating 20-min light/dark cycles. Membrane potential collapse was induced with 1 μL of 0.4 mM CCCP (Millipore Sigma, C2759-100MG) and fluorescence was measured until equilibrium. Experimental groups included blank DMPC liposomes, PM liposomes (positive control), bR proteoliposomes (low, medium, high plasmid concentrations), and buffer-only controls. Fluorescence traces were analyzed to calculate slopes during valinomycin, light, and dark phases, normalized to blank DMPC liposomes. Slope recovery, defined as the product of the valinomycin-phase slope and the sum of light- and dark-phase slopes, was used to assess light responsiveness. Positive control data confirmed assay validity but were excluded from quantitative comparisons due to unknown bR content.

**Reporting summary**
Further information on research design is available in the Nature Portfolio Reporting Summary linked to this article.

## Data availability
All materials are available upon request. All data presented in the figures of this paper has been included in the Source Data File provided with this paper. All raw and analyzed data has been uploaded to Zenodo with the https://doi.org/10.5281/zenodo.15086208[68]. Source data are provided with this paper.

## Code availability
The code used to perform the analyses and generate results in this study is publicly available and has been deposited in GitHub at https://github.com/ccmeyer/Memplex_platform/tree/main, under the MIT license. The specific version of the code associated with this publication (v1.0.0) is archived in Zenodo and is accessible via DOI: 10.5281/zenodo.15086490[69].

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

## Acknowledgements
We thank Fei Guo from the UC Davis BioEM Core Facility for their technical assistance with the cryo-electron microscopy sample preparation and imaging. We thank Steven Lucero in the UC Davis Team lab for helping with the generation of the 3D printed components required for the droplet printer. We also thank Dr. Renato Bruni for providing plasmids expressing some of the membrane proteins. We appreciate the discussion of the paper with the members of the Tan Lab. The research is supported by NIH/NIGMS R35GM142788 (Tan), NIH/NIBIB 5R01EB034279 (Tan), NDSEG Fellowship (Henson), ARCS Fellowship (Meyer), and NIH eMCDB T32 (Arizzi).

## Author contributions
C.M. and C.T. conceived the MEMPLEX platform. C.T. supervised the project. M.L. consulted on the membrane-protein synthesis. S.A. consulted on the computational methods. A.W supervised the work by T.H. C.M. conducted all experiments and computations. A.A. assisted in generating the membrane protein synthesis reactions and functional testing of the proteins. T.H. performed western blot and nanoparticle tracking analysis, and aided the flow cytometry experiments. C.M., A.A., and C.T. wrote the paper.

## Competing interests
The authors declare no competing interests.
