## [Transparent Peer Review file · Nature Communications]

Designer artificial environments for membrane protein synthesis

Corresponding Author: Professor Cheemeng Tan

Version 0:

Reviewer comments:

Reviewer #1

(Remarks to the Author)

The authors describe a cell-free throughput screening process for the evaluation of membrane protein synthesis. The authors used a standard *E. coli* based CFPS batch system in 2 μ L reactions for throughput analysis. As variable expression optimization parameters, the concentrations of five reaction compounds (K⁺, Mg⁺, PEG, lipids (DOPC, DMPC, DPPC), SEC translocon) were systematically screened. As expression and folding monitor, a conventional split GFP system was implemented. Based on the obtained results, algorithms were developed and proposed to predict future membrane protein synthesis in CFPS.

The study provides some interesting aspects for future cell-free membrane protein synthesis. However, to my opinion some messages of the study are not justified and a more thorough analysis of the synthesized targets is necessary in order to judge the value of the presented process.

Some major critics:

1. Generally, the authors mix synthesis, membrane association/insertion and functional folding of the membrane proteins. These are three different steps of membrane protein expression and must be analyzed in different ways. Synthesis is just the amount of translated protein, regardless whether folded, unfolded, precipitated or solubilized by lipids. Synthesis, i.e. the total amount of synthesized membrane protein was not analyzed in the study, as potentially non-fluorescent precipitated or lipid solubilized protein could not be detected.

2. The authors monitor lipid-solubilized membrane proteins that are fluorescent by the split-GFP complementation strategy. The data might then represent fractions of the totally lipid-solubilized membrane protein of a given reaction. These are certainly important data as one can assume that any non-fluorescent target is unlikely to be functionally folded. However, I disagree with the assumption of the authors, that GFP complementation is a proof of functional folding of the attached membrane protein. It has been shown in previous publications that the fluorescence of C-terminally attached GFP is not a general measure for the functional folding of cell-free synthesized and lipid-solubilized membrane proteins. It may correlate for some targets, but in particular for GPCRs this is mostly not the case. This is in contrast to the conventional FSEC approach upon *in vivo* expression, where non-folded proteins will be removed from the membrane by degradation. If the authors want to judge the quality of the synthesized membrane proteins, functional assays of at least some selected targets is necessary. Thus, a major weakness of the study is that no synthesized target has been analyzed for functionality and thus nothing can be said about their quality.

3. The rationale behind the selected optimization compounds is unclear. The five compounds affect different steps of cell-free membrane protein expression: Synthesis efficiency (K⁺, Mg⁺, PEG) and membrane association/insertion (lipids, SEC). In the *E. coli* cell-free system, the complexity of protein synthesis is largely reduced to the basic translation step. Here, the initiation of translation is most critical and several studies showed that corresponding DNA template optimization of the first codons is essential to obtain significant synthesis of many membrane proteins (e.g. Cho et al., 2023). A properly designed DNA template is thus the prime prerequisite for efficient cell-free membrane protein production and this was not considered in the study. The applied random codon optimization does not address the problem of ineffective initiation of translation. A further point is the absence of any redox system. The proper formation of disulfide bridges is essential for the folding of eukaryotic membrane proteins and it is thus unlikely that these targets have been synthesized in good quality. As the authors mentioned, the complexity of the system is rapidly increasing with the number of screening compounds and thus it is

doubtful whether all expression steps, synthesis, solubilization and folding, can be covered with one screening program.

4. The authors used 28 membrane protein targets for their study and claim that 17 were successfully synthesized. The results are in agreement with numerous previous cell-free expression screens of membrane proteins covering even higher number of targets (e.g. Bruni 2022, Isaksson 2012, Schwarz 2008, Savage 2007). Having already comprehensive data of cell-free synthesized membrane proteins available, it remains questionable why only few of them have been used for the development of predictive algorithms.

Some specific comments:

Figs 1A and S2: This is too simplified. The reaction is much more complex as many proteins will be lipid associated or only partially membrane inserted.

Figs. S4 and S5: Nanoparticle tracking of liposomes before or after CFPS reaction? Fusion of the liposomes during reaction is quite likely.

Fig. S11: The histogram is labeled as change in yield, but what are the absolute yields? What is the default yield/reaction condition the change is calculated from?

Fig. S12: The figure does not show synthesis as fluorescence does not cover synthesized but precipitated or any other unfolded but non-fluorescent protein. What this figure might rather show is dependency of membrane association/integration on protein features. This is reasonable and a valuable information as it is prerequisite for functional folding. However, it still does not show that the membrane proteins are functionally folded.

Fig. S15A: Why such high variations of the Cy5 values of liposomes from different reactions? Wasn't the liposome concentration in each reaction identical? N=3 is stated, were these independent experiments, meaning with different batches of Cy3-labeled lipids prepared liposomes? It rather appears that there is a high batch to batch variation of Cy3 fluorescence in liposome preparations.

Reviewer #2

(Remarks to the Author)

In this interesting work, the authors developed MEMPLEX (Membrane Protein Learning and Expression) which uses a machine learning (ML) approach, nanoliter droplet printing, and a membrane protein reporter in order to design artificial synthesis environments for membrane proteins. Using their approach through MEMPLEX, the authors also found that the hydrophobicity of the membrane-contacting amino acids can help predict membrane protein synthesis in artificial environments. This work is novel and timely in that it combines ML with cell-free synthesis of membrane proteins that are difficult to produce. The manuscript is also well-written, and I have only several comments to strengthen this exciting and significant work.

1. It would be practically more useful if the authors select a few membrane proteins and their synthesis conditions, and then demonstrate their production at a larger scale. I see the technological significance of this work, but such a demonstration will show a bigger impact of this work on the biotechnology field.
2. Membrane protein biosynthesis is very challenging when using cellular systems as the authors mentioned. Even if the attempts fail, it would be great to compare cell-free production-based yields (from Comment 1-based experiments) with those of cellular protein production. It would be fine for the authors to try a few times and say cellular production of the selected membrane proteins has failed. Such negative data would also make their work more significant.
3. It would be convenient to see that supp fig captions appear immediately after each figure. It was hard for me to review each supp fig.
4. Although cell-free systems give us a lot of promise, there are still many limitations and challenges to overcome, compared to cellular production. I highly recommend that the authors should discuss the limitations of and challenges in cell-free systems, compared to cellular production. As is, I feel the manuscript could be seen as a bit biased article.
5. Nanoliter droplet printing is pretty exciting, and readers would appreciate it if the authors could add more explanations to the supplemental information. The manuscript and supplemental information have some texts on it, but a bit more elaboration would be useful.

Reviewer #3

(Remarks to the Author)

Summary:

In this manuscript, the authors develop MEMPLEX, a high throughput workflow combining cell-free protein synthesis, robotics, and machine learning, to predict optimal cell-free reaction environments for expressing membrane proteins. In an initial set of 8,296 unique cell-free reactions, the authors screen for the expression of 28 different membrane proteins in conditions that modulated five different parameters: potassium concentration, magnesium concentration, PEG8000 concentration, SecYE coexpression, and liposome lipid composition. Importantly, they note that there is no "standard" set of reaction conditions that uniformly results in high soluble protein yields for all membrane proteins and that there is significant interdependence between reaction variables for each membrane protein. Using this initial dataset, they then use an ensemble-based active learning approach to predict expression conditions with increased soluble expression, increasing yields of 21 of 28 proteins. Interestingly, the authors find that they can not predict optimal expression for new proteins based

on sequence or other characteristics, but can predict optimal conditions by evaluating the response to yield under a subset of expression conditions and then comparing that response in the embedded system of the initial dataset/model. Finally, the authors develop a quantitative metric describing how each amino acid residue interacts with its environment that can then be used to predict whether an expression condition exists that will enable successful production.

While the authors present a large body of work, including exploration of thousands of different expression conditions, the lack of emphasis on quantitative yields in g/L (i.e., using relative yields) lowered my enthusiasm. It is also unclear how much this really moves the needle on membrane protein expression. At times, the paper was difficult to follow and some of the data were not presented in a manner that ultimately supported their major conclusions.

Major Questions/Comments:

- Figure 2D: For calculating the values for Figure 2D, the authors use the top three expressing conditions and calculate the average value for each variable to display in the heat map. If the authors state that each of the variables are interdependent, does displaying average values for individual variables add value if they can not be interpreted in isolation? Furthermore, given the low, medium, and high values for potassium, magnesium, and PEG8000 concentration are evenly spaced, this would appear to be misleading as one protein that had a top expressing condition with each concentration of potassium would appear the same as a protein that the top condition was exclusively at the medium concentration. It appears that considering the values in the heat map as "optimal reaction compositions" is misleading as in most cases, the actual values shown were never shown or validated. Additionally, the categorical lipid conditions (DMPC, DPPC, and DOPC) are presented as low/medium/high variables alongside the other concentration-dependent parameters. It is unclear how these lipids were assigned values or averaged.
- In the discussion, the authors indicate that their platform could be improved upon by incorporating additional variables to test (and spend a significant amount of text listing additional parameters that could be considered). However, they then state that in their current work incorporating 3 additional parameters into the workflow, they were unable to substantially improve protein yield and caution against "the 'curse' of expanded search space." In their final paragraph, the authors again state that their platform could be improved by incorporating a broader parameter search space. These statements are contradictory and call into question the effectiveness of their machine learning model.
- It is unclear how the greater scientific community would be able to apply this workflow to their own work given the complicated nature of the workflow (both requiring advanced robotic systems as well as a large number of models with somewhat limited accuracy). How would a user know what initial reaction conditions to test in order to accurately place a new membrane protein into the embedded space? Do the authors anticipate issues with cross-lab challenges?
- The authors devote the end of the results section to a detailed discussion of an AlphaFold-informed predictive model for the larger unexamined membrane proteome. They post-facto include their tested proteins, which appear to match this model. However, no validation of these results is presented using previously-untested proteins, which makes further claims about predicted protein synthesizability appear unsupported. Did the authors consider validation of this model using MEMPLEX or other methods? This should be done.

Minor Questions/Comments:

- Abstract, Sentence 4, as well as Results, Paragraph 1, Sentence 1, the authors characterize their choice of the split folding GFP as a "novel" reporter. However, this reporter protein has been used in a number of different applications. The authors should soften their language regarding the use of this reporter to emphasize the other novel parts of their work.
- Introduction, Paragraph 3, Sentence 5, the authors state that included in the parameters that they modulate are plasmid sequence and liposome concentration. However, the rest of the manuscript/data do not mention altering either of these parameters. The authors should clarify whether these parameters were modulated and to what extent/effect.
- Figure 1A: The addition of chaperones to the reaction mixtures is depicted as proteins as opposed to plasmid encoding the proteins for co-translational production. The authors should alter the schematic to make this clear. Furthermore, the authors included the proteins YidC and SecB, which are not mentioned at all in the main text of the manuscript. Were these proteins included in the study?
- Results Section 1, Paragraph 4, Sentence 5 refers to Figure 3D. Should this instead read Figure 1D?
- Results Section 2, Paragraph 1 refers to the inclusion of the translocating channel SecYE. For additional clarity, the authors should consider editing the sentence to indicate the variable is whether to include plasmid for co-expression of SecYE (as opposed to purified SecYE protein). Additionally, do the authors know that SecYE is solubly expressed in all conditions tested? If so, the authors should state this and also provide supporting evidence in the supplementary information.
- Figure 2A: The authors should consider indicating that the values for SecYE are indicating plasmid concentration, not addition of protein.
- Figure 2A caption: The authors highlight certain conditions as "standard" reaction conditions. What is the rationale behind considering these conditions "standard"?
- Figure 2B: For additional clarity, the authors should include in the figure captions what the magenta circles represent.
- Figure 2E: The authors should add information in the figure captions indicating that the red and blue outlined boxes are pointing out the specific variables displayed in panels 2F and 2G.
- Figure 2G and F. The authors should consider displaying error bars more clearly, since some error bars are almost fully covered by others.
- Figure 3C: Why color code the "Made Previously" label and only provide a legend in the caption?
- Results Section 2, Paragraph 1: The authors state that they investigated 5 reaction parameters, but only explicitly name 4. Based on the figures, the two salts were considered to be separate parameters, but this is not clearly indicated within this section since the specific salts are named in parentheses.
- Results Section 2, Paragraph 2: The authors state that the third criteria used to select membrane proteins was that the proteins were "expected to give a variety of different or zero yields". Why did the authors believe a priori that some

membrane proteins would not express in any of the tested conditions? Additionally, the criteria for selection are stated awkwardly, making it difficult to determine the authors' rationale for choosing these proteins.

- Results Section 2, Paragraph 3: The phrasing of the sentence, "For 39.3% of the proteins, the top 3 reactions used 2% PEG while 25% of them used 0%" it is difficult to parse the meaning. Presumably this refers to 25% of the top reactions and 0% PEG.

- Results Section 3, Paragraph 3: The authors state that for each of the 28 membrane proteins, they selected 12 reaction conditions to test. However, they also indicate that they performed >11,000 unique reactions. These values appear to be one order of magnitude inconsistent with one another - how was the >11,000 calculated?

Reviewer #4

(Remarks to the Author)

Meyer et al. report carrying out a high-throughput screen of in vitro translation conditions to improve expression yields of 28 *E. coli*, *A. thaliana*, or human transmembrane proteins using *E. coli* whole-cell extract, liposomes, and the SecYE translocon. The authors screen three concentrations of three buffer components (85, 135, or 185 mM K⁺; 8, 14, or 20 mM Mg²⁺; and 0, 1, or 2% PEG), three types of lipids for the liposomes (DOPC, DMPC, or DPPC), and run reactions with or without added plasmids for co-expression of the translocon. (Presumably, in the absence of added translocon plasmids, translocons come from the whole-cell lysate.) Robotic pipetting (aka "droplet printing") helps the authors assemble 2- μ L reactions in 384-well plates. Split GFP is used as a reporter of protein aggregation, with GFP-dark reactions presumed to have failed the expression test due to aggregation of the expressed protein and GFP fluorescence used as a proxy for the concentration of properly folded proteins in the rest. Finally, the authors carry out statistical analyses, embedding, and neural network training in an effort to predict optimal expression conditions for transmembrane proteins. Using a relatively generous threshold of >1 pmol protein produced with $p < 0.05$ for protein expression, the initial screening campaign of ~8,000 reactions (including replicates), the authors report expression of all 28 proteins, including 7 that had eluded prior efforts at in vitro expression in liposomes. However, in some cases "expression" may be mere noise of the GFP assay, and this needs to be clarified. This initial dataset is used to predict more optimized expression conditions via machine learning, which motivates a second screening campaign of ~11,000 reactions, as a result of which the authors report improved yields for 21 of the 28 proteins compared to the initial, randomly chosen reactions. An expansion of the search space and a further 17,000-reaction screening campaign does not lead to further improvement. Low-throughput validation of expression of a small subset of proteins is carried out.

To rationalize their results, the authors turn to basic and statistical principles of protein-lipid interaction: specifically, hydrophobicity of lipid-facing side chains and the statistical likelihood of a given type of residue on the lipid-facing surfaces of a training set of transmembrane protein structures in the PDB. Only alpha-helical proteins are examined; the authors do not consider beta-barrel proteins. These properties of the protein surface are found to be useful predictors of its expression yield in the liposomes.

Overall, this is an ambitious and integrated computational-experimental method development study addressing a major challenge in biochemistry of transmembrane proteins. The data are likely to be of interest to researchers in this field, assuming data analysis is valid. The quality of writing is good, particularly in presenting examples of computational and experimental conditions, and the figures are mostly clear and informative. The Discussion section addresses several important points, notably the omission of cholesterol or any other sterols from the lipid membranes tested in this study. There are several other weaknesses in the study as currently reported, which should be addressed. The methodology is innovative and ambitious, but some aspects of experimental methodology and statistical analysis needs clarification, and there seems to be a tendency to "overhype" results.

First, I find myself confused about the statistical significance of results in this study. If I understand correctly, the " Δ Yield" metric refers to the difference in split-GFP intensity in the presence vs. absence of liposomes. As such, a Δ Yield of 0 should be interpreted as no detectable expression of the protein in liposomes. The average Δ Yield exceeds 1 pmol for only 3 of the 28 proteins in Figure 2C or Supplementary Figure 10 during the initial screening campaign of ~8,000 reactions. Following the authors' machine learning campaign, this number increases to 7 of the 28. Apparently, the authors trained a neural network model on 8,000 reactions across 28 proteins despite the training data for 25 of the 28 proteins being mostly noise? And for the few conditions that did yield statistically significant expression, is $p < 0.05$ truly an adequate metric? Did the authors include any correction for multiple hypothesis testing? How many replicates were used for each condition, and how many were discarded as outliers due to being >2 SDs away from the mean of other replicates? As with any high-throughput screen, a detailed discussion of the statistical analysis should be a very important part of the Methods.

Second, the paper does not straightforwardly evaluate the screening campaigns' success. The authors' dataset includes seven proteins for which previous efforts to reconstitute them in liposomes were unsuccessful. How many of these seven previously-failed proteins were successfully expressed in the present study, when multiple hypothesis testing is taken into account? Has their successful expression been validated using low-throughput methods? If not, doing the validation would dramatically improve the biological usefulness of the present study. A related point is that the success of the computational prediction does not seem to be rigorously evaluated. It certainly seems to have improved expression yields for most proteins, but what would be the expected outcome if the authors simply screened 11,000 conditions chosen randomly from the same search space as the 11,000 they did screen in the second screening campaign?

Third, what is the topology of protein insertion into the membrane for the 28 proteins chosen in this study? Since liposomes are generated prior to protein insertion, and the split-GFP tag is always on the C-terminus of each protein, this means the C-terminus of each protein must be on the outside of the liposome (equivalent to the cytoplasmic side of the cell membrane) in

order for the split-GFP assay to detect it. Do all 28 proteins have cytoplasmic C-termini? If not, then the assay used in this study would not be expected to work for them. Or have the authors previously observed that proteins in this system incorporate into the liposomes in random orientation?

Fourth, the claim that the authors “discover a new metric, based on hydrophobicity of the membrane-contacting amino acids, that predicts membrane protein synthesis” seems like a stretch. It is rather obvious that proteins with a hydrophobic transmembrane surface are more stable within lipid bilayer membranes than proteins with a hydrophilic surface. The bigger biological puzzle here is why do some natural transmembrane proteins have more hydrophilic surfaces than others.

Fifth, the authors mention chaperone proteins as one of the important conditions, and they are among the four reaction components depicted in Figure 1. However, I could not find any results with chaperones in the paper. Did the authors add chaperones or vary their concentrations in any experimental conditions? If not, then please remove them from Figure 1 to avoid confusion.

Sixth, I am confused about the batch-to-batch benchmarking of whole-cell extracts in this study. The authors state in M2 that each extract was “benchmarked against the previously used extract across several different magnesium (8-16mM) and potassium (85-185mM). The coefficient of variance (standard deviation / mean) between the replicates of each extract at each unique salt concentration were calculated to assess their similarity.” How was this benchmarking done? Was it by expressing a subset of proteins and monitoring split-GFP fluorescence? What are the replicates the authors reference in this case? Do the authors mean variance in split-GFP fluorescence among identically prepared reaction mixes using extract N and extract N+1?

Seventh, the authors mention in M17 that, in the interest of increasing the number of data points available for training, replicate samples were treated as independent samples, but “All replicates for a given reaction condition were kept within the same split.” Could you please clarify this statement?

Eighth, M20 states that “A concave hull around the membrane spanning region was calculated for each protein.” How was it calculated?

Version 1:

Reviewer comments:

Reviewer #1

(Remarks to the Author)

Two major concerns are still not fully addressed.

1: The first challenging step in cell-free protein synthesis is screening of proper translation by modification of the initiation of translation. As shown in previous publications, this can bring membrane protein production from almost nothing to mg amounts per mL. If the basic production is already suboptimal, which can thus be assumed for many of the targets, the value of any further solubilization optimization by screening of compound concentrations is limited. Furthermore, membrane protein production and solubilization/membrane insertion must not be correlated. The cotranslational membrane protein solubilization depends on the presence of a suitable hydrophobic environment in sufficient concentration. The screening strategy included a mix of compounds affecting yield (Mg^{2+} , K^+ , PEG) or membrane insertion (lipid types). However, as only membrane insertion/association was monitored as output, the effect of the yield affecting compounds is not clearly evident. The presented pipeline is thus to my opinion preliminary and would require significant modifications such as an initial production screening (i.e. the optimization of the pure protein yield e.g. by template design and others), for rather professional applications. The current protocol omits this first step and instantly starts with quality screening (i.e. membrane insertion and folding). Of course, this may still give valuable hints for the lipid-based solubilization of individual membrane proteins and their preferences for some of the tested compounds, but any messages of yields or efficiencies are relative to the implemented suboptimal conditions and thus less meaningful.

2. I completely agree that GFP fluorescence would be great for htp screening. But unfortunately it does not correlate with function. The authors speculate that split GFP may be a better folding monitor if compared with terminally attached full-length GFP. I rather think that it may be worse as the large GFP domain is added in already folded condition and the presentation of a small terminal domain attached to an unfolded or partly unfolded membrane protein appears to be even more likely. To show the correlation of split GFP fluorescence in a number of conditions with the activity of at least one target as proof of principle would thus still significantly improve the manuscript and support the value of the presented strategy. Suitable targets of the list are e.g. the adrenergic receptors, where well established and relatively easy to perform binding assays are available. The shown antibody approach is of limited value as it only shows the recognition of relatively short domains and not the function of a protein. The argument that in vivo the antibodies were shown to recognize functionally folded protein does not work for the cell-free system. As with FSEC, where in vivo the folding of GFP can be correlated with folded membrane protein, this cannot be assumed for the cell-free system as any proof-reading systems removing non-functional proteins from the membrane are missing.

The mentioned liposome control does not consider improperly inserted, unfolded and partially unfolded membrane proteins which are then in the soluble fraction in presence of liposomes. Again, the pure solubilization of a cell-free synthesized membrane protein by supplied lipids does not indicate its functional folding. The authors also ignore previous publications showing that frequently only fractions of cell-free produced and membrane solubilized membrane proteins are functionally

folded.

The authors mention that in previous htp reports physical methods such as solubilization, labelling, SEC profiling, thermophoresis or CD spectroscopy have been used to speculate about membrane protein functionality and they now combined a higher number of these techniques in their manuscript. I agree with the authors that these techniques are not proof of functionality, but also a more comprehensive combination of these techniques cannot replace a quantitative functional assay. In addition, the presented production of bacteriorhodopsin shows that their cell-free system is able to produce functional protein, which was not doubted. But it has nothing to do with the split GFP approach and thus does not support it.

(Remarks on code availability)

Reviewer #2

(Remarks to the Author)

The authors have addressed my comments to my satisfaction.

(Remarks on code availability)

Reviewer #3

(Remarks to the Author)

The authors have tried to respond to the concerns as directly as possible, including my original comments. I think this has been reasonable. Given the limited number of activity assays (which are described in response to Reviewer 1), it is still a bit unclear how much this really moves the needle on membrane protein expression, which of course requires activity. The work does catalog many conditions for membrane protein synthesis that the abstract now better highlights.

(Remarks on code availability)

Reviewer #4

(Remarks to the Author)

The authors have addressed some, but not all, of my concerns. I appreciate the bolstered analysis of the effects of active learning vs. random reaction choice. However, I continue to have concerns about how the authors assess the success or failure of their method in improving or rescuing protein expression.

Comment 1

My best understanding of the authors' method as presented is that, for any given protein, it takes only one expression condition with improved yield at $p < 0.05$ for the authors to declare that they have successfully improved liposome-bound expression of that protein. Since the authors screen hundreds of conditions per protein, they are, in fact, testing hundreds of hypotheses and declaring success if any one of those hundreds of hypotheses appears true with $p < 0.05$. Unless I am really missing something, there must be a multiple-hypotheses correction when drawing conclusions like this. Otherwise, the authors could simply set up 20 identical conditions in quadruplicate and expect that one of those conditions might by chance appear to have improved expression and therefore declare "success." If my understanding is correct, then please carry out the multiple hypotheses correction. If my understanding is not correct, and the authors use some other definition of success, then please make that definition explicit in the paper.

Comment 2

The authors have not addressed my question. To clarify what I mean, Figure 3C lists six of the proteins whose expression previously failed in other studies: Glut, Neu, Vol, Aux, InP, and Dia. In their revised text, the authors state that all six were successfully expressed by MEMPLEX. Have the authors validated successful expression of any of those six proteins by low-throughput methods? In other words, did MEMPLEX actually find conditions that successfully express transmembrane proteins that other researchers had previously failed to express? That should be a straightforward and very important assessment of the new method's real-world success rate, yet I can't find that assessment in the paper.

(Remarks on code availability)

Version 2:

Reviewer comments:

Reviewer #1

(Remarks to the Author)

To my opinion, the responses to my concerns is still not adequate as specified below. Nevertheless, the authors invested a remarkable workload and their presented memplex approach goes into the right direction. However, interpretation of the current results still appears sometimes to be overestimated and I would appreciate if rather preliminary and misleading statements regarding the below mentioned topics would be clearly modified as exemplified below. I would find it more valuable to present the pipeline not as a high-end tool but rather as a first case study with an initial basic platform that may become individually adjusted in future to the requirements of certain membrane protein types or families.

The high impact of proper initiation of translation was shown in several previous reports and is well recognized. The shown calculation of translation initiation strength of their targets is hard to evaluate as method data are missing. I assume that just the promoter and nearby regions were considered. The algorithms of the cited references are to my opinion not suitable, as they do not consider sequences and potential secondary structure formations of the individual codon regions. Secondary structure formation of the initiation area with further downstream sequences can protect proper interaction with the ribosome and thus prevent translation. Programs are available to predict potential secondary structure formations but they have to be applied to each individual codon region and usually result into too many possibilities. Thus, empirical approaches as e.g. published in Cho et al 2023 are more straightforward and highly successful. It would have been interesting to perform that with some of the low expressing targets.

For the usefulness of the split-GFP assay, the manuscript is lacking a clear proof of concept. In order to provide this, the authors now extended their previous analysis on the CF synthesized bR. In fact, bR and the related proteorhodopsin are frequently taken as "proof of principle" membrane protein targets in cell-free expression studies. The reason is that they are very well translated and they rapidly integrate and fold in a large diversity of membranes and other hydrophobic environments. In short, they are "easy" membrane proteins present in almost 100% folded conformation. Consequently, if GFP or a split GFP moiety is attached, their fluorescence linearly corresponds to the synthesis of active bR, as it was shown in the new Fig. 2/supp Fig. 6. bR is thus not suitable to show that split-GFP only complements if attached to folded protein as there is simply no inactive bR present. The Supp Fig. 6 just shows that activity correlates with increasing protein yield, which is certainly expected.

This exceptional perfect situation with bR is unfortunately different with most other, less easy, membrane proteins, such as most GPCRs, transporters or channels. In these cases, similar to bR most of the CF synthesized protein associates with the membrane. However, as remarkable difference, only a fraction folds into functional conformations. A key problem of CF synthesis is that these unfolded fractions remain in the sample and are not removed by proofreading systems like in living cells. That is why FSEC does not work with CF systems. As an example, only 50% of totally membrane solubilized GPCR may be folded. The still unanswered question is now: Does the fluorescence of the complemented split-GFP correlates with the determined functional fraction of membrane protein (50%), or is it significantly higher and rather correlates with the total amount of synthesized membrane protein (100%). With full-length GFP fusions, it was shown that fluorescence clearly correlated with the total amount of synthesized protein independent of folded conformations. With split-GFP, it still would be very interesting to know it.

The message "we could not find successful conditions for beta2 and beta3" is not clear to me. As the proteins are synthesized, do they mean conditions for ligand binding? In that case it would be a strong hint that split-GFP fluorescence does not correlate with protein folding. A further evidence in that direction is the mentioned measured fluorescence of complemented split GFP after bR expression without liposomes. Can it be assumed that bR also folds without any hydrophobic environment?

To subtract the GFP fluorescence obtained after expression without liposomes is not a proper control for the expression with liposomes. These are two completely different expression modes and the fact that membrane proteins also incompletely insert into membranes or only associate with them is again ignored. In that context, supp Fig. 2 is too simplified and misleading. Unfolded proteins have random structures and consequently the exposure of the GFP11 moiety is random, meaning sometimes exposed and sometimes not. The random fluorescence of a protein precipitate has therefore nothing to do with the fluorescence of a mixture of membrane attached unfolded protein and membrane inserted folded protein.

L. 443: The message is misleading. Structural approaches need pure protein transferred into micelles, polymers or nanoparticles. Considering the expected high losses during purification and processing, a basic expression yield of max 100 µg/mL is far too low.

L. 46: Ref 10 is not about CF membrane protein synthesis

L. 86: not folding but fluorescent protein "solubilization" monitor

L. 123/124: ..GFP-11 is inaccessible when the attached protein misfolded. This interpretation is not correct and misleading. The authors use the cited references as proof for their statement that split-GFP complementation only happens with functional proteins. In the cited references, fluorescence of complemented split GFP was used to distinguish precipitate from soluble protein. Precipitated protein is certainly misfolded, However, it was not shown that the soluble proteins in addition contain a misfolded fraction that may give fluorescence. Solubilized CF generated membrane proteins frequently contain a mixture of folded and misfolded protein. Thus, solubilization is not identical to functional folding.

L 127/128: Based on the current data, this statement is pure speculation.

L 152: general membrane assoziation

(Remarks on code availability)

Reviewer #4

(Remarks to the Author)

The authors have addressed my remaining concerns reasonably. Overall, the study is technology-focused, with relatively limited demonstration of practical biological utility; however, as a demonstration of the technology platform, it is sufficient. I hope the authors will build on this platform in future work.

(Remarks on code availability)

Version 3:

Reviewer comments:

Reviewer #1

(Remarks to the Author)

My queries were addressed and I have no further comments or remarks. The presented approach is useful to obtain an initial overview on the expressability of membrane protein targets and the manuscript gives now further suggestions to subsequently evaluate and to improve product quality.

(Remarks on code availability)

Cheemeng Tan
Associate Professor, Department of Biomedical Engineering
One Shields Ave, Davis, CA 95616

Phone: 530-752-7849
Fax: 530-754-5739
email: cmtan@ucdavis.edu

July 16th, 2024

Dear Reviewers,

Please find attached the revised manuscript, “Designer artificial environments for membrane protein synthesis ” for publication in Nature Communications.

We greatly appreciate the feedback with regards to our manuscript. We have addressed all reviewer comments, and all changes are highlighted in yellow. Below, we have included a detailed point-by-point response to comments by the three reviewers.

I hope that you will find our response adequate. Please let us know if you have any concerns regarding the changes that we made.

Reviewer comments

Reviewer #1:

Comments for the Author:

The authors describe a cell-free throughput screening process for the evaluation of membrane protein synthesis. The authors used a standard E. coli based CFPS batch system in 2 μ L reactions for throughput analysis. As variable expression optimization parameters, the concentrations of five reaction compounds (K⁺, Mg⁺, PEG, lipids (DOPC, DMPC, DPPC), SEC translocon) were systematically screened. As expression and folding monitor, a conventional split GFP system was implemented. Based on the obtained results, algorithms were developed and proposed to predict future membrane protein synthesis in CFPS.

The study provides some interesting aspects for future cell-free membrane protein synthesis. However, to my opinion some messages of the study are not justified and a more thorough analysis of the synthesized targets is necessary in order to judge the value of the presented process.

We thank the reviewer for their time, and we appreciate the comments regarding our work.

Major comments:

(1) Generally, the authors mix synthesis, membrane association/insertion and functional folding of the membrane proteins. These are three different steps of membrane protein expression and must be analyzed in different ways. Synthesis is just the amount of translated protein, regardless whether folded, unfolded, precipitated or solubilized by lipids. Synthesis, i.e. the total amount of synthesized membrane protein was not analyzed in the study, as potentially non-fluorescent precipitated or lipid solubilized protein could not be detected.

We appreciate the reviewer’s observation. We agree that while the synthesis of a protein and its incorporation into the membrane are likely correlated, they represent different things. In this work, we specifically tailored our reporting system to monitor the amount of protein incorporated into the

membrane, so we have modified the text to remove any confusion. Our manuscript focuses on the first challenging step of membrane-protein synthesis – to get them into a synthetic lipid bilayer. Many computational frameworks for membrane protein folding show the insertion step as the primary entry point for subsequent folding and, therefore, the first critical step for folding^{1,2}. This common assumption in the field motivated our selection of insertion as the metric for success. As shown by our study and many others³, this first step fails for many membrane proteins under non-optimal conditions.

(2) The authors monitor lipid-solubilized membrane proteins that are fluorescent by the split-GFP complementation strategy. The data might then represent fractions of the totally lipid-solubilized membrane protein of a given reaction. These are certainly important data as one can assume that any non-fluorescent target is unlikely to be functionally folded. However, I disagree with the assumption of the authors, that GFP complementation is a proof of functional folding of the attached membrane protein. It has been shown in previous publications that the fluorescence of C-terminally attached GFP is not a general measure for the functional folding of cell-free synthesized and lipid-solubilized membrane proteins. It may correlate for some targets, but in particular for GPCRs this is mostly not the case. This is in contrast to the conventional FSEC approach upon *in vivo* expression, where non-folded proteins will be removed from the membrane by degradation. If the authors want to judge the quality of the synthesized membrane proteins, functional assays of at least some selected targets is necessary. Thus, a major weakness of the study is that no synthesized target has been analyzed for functionality and thus nothing can be said about their quality.

Thank you for this excellent observation. Given the high throughput nature of the MEMPLEX platform, we prioritized methods that enabled us to screen successful protein synthesis and membrane integration in a rapid and scalable manner, while gathering as much information from the reactions as possible. The state-of-the-art in the field that studied multiple membrane proteins uses Western blots^{3,4,5}. These studies compared the pre and post-centrifugation samples to assess successful synthesis and required sample volumes 10-100x the size of our reactions. Each assumed that if the protein did not aggregate, or became large enough to pellet during centrifugation, the protein had been successfully produced. Other groups operate with the same assumption but quantify the amount of protein in each fraction using TCA precipitation and liquid scintillation counting^{6,7}.

We agree with the reviewer's comment that "It has been shown in previous publications that the fluorescence of C-terminally attached GFP is not a general measure for the functional folding of cell-free synthesized and lipid-solubilized membrane proteins." For instance, Jacobs et al. 2019⁸ used C-terminally attached GFP to assess the insertion of MscL, and their data showed high background fluorescence when no liposomes were added to the reaction. This result agrees with our earlier data (not included) when we used full-length GFP attached to the target protein. We suspected that GFP was too stable to be pulled apart by aggregation of the target protein. Therefore, for this manuscript, we did not use the C-terminal attached GFP as the detection method. Instead, we used the split GFP complementation assay. The single beta strand attached to the protein is more likely to aggregate with the tagged protein. This change reduced our background signal and gave us higher confidence in the observed signal.

An important additional consideration of our work was how the signal changes based on the presence or absence of liposomes. All the earlier referenced papers only looked at samples with the lipidic or lipid-mimetic environment included. Our data showed that while many proteins have very low GFP signal when no liposomes are present, some showed fluorescence above the background.

A high GFP signal would correlate with the protein appearing in the soluble fraction of a western blot. However, this signal cannot indicate that the protein is properly integrated. That is why every reaction that was tested had the corresponding no liposome condition. This way, we only see the increase in signal because of the presence of liposomes.

However, these other groups did strive to show additional evidence beyond Western blot data. Bruni et al.³ used microscale thermophoresis to show that their synthesized mitochondrial pyruvate carrier complex interacts with pyruvate but did not demonstrate that the protein can transport pyruvate, which would show full functional folding. Isaksson et al.⁴ used circular dichroism spectroscopy to evaluate the secondary structure of two proteins and showed that they are mostly α -helical but did not prove that they are functionally folded either. They did show that the proteorhodopsin shifted the spectra of retinal, indicating that the binding pocket of the protein was correctly folded, but that is a common assay for a stable set of proteins. Savage et al.⁵ used gel filtration to further investigate a subset of the proteins they produced. They assumed that if the protein sample eluted as a single tight peak, it was homogeneous and therefore “well behaved and similar to their in vivo counterparts”.

We used size exclusion chromatography and flow cytometry on a subset of our samples to verify that the proteins were associated with the membrane. This is an added verification missing in these other papers. With these data in the original manuscript, the number of membrane proteins, the improvement in detection method, the controls, and the multiple verification experiments already greatly exceed the state-of-the-art in the field that studied the synthesis or membrane-insertion of multiple membrane proteins in a single study.

However, we concede that a protein with an exposed tag and in association with liposome is not direct evidence that the protein is functional. To demonstrate the power of our platform, most of our target membrane proteins have not been synthesized in the literature. Due to this choice, it is out of scope to create biochemical functional tests for each of these new proteins while synthesizing them in the same work. In the original manuscript, we conducted a Western Blot that used a CD47-specific antibody to show that the correct sequence for CD47 was produced, as shown in Supplementary Figure 18.

In addition, in benchmarking experiments prior to the high-throughput assays, we have demonstrated the successful production of bacteriorhodopsin (bR), a seven-transmembrane domain protein that has been successfully produced in cell-free by other groups^{9,10}. bR binds the chromophore all-*trans* retinal and incorporates it into the core of the protein. This causes a red shift in the absorption spectra. All-*trans* retinal is yellow when unbound but shifts to purple when successfully incorporated into bR¹¹. In the image below, the Liposome+Retinal samples include the full cell-free reaction with the retinal added but exclude the DNA encoding bR. The Liposome+Retinal+bR sample includes the same components plus the bR plasmid. There is a clear shift from the characteristic yellow color of the retinal in the liposome control samples to a purple-pink color with the expression of bR. This color shift is indicative of retinal binding, which is only possible when the protein is correctly folded¹¹. This result was not included in the original and modified manuscript as it has a low throughput and has been published routinely by others. But, it corroborates that our system can produce a classical and functional membrane protein as others.

To further address the reviewer's comment, we identified some proteins in the set, namely CD9 and CD81, that have available antibodies binding to the native form of the protein on cell surfaces or vesicle membranes. In new experiments, we have conducted immunoassays to demonstrate that the synthesized proteins have exposed functional domains. These assays primarily employed flow cytometry (Cytek Aurora, spectral cytometer), adaptable to various protein-specific antibodies and able to detect particles at 60-150nm diameter range – the size range of our liposomes. Overall, we demonstrate that >40% of the observed CD9 liposomes and >15% of the observed CD81 liposomes bound their respective antibodies. (Supplementary Figure 18B&C) The differences observed in antibody-specific detection are expected due to differences in protein yield of the two proteins, as shown by the split-GFP reporter (see Figure 3C).

This finding is also corroborated by Gunasekaran et al. 2020¹², Zhang X et al. 2020¹³, and Hasselman et al. 2021¹⁴, who utilized these antibodies to identify native proteins on cell or vesicle surfaces. For instance, Hasselman et al. 2021¹⁴ used the CD9-antibody and showed the association of upregulated CD9 on disease-associated microglia with amyloid plaques. The corresponding manufacturer notes are added here for the reviewer's reference:

Anti-CD9: <https://d1spbj2x7qk4bg.cloudfront.net/en-ie/products/fitc-anti-human-cd9-antibody-2212?pdf=true&displayInline=true&leftRightMargin=15&topBottomMargin=15&filename=FITC%20anti-human%20CD9%20Antibody.pdf&v=20240412063148>

Anti-CD81: [https://d1spbj2x7qk4bg.cloudfront.net/en-ie/products/fitc-anti-human-cd81-tapa-1-antibody-6766?pdf=true&displayInline=true&leftRightMargin=15&topBottomMargin=15&filename=FITC%20anti-human%20CD81%20\(TAPA-1\)%20Antibody.pdf&v=20240411093413](https://d1spbj2x7qk4bg.cloudfront.net/en-ie/products/fitc-anti-human-cd81-tapa-1-antibody-6766?pdf=true&displayInline=true&leftRightMargin=15&topBottomMargin=15&filename=FITC%20anti-human%20CD81%20(TAPA-1)%20Antibody.pdf&v=20240411093413)

With the results, we underscore our platform's capacity to synthesize membrane proteins with intact functional domains, but also emphasize the necessity for community-wide efforts to rigorously test their functions. We believe that collaborative exploration will be key to unlocking the full potential of these newly synthesized proteins for scientific and therapeutic applications. Even with the throughput and results of our work that have greatly transcended prior work, we believe that no single publication can synthesize >20 new membrane proteins and fully validate the functions and structures of all the proteins, which is also not the goal of this work. We have now discussed this in the Discussion to constrain the scope of our claims and to discuss future work (Discussion paragraph 3, page 15).

(3) The rationale behind the selected optimization compounds is unclear. The five compounds affect different steps of cell-free membrane protein expression: Synthesis efficiency (K⁺, Mg⁺, PEG) and membrane association/insertion (lipids, SEC). In the E. coli cell-free system, the complexity of protein synthesis is largely reduced to the basic translation step. Here, the initiation of translation is most critical and several studies showed that corresponding DNA template optimization of the first codons is essential to obtain significant synthesis of many membrane proteins (e.g. Cho et al., 2023). A properly designed DNA template is thus the prime prerequisite for efficient cell-free membrane protein production and this was not considered in the study. The applied random codon optimization does not address the problem of ineffective initiation of translation. A further point is the absence of any redox system. The proper formation of disulfide bridges is essential for the folding of eukaryotic membrane proteins and it is thus unlikely that these targets have been synthesized in good quality. As the authors mentioned, the complexity of the system is rapidly increasing with the number of screening compounds and thus it is doubtful whether all expression steps, synthesis, solubilization and folding, can be covered with one screening program.

We appreciate the reviewer's comment. While we agree that K⁺, Mg²⁺, and PEG8000 will impact the translation process, they can also contribute to the association and insertion of membrane proteins into liposomes. Cations, such as K⁺ and Mg²⁺, have been shown to interact with lipids, causing changes in the hydration state and overall packing of the lipids¹⁵. It has also been shown to directly impact the insertion rates of cell-free expressed proteins¹⁶. While PEG8000 serves as a molecular crowding agent to improve cell-free production¹⁷, it can also interact directly with the surface of the liposome¹⁸. This interaction could interfere with the nascent membrane protein's ability to interact with the membrane and decrease insertion rates. These parameters were chosen because they affected the translation and insertion processes, and the optimal conditions for each protein were presumed to differ. We agree that the motivation behind the choice of these parameters was not well articulated in the manuscript, so we have added a supplementary note detailing it (Supplementary Note 4).

DNA sequence optimization to improve yield has been demonstrated by several groups, but MEMPLEX, in its current form, is limited in its ability to rapidly screen variants of each DNA sequence. Each sequence would need to be individually cloned and added in as a separate DNA sequence. We did however analyze the codon usage in membrane proteins in E. coli compared to its soluble proteins to see if we should bias the codon usage to better adapt the codon choice. We found that the codon usage was nearly identical, so we stuck with the typical codon usage table for E. coli. Engineering of the initial ~30 nucleotides of the sequence to better account for the transcribed RNA structure would be beneficial but was beyond the scope of this work. The same applies to modulating the redox potential for disulfide bond formation. It falls outside the scope of this work but is a relevant parameter to be tested in future work. Both points have been added to the Discussion section of the manuscript (Discussion paragraph 2, page 14).

(4) The authors used 28 membrane protein targets for their study and claim that 17 were successfully synthesized. The results are in agreement with numerous previous cell-free expression screens of membrane proteins covering even higher number of targets (e.g. Bruni 2022, Isaksson 2012, Schwarz 2008, Savage 2007). Having already comprehensive data of cell-free synthesized membrane proteins available, it remains questionable why only few of them have been used for the development of predictive algorithms.

We appreciate the reviewer's insightful comment. We note that the referenced papers (Bruni 2022, Isaksson 2012, Savage 2007) largely relied on detergents to successfully synthesize the target

proteins. In Bruni et al., they reported success for 57% of the tested proteins across all conditions. When looking at proteins that were successfully synthesized using liposomes, the number drops to 9 out of 61 (14.7%) that received a “weak” score or higher, and 4 out of 61 (6.5%) that received a “medium” score or higher. Isaksson et al. claim to produce 37 out of 38, but like Bruni et al., most of the successful reactions used detergents. Only 7 of the 38 (18.4%) were synthesized with lipids. In Savage et al., they only used detergents for all proteins. We focused entirely on lipidic environments to allow for direct use and characterization of the proteins in a near-native environment after the identification of the ideal composition to produce them. In addition, we intentionally chose “difficult” proteins (see Figure 3C) that failed prior attempts. The choice of protein set in each study can greatly affect the perceived “success” rate. This information has been added to the Introduction section of the main text (Introduction, paragraph 2, page 2).

We disagree with the notion of “Having already comprehensive data of cell-free synthesized membrane proteins available”. The best work thus far has systematically studied ~10 synthesis conditions per membrane protein. It is also challenging to simply collect data from different work as the cell-free mix is not standardized nor well-controlled in the field. Therefore, the literature data would be too sparse, too little, and too noisy for inclusion.

Specific comments:

(1) Figs 1A and S2: This is too simplified. The reaction is much more complex as many proteins will be lipid associated or only partially membrane inserted.

We appreciate the reviewer’s comment. We interpret lipid-associated to mean that the protein is non-specifically interacting with the membrane. Our data using size exclusion chromatography includes washing the liposomes, which should remove proteins that are not inserted into the membrane, and we see that the protein remains attached. The scenario of partial insertion is more challenging to rule out as the protein would still be tightly coupled with the membrane. It is, however, not favorable for a membrane protein to be only partially inserted into the membrane. This would mean that the highly hydrophobic transmembrane helices that are not inside the membrane would be trapped in the interface region with the polar lipid heads or in the aqueous phase. The intermediate state of partially inserted membrane protein would likely collapse into a fully inserted state or pull out of the membrane to shield all the hydrophobic residues through aggregation.

However, we agree that the data does not definitively show correct folding, rather that the protein is not aggregated, and so we have changed the text in the figure to say “Folded” instead of “Correctly folded” as there is no guarantee that the helices are packed with one another correctly (Figures 1A and S2). We also added new experimental data (as above) to show the proper binding of antibodies to select proteins.

(2) Figs. S4 and S5: Nanoparticle tracking of liposomes before or after CFPS reaction? Fusion of the liposomes during reaction is quite likely.

Thank you for this comment. We have now included nanoparticle tracking analysis data for samples expressing the protein AqpZ and the plain liposomes to illustrate the change in diameter due to protein expression. The expression of AqpZ leads to a ~50% increase in the median diameter of the liposomes (Liposomes-75nm, AqpZ-115nm, $p < 10^{-4}$). The apparent increase in the liposome size could be due to the modified diffusion constant of AqpZ-liposomes (NTA uses the Stokes-Einstein equation to estimate liposome size from diffusion data) or the fusion of liposomes. This data has been included in Supplementary Figure 4.

(3) Fig. S11: The histogram is labeled as change in yield, but what are the absolute yields? What is the default yield/reaction condition the change is calculated from?

We appreciate the reviewer for bringing up this point. There is no default yield used to calculate the change. Each reaction composition has a unique control reaction, which is composed of the same composition but omitting the liposomes. The change in yield that is shown in this figure is calculated in the same way as the other figures throughout the manuscript. We modified the figure caption to make this histogram clearer (Supplementary Figure 11).

(4) Fig. S12: The figure does not show synthesis as fluorescence does not cover synthesized but precipitated or any other unfolded but non-fluorescent protein. What this figure might rather show is dependency of membrane association/integration on protein features. This is reasonable and a valuable information as it is prerequisite for functional folding. However, it still does not show that the membrane proteins are functionally folded.

We appreciate the reviewer's note. This comment points to a specific instance of the reviewer's first comment about the use of "synthesis" throughout the work. We agree that the nomenclature used is confusing and have changed the text to say successful membrane integration instead of successful synthesis (Supplementary Figure 12).

(5) Fig. S15A: Why such high variations of the Cy5 values of liposomes from different reactions? Wasn't the liposome concentration in each reaction identical? $N=3$ is stated, were these independent experiments, meaning with different batches of Cy3-labeled lipids prepared liposomes? It rather appears that there is a high batch to batch variation of Cy3 fluorescence in liposome preparations. We appreciate the reviewer highlighting this. Three populations of liposomes were prepared, and one was used for a single replicate of each of the tested proteins. The Cy5 values for each replicate for a given protein were very consistent, indicating that the liposome populations are very similar. However, across proteins, we observed differences in the Cy5 intensities even though they were using the same batches of liposomes. This shows that the variance is due to the protein, not the liposome preparation. The likely cause for this difference is the centrifugation step required to pellet large components of the cell-free reaction before adding the sample to the SEC column. Varying pellet sizes were observed across the different proteins, likely explaining the variable loss of liposome fluorescence. It has also been previously observed that the SEC columns used can interact with certain proteins, which could cause a protein-specific change in the number of liposomes that pass through the column. It is important to note that the correlation between the Cy5 and GFP signals in the elution profile is the key message from this figure, not the absolute amount of either value. It is intended to show the association of liposomes with the synthesized membrane protein and then use those filtered samples in flow cytometry to identify the amount of protein attached to each liposome. (Supplementary Figure 15A).

Reviewer #2:

Comments for the Author:

In this interesting work, the authors developed MEMPLEX (Membrane Protein Learning and Expression) which uses a machine learning (ML) approach, nanoliter droplet printing, and a membrane protein reporter in order to design artificial synthesis environments for membrane proteins. Using their approach through MEMPLEX, the authors also found that the hydrophobicity of the membrane-contacting amino acids can help predict membrane protein synthesis in artificial environments. This work is novel and timely in that it combines ML with cell-free synthesis of membrane proteins that are difficult to produce. The manuscript is also well-written, and I have only several comments to strengthen this exciting and significant work.

We thank the reviewer for their time, and we appreciate the acknowledgment of the novelty of our work.

Major comments:

(1) It would be practically more useful if the authors select a few membrane proteins and their synthesis conditions, and then demonstrate their production at a larger scale. I see the technological significance of this work, but such a demonstration will show a bigger impact of this work on the biotechnology field.

Thank you for raising this concern. The initial focus of MEMPLEX was the screening aspect of the platform, and we, therefore, focused more on optimizing throughput than yield in this method. We nevertheless find the concern justified and relevant. We conducted an experiment to evaluate the impact of increasing the reaction volume eight-fold and changing the reaction vessel from a well in a well plate to a microfuge tube. The data has been added as Supplementary Figure 19. The data shows minimal impact on protein production based on the volume or vessel type. The largest statistically significant change we observed was a 16% increase in production when looking at the production of Beta in tubes versus wells at 2uL volumes. (Supplementary Figure 19 and Results Section 3-Paragraph 3-page 10)

An important application of the resulting vesicles is their testing with in vitro cultures. With our current reactions, we obtain a particle concentration of $\sim 10^{12}$ particles/mL. In cell-culture-based assays performed with cell-free generated vesicles, we have found this concentration to be greatly excessive, and have needed to dilute our samples to concentrations of $10^9 - 10^{10}$ particles/mL. To obtain these concentrations in an in-vitro cell-based assay, in a 12-well plate, for example, 1-2uL of a unique cell-free reaction has been amply sufficient. We have included the particle count data but not the in vitro cell-culture testing data as we are wrapping up a manuscript on large-scale functional testing of the resulting vesicles.

(2) Membrane protein biosynthesis is very challenging when using cellular systems as the authors mentioned. Even if the attempts fail, it would be great to compare cell-free production-based yields (from Comment 1-based experiments) with those of cellular protein production. It would be fine for the authors to try a few times and say cellular production of the selected membrane proteins has failed. Such negative data would also make their work more significant.

Thank you for this interesting comment. Firstly, we would like to clarify that cell-free methods complement cell-based methods. We do not state that the cell-free method can yield more protein than the cell-based method. However, cell-free methods have demonstrated advantages in a few ways. A key advantage is that in cell-free reactions, membrane proteins can be directly inserted into the desired lipid environment. In cell-based methods, the protein must be extracted from the native membrane that it is inserted into. This step usually necessitates the use of detergents, which can

damage the protein and limit its downstream utility. Cell-free systems are also more tunable, allowing for the development of tailored environments for each protein, whereas cell-based systems are generally limited to growth condition changes and gene knockouts. Savage et al. 2007⁵ did a thorough investigation into the synthesis of 120 E. coli membrane proteins using both cells and cell-free. They found that some proteins could only be made using the cell-based method but that, in general, cell-free was more effective for a wider range of proteins.

Secondly, the yields obtained MEMPLEX platform, though in small volumes, range from 10-100ug/mL, which is more than sufficient for 1) subsequent cell-culture test as stated in response #1, 2) cryo-EM that requires 0.05-5 μ M¹⁹ (equivalent to 0.1-10 pmol reported in this study).

We have clarified these points in Discussion paragraph 4, page 15.

(3) It would be convenient to see that supp fig captions appear immediately after each figure. It was hard for me to review each supp fig.

We thank the reviewer for the suggestion. We have now included the figure with each supplementary figure caption.

(4) Although cell-free systems give us a lot of promise, there are still many limitations and challenges to overcome, compared to cellular production. I highly recommend that the authors should discuss the limitations of and challenges in cell-free systems, compared to cellular production. As is, I feel the manuscript could be seen as a bit biased article.

We appreciate the reviewer's concern. To address this point, we have added a section to the Discussion to highlight the limitations and challenges of using cell-free protein synthesis for membrane protein synthesis. (Discussion paragraph 4, page 15)

(5) Nanoliter droplet printing is pretty exciting, and readers would appreciate it if the authors could add more explanations to the supplemental information. The manuscript and supplemental information have some texts on it, but a bit more elaboration would be useful.

We appreciate the reviewer's interest in the droplet printing aspect of the work. We have modified the methods section on the droplet printer and added additional details for readers who would like to know more about the system. (Method M5)

Reviewer #3:

Comments for the Author:

In this manuscript, the authors develop MEMPLEX, a high throughput workflow combining cell-free protein synthesis, robotics, and machine learning, to predict optimal cell-free reaction environments for expressing membrane proteins. In an initial set of 8,296 unique cell-free reactions, the authors screen for the expression of 28 different membrane proteins in conditions that modulated five different parameters: potassium concentration, magnesium concentration, PEG8000 concentration, SecYE coexpression, and liposome lipid composition. Importantly, they note that there is no “standard” set of reaction conditions that uniformly results in high soluble protein yields for all membrane proteins and that there is significant interdependence between reaction variables for each membrane protein. Using this initial dataset, they then use an ensemble-based active learning approach to predict expression conditions with increased soluble expression, increasing yields of 21 of 28 proteins. Interestingly, the authors find that they cannot predict optimal expression for new proteins based on sequence or other characteristics, but can predict optimal conditions by evaluating the response to yield under a subset of expression conditions and then comparing that response in the embedded system of the initial dataset/model. Finally, the authors develop a quantitative metric describing how each amino acid residue interacts with its environment that can then be used to predict whether an expression condition exists that will enable successful production.

While the authors present a large body of work, including exploration of thousands of different expression conditions, the lack of emphasis on quantitative yields in g/L (i.e, using relative yields) lowered my enthusiasm. It is also unclear how much this really moves the needle on membrane protein expression. At times, the paper was difficult to follow and some of the data were not presented in a manner that ultimately supported their major conclusions.

We thank the reviewer for their time, and we appreciate the comments regarding our work. The yields were expressed in molar quantities instead of mass per volume so that the reported yields were not dependent on the size of the protein. The molecular weights of proteins in the set span 20-60 kDa, so we felt that the amount of each protein produced was more informative. However, we agree that the standard in the literature is to report yields in mass per volume, so we include a reference conversion of pmol to $\mu\text{g/mL}$. For instance, 1 pmol of a 40 kDa protein in a $2\mu\text{L}$ reaction is equivalent to $20\ \mu\text{g/mL}$. (Results Section 2, Paragraph 2, page 7)

Major comments:

(1) Figure 2D: For calculating the values for Figure 2D, the authors use the top three expressing conditions and calculate the average value for each variable to display in the heat map. If the authors state that each of the variables are interdependent, does displaying average values for individual variables add value if they cannot be interpreted in isolation? Furthermore, given the low, medium, and high values for potassium, magnesium, and PEG8000 concentration are evenly spaced, this would appear to be misleading as one protein that had a top expressing condition with each concentration of potassium would appear the same as a protein that the top condition was exclusively at the medium concentration. It appears that considering the values in the heat map as “optimal reaction compositions” is misleading as in most cases, the actual values shown were never shown or validated. Additionally, the categorical lipid conditions (DMPC, DPPC, and DOPC) are presented as low/medium/high variables alongside the other concentration-dependent parameters. It is unclear how these lipids were assigned values or averaged.

We appreciate the reviewer highlighting this. While we do state that the variables are interdependent, we do not show that until after this panel. The goal of this panel is to highlight that there is no clear

ideal condition for all the tested membrane proteins. This serves as motivation for this protein-specific optimization. The exact values for any given protein should not be used as the target reaction composition as they do not show the best combination of reaction parameters. The text describing these values has been modified to state explicitly that the presented values are averages of the top candidates and not the values that resulted in the highest yield (Results Section 2, Paragraph 3, page 7).

We also appreciate the note about converting the lipid type to a numerical value. The lipid type was converted into numerical values based on their hydrocarbon tail length, i.e., DMPC is 14 and DOPC is 18. Text explaining this has been added to both the main text (Results Section 2, Paragraph 1, page 6) and the Figure 2 caption.

(2) In the discussion, the authors indicate that their platform could be improved upon by incorporating additional variables to test (and spend a significant amount of text listing additional parameters that could be considered). However, they then state that in their current work incorporating 3 additional parameters into the workflow, they were unable to substantially improve protein yield and caution against “the ‘curse’ of expanded search space.” In their final paragraph, the authors again state that their platform could be improved by incorporating a broader parameter search space. These statements are contradictory and call into question the effectiveness of their machine learning model.

We appreciate the reviewer highlighting this point. We agree that the statement in the original version was internally contradictory. The primary challenge with the experiments using the additional reaction parameters was that we increased the number of variables, which greatly increased the total search space, yet we only tested the same number of reactions per protein that we tested in our initial experiments. This discrepancy is the likely cause of the poor model performance. We would likely need to run several more rounds of active learning to train a more accurate model. Future work could further explore these additional variables, but it falls out of the scope of this work. We have modified the text in the Discussion section accordingly (Discussion paragraph 4, page 15). We also hope that our work, being the first in the field, will provide a reference point for the required number of experiments for such machine-learning workflow.

(3) It is unclear how the greater scientific community would be able to apply this workflow to their own work given the complicated nature of the workflow (both requiring advanced robotic systems as well as a large number of models with somewhat limited accuracy). How would a user know what initial reaction conditions to test in order to accurately place a new membrane protein into the embedded space? Do the authors anticipate issues with cross-lab challenges?

Thank you for this comment. Our work has contributed to the scientific and industry communities in a few ways:

Scientific community:

- Conceptual advancement: We demonstrate in this work that it is possible to systematically automate the high throughput screening and synthesis optimization of membrane proteins using a machine-learning approach.
- Conceptual advancement: Prediction of synthesis success of a new membrane protein, proposed by our AlphaFold-informed model. This model enables users to determine a priori whether experimental efforts in synthesizing a given membrane protein are worth pursuing, and whether other similar proteins might be more successfully synthesized in this system.

The predictions remove the typical guesswork before attempting to synthesize membrane proteins using a cell-free approach.

- Technical advancement: This work presents the largest dataset for the screening of cell-free reactions (>20,000) that we have found. Borkowski et al. 2020²⁰ evaluated ~1,000 cell-free reaction compositions expressing the same protein to evaluate their active learning strategy. Pandi et al. 2022²¹ also evaluated ~1,000 reactions to optimize cell-free reaction compositions and a carbon fixation pathway.
- Technical advancement: Once the synthesis condition of a membrane protein is identified, its synthesis does not require the complexity or the throughput presented here. One could simply synthesize the membrane-proteins with the suggested reaction conditions.

Industry or technological community:

- Technical advancement: As the reviewer suggested, the complexity of this system is not amenable for most small labs. In our lab, we could easily execute at least 4 x 384 well plate assays per day. Our work also paves the way for the automated and AI-guided synthesis of membrane proteins in the industry, as recently discussed in this article [Pharma Turns to New Tech for Biologics \(genengnews.com\)](https://www.genengnews.com/pharma-turns-to-new-tech-for-biologics/).
“One of the big problems is making them,” he points out, explaining that inorganic chemical processes are more clear-cut than biological manufacturing, which can be expensive, complex, and unpredictable.

(4) The authors devote the end of the results section to a detailed discussion of an AlphaFold-informed predictive model for the larger unexamined membrane proteome. They post-facto include their tested proteins, which appear to match this model. However, no validation of these results is presented using previously-untested proteins, which makes further claims about predicted protein synthesizability appear unsupported. Did the authors consider validation of this model using MEMPLEX or other methods? This should be done.

Thank you for this excellent observation. We have tested 3 additional proteins, one with high predicted success and low standard deviation among predictions and 2 that were still predicted to be successful but with lower confidence. All 3 proteins passed the threshold set for a successfully synthesized protein. The positions of these proteins are shown in Figure 5D as purple markers. This data is also discussed in Results - Section 5 - Paragraph 4 (page 13).

Minor comments:

(1) Abstract, Sentence 4, as well as Results, Paragraph 1, Sentence 1, the authors characterize their choice of the split folding GFP as a “novel” reporter. However, this reporter protein has been used in a number of different applications. The authors should soften their language regarding the use of this reporter to emphasize the other novel parts of their work.

We appreciate the reviewer’s comment. We agree that the application of the reporter to membrane proteins is novel and not the reporter itself. The text in the Abstract, Introduction paragraph 3, and in the Results sections were changed to reflect that.

(2) Introduction, Paragraph 3, Sentence 5, the authors state that included in the parameters that they modulate are plasmid sequence and liposome concentration. However, the rest of the manuscript/data do not mention altering either of these parameters. The authors should clarify whether these parameters were modulated and to what extent/effect.

We appreciate the reviewer's observation. These parameters were not modulated in the data included in this manuscript, so we have modified the text to remove it (Results Section 2 - paragraph 1, page 6).

(3) Figure 1A: The addition of chaperones to the reaction mixtures is depicted as proteins as opposed to plasmid encoding the proteins for co-translational production. The authors should alter the schematic to make this clear. Furthermore, the authors included the proteins YidC and SecB, which are not mentioned at all in the main text of the manuscript. Were these proteins included in the study?

We appreciate the reviewer's comment. Figure 1A has been modified to show that it is the plasmid encoding the proteins that are added, not the proteins directly. In addition, YidC and SecB were not evaluated in the data presented, and so they were removed from the figure. (Figure 1A)

(4) Results Section 1, Paragraph 4, Sentence 5 refers to Figure 3D. Should this instead read Figure 1D?

We greatly appreciate the reviewer pointing this out. The figure reference has been corrected (Results Section 1 - paragraph 4, page 5).

(5) Results Section 2, Paragraph 1 refers to the inclusion of the translocating channel SecYE. For additional clarity, the authors should consider editing the sentence to indicate the variable is whether to include plasmid for co-expression of SecYE (as opposed to purified SecYE protein). Additionally, do the authors know that SecYE is solubly expressed in all conditions tested? If so, the authors should state this and also provide supporting evidence in the supplementary information.

Thank you for this comment. SecYE is indeed added in the form of a plasmid to be co-expressed in the cell-free reaction; we have modified the text to make this explicitly clear. Figure 1A has also been modified to highlight that the DNA, not the purified protein, was added to the reactions. The E. coli Sec translocon has been demonstrated to be successfully synthesized in cell-free protein synthesis by others²². In addition, we conducted our own expression screen of SecYE using the folding reporter. We found that when the SecE plasmid is added at the same concentration as SecY, successful synthesis is observed for all salt conditions tested. However, low potassium appears to negatively impact synthesis. This additional data is presented in Supplementary Figure 8C. Even though we show synthesis of the translocon in these test reactions, we cannot guarantee that the translocon is successfully made in all reactions where the DNA is added. We further state this in Supplementary Note 4. (Figure 1A, Supplementary Figure 8C, and Supplementary Note 4)

(6) Figure 2A: The authors should consider indicating that the values for SecYE are indicating plasmid concentration, not addition of protein.

We agree with the reviewer's comment and have modified Figure 2A to specifically state that it is the plasmid concentration. (Figure 2A)

(7) Figure 2A caption: The authors highlight certain conditions as "standard" reaction conditions. What is the rationale behind considering these conditions "standard"?

We appreciate the reviewer highlighting the confusion around the "standard" condition. The "standard" CFPS composition was set as the tested concentration, which was most like the average of the common reagent concentrations seen across other cell-free papers. The primary motivation for including this was to highlight that using the same reaction composition that is found to be most effective for producing GFP might not be the best for all other proteins. We have added a

description of the “standard” condition in the main text of the manuscript. (Results Section 2 - paragraph 3, page 7)

(8) Figure 2B: For additional clarity, the authors should include in the figure captions what the magenta circles represent.

We appreciate the reviewer’s observation. The magenta circles represent the reactions that passed both the yield and significance thresholds. We have modified Figure 2B caption to explain this. (Figure 2B)

(9) Figure 2E: The authors should add information in the figure captions indicating that the red and blue outlined boxes are pointing out the specific variables displayed in panels 2F and 2G.

We appreciate the reviewer’s observation. We have modified the caption of Figure 2E to explicitly indicate the relationship between the shown boxes and the data in 2F and 2G. (Figure 2G-F)

(10) Figure 2G and F. The authors should consider displaying error bars more clearly, since some error bars are almost fully covered by others.

We agree with the reviewer’s recommendation. We have modified both Figures 2F and 2G to have end caps to make the positions clearer. (Figure 2F&G)

(11) Figure 3C: Why color code the “Made Previously” label and only provide a legend in the caption?

We appreciate the reviewer’s critique. We agree that a legend would make the figure easier to read. We have added a legend to explain the previously made colors. (Figure 3C)

(12) Results Section 2, Paragraph 1: The authors state that they investigated 5 reaction parameters, but only explicitly name 4. Based on the figures, the two salts were considered to be separate parameters, but this is not clearly indicated within this section since the specific salts are named in parentheses.

We appreciate the reviewer's comment. The text made it unclear that the concentrations of Magnesium and Potassium were independently modified. We have clarified the text (Results section 2, paragraph 1, page 6).

(13) Results Section 2, Paragraph 2: The authors state that the third criteria used to select membrane proteins was that the proteins were “expected to give a variety of different or zero yields”. Why did the authors believe a priori that some membrane proteins would not express in any of the tested conditions? Additionally, the criteria for selection are stated awkwardly, making it difficult to determine the authors’ rationale for choosing these proteins.

Thank you for the comment. We agree that the language used while explaining the selection criteria was confusing. The statement “expected to give a variety of different or zero yields” referred to the outcomes that were observed in other papers. We selected proteins that have been shown to work and some that were shown to fail. We have modified the text to clarify the selection criteria (Results section 2 - paragraph 2, page 6).

(14) Results Section 2, Paragraph 3: The phrasing of the sentence, “For 39.3% of the proteins, the top 3 reactions used 2% PEG while 25% of them used 0%” it is difficult to parse the meaning. Presumably this refers to 25% of the top reactions and 0% PEG.

We appreciate the reviewer pointing this out. The text has been modified to clarify the confusion (Results Section 2 - paragraph 3, page 7).

(15) Results Section 3, Paragraph 3: The authors state that for each of the 28 membrane proteins, they selected 12 reaction conditions to test. However, they also indicate that they performed >11,000 unique reactions. These values appear to be one order of magnitude inconsistent with one another - how was the >11,000 calculated?

Thank you for highlighting this. The text was not as explicit as it should have been. The 11,000 reactions refer to the complete set of reactions that were tested, which includes the reactions from the screen and the reactions selected through active learning, where each replicate is included as a unique observation. We have modified the text to make this clear. (Results section 3 - paragraph 3, page 9)

Reviewer #4:

Comments for the Author:

Meyer et al. report carrying out a high-throughput screen of in vitro translation conditions to improve expression yields of 28 E. coli, A. thaliana, or human transmembrane proteins using E. coli whole-cell extract, liposomes, and the SecYE translocon. The authors screen three concentrations of three buffer components (85, 135, or 185 mM K⁺; 8, 14, or 20 mM Mg²⁺; and 0, 1, or 2% PEG), three types of lipids for the liposomes (DOPC, DMPC, or DPPC), and run reactions with or without added plasmids for co-expression of the translocon. (Presumably, in the absence of added translocon plasmids, translocons come from the whole-cell lysate.) Robotic pipetting (aka “droplet printing”) helps the authors assemble 2- μ L reactions in 384-well plates. Split GFP is used as a reporter of protein aggregation, with GFP-dark reactions presumed to have failed the expression test due to aggregation of the expressed protein and GFP fluorescence used as a proxy for the concentration of properly folded proteins in the rest. Finally, the authors carry out statistical analyses, embedding, and neural network training in an effort to predict optimal expression conditions for transmembrane proteins. Using a relatively generous threshold of >1 pmol protein produced with $p < 0.05$ for protein expression, the initial screening campaign of ~8,000 reactions (including replicates), the authors report expression of all 28 proteins, including 7 that had eluded prior efforts at in vitro expression in liposomes. However, in some cases “expression” may be mere noise of the GFP assay, and this needs to be clarified. This initial dataset is used to predict more optimized expression conditions via machine learning, which motivates a second screening campaign of ~11,000 reactions, as a result of which the authors report improved yields for 21 of the 28 proteins compared to the initial, randomly chosen reactions. An expansion of the search space and a further 17,000-reaction screening campaign does not lead to further improvement. Low-throughput validation of expression of a small subset of proteins is carried out.

To rationalize their results, the authors turn to basic and statistical principles of protein-lipid interaction: specifically, hydrophobicity of lipid-facing side chains and the statistical likelihood of a given type of residue on the lipid-facing surfaces of a training set of transmembrane protein structures in the PDB. Only alpha-helical proteins are examined; the authors do not consider beta-barrel proteins. These properties of the protein surface are found to be useful predictors of its expression yield in the liposomes.

Overall, this is an ambitious and integrated computational-experimental method development study addressing a major challenge in biochemistry of transmembrane proteins. The data are likely to be of interest to researchers in this field, assuming data analysis is valid. The quality of writing is good, particularly in presenting examples of computational and experimental conditions, and the figures are mostly clear and informative. The Discussion section addresses several important points, notably the omission of cholesterol or any other sterols from the lipid membranes tested in this study. There are several other weaknesses in the study as currently reported, which should be addressed. The methodology is innovative and ambitious, but some aspects of experimental methodology and statistical analysis needs clarification, and there seems to be a tendency to “overhype” results.

We thank the reviewer for their time, and for acknowledging the novelty in our work.

Major comments:

(1) First, I find myself confused about the statistical significance of results in this study. If I understand correctly, the “ Δ Yield” metric refers to the difference in split-GFP intensity in the presence vs. absence of liposomes. As such, a Δ Yield of 0 should be interpreted as no detectable expression of the protein in liposomes. The average Δ Yield exceeds 1 pmol for only 3 of the 28

proteins in Figure 2C or Supplementary Figure 10 during the initial screening campaign of ~8,000 reactions. Following the authors' machine learning campaign, this number increases to 7 of the 28. Apparently, the authors trained a neural network model on 8,000 reactions across 28 proteins despite the training data for 25 of the 28 proteins being mostly noise? And for the few conditions that did yield statistically significant expression, is $p < 0.05$ truly an adequate metric? Did the authors include any correction for multiple hypothesis testing?

Thank you for this observation. You are correct that for many of the proteins, most of the reactions yielded a minimal increase in signal. This is likely why the active learning method failed for the low-producing proteins. With no signal, there is little informative data. We did, however, find that even when none of the screening reactions passed the threshold set for a successful reaction, the changes in the signal could lead the model to find better reactions.

We would like to clarify that this figure only tests a single hypothesis, i.e., whether two populations of a given unique reaction, one with liposomes and one without, are statistically distinct from one another. Given this is a single hypothesis, we deem the t-test p-value to be an adequate metric for statistical significance, without the need for any further corrections.

How many replicates were used for each condition, and how many were discarded as outliers due to being >2 SDs away from the mean of other replicates? As with any high-throughput screen, a detailed discussion of the statistical analysis should be a very important part of the Methods.

Thank you for pointing this out. All reactions were tested in at least quadruplicate. If an outlier was removed, there would still be at least three replicates for downstream statistical testing. Out of all reactions, 7.5% (874 reactions) were flagged as outliers and removed from the data set. We have modified the methods of data preparation to include this information. (Method 17 paragraph 1)

(2) Second, the paper does not straightforwardly evaluate the screening campaigns' success. The authors' dataset includes seven proteins for which previous efforts to reconstitute them in liposomes were unsuccessful. How many of these seven previously failed proteins were successfully expressed in the present study, when multiple hypothesis testing is taken into account?

Thank you for this comment. Like our response above, only a single hypothesis is tested for each protein in a unique reaction composition. The liposome and corresponding no-liposome replicate reactions of a unique composition are taken as the two populations and evaluated to assess if they are statistically significantly different from one another. If the p-value is low enough to reject the null hypothesis that the two populations are the same and if the difference between the mean of the populations is above the threshold, then the reaction composition is labeled as successfully produced. We do agree that the success of the screening campaign could be more explicit, so we have clarified the text accordingly (Results Section 3 - paragraph 3, page 9).

Has their successful expression been validated using low-throughput methods? If not, doing the validation would dramatically improve the biological usefulness of the present study.

Thank you for this remark. We used several low throughput methods to validate successful expression. We used size-exclusion chromatography and flow cytometry (shown in Figure 1C-E and Supplementary Figures 6 and 15) to demonstrate the stable insertion of the protein into the liposomal membrane. We also use Western Blots, which is the gold standard for membrane protein expression papers^{3,4,5}, to show the binding of CD47 with an antibody specific to one of the protein's extracellular domains (shown in supplementary figure 18A). In new results, we performed additional flow cytometry experiments using antibodies that are specific to CD9 and CD81, and the results show clear enrichment of this protein on the liposomal membrane (Supplementary Figure 18B&C)

A related point is that the success of the computational prediction does not seem to be rigorously evaluated. It certainly seems to have improved expression yields for most proteins, but what would be the expected outcome if the authors simply screened 11,000 conditions chosen randomly from the same search space as the 11,000 they did screen in the second screening campaign?

We appreciate the reviewer's question. To answer this question, we conducted an additional analysis to investigate the difference between active learning and randomly selected reactions. We trained an ensemble of 45 neural networks using different model parameters on all available data and predicted all possible reaction conditions that could be tested for each protein. We use the outcomes of all possible reactions to represent the random sampling population. We then compared all possible outcomes with the outcomes of the reactions chosen by active learning.

Below is a summary plot showing the percent difference between the median Δ Yield for each protein on the y-axis, and the median Δ Yield for the active learning selected reactions on the x-axis. This plot illustrates the improvement of membrane-protein production due to the use of the active learning method vs. random sampling. This plot shows that only 4 proteins see less than a 100% increase in the median Δ Yield when using active learning. The 4 proteins have generally low yield (x-axis), resulting in less effective active learning to enhance their production yield.

We included another plot illustrating the likelihood of finding a reaction that produces a Δ Yield of greater than 0.5pmol. This data shows the advantage of active learning in finding reactions with a Δ Yield of greater than 0.5pmol.

(3) Third, what is the topology of protein insertion into the membrane for the 28 proteins chosen in this study? Since liposomes are generated prior to protein insertion, and the split-GFP tag is always on the C-terminus of each protein, this means the C-terminus of each protein must be on the outside of the liposome (equivalent to the cytoplasmic side of the cell membrane) in order for the split-GFP assay to detect it. Do all 28 proteins have cytoplasmic C-termini? If not, then the assay used in this study would not be expected to work for them. Or have the authors previously observed that proteins in this system incorporate into the liposomes in random orientation?

We appreciate the reviewer's observation. All 28 proteins tested in this work indeed have cytoplasmic C-termini. A supplementary note has been included to clarify this. This criterion was specified in the selection criteria used to choose the set of proteins (Results Section 2 - paragraph 2, page 6 and Supplementary Note 3).

(4) Fourth, the claim that the authors “discover a new metric, based on hydrophobicity of the membrane-contacting amino acids, that predicts membrane protein synthesis” seems like a stretch. It is rather obvious that proteins with a hydrophobic transmembrane surface are more stable within lipid bilayer membranes than proteins with a hydrophilic surface. The bigger biological puzzle here is why do some natural transmembrane proteins have more hydrophilic surfaces than others.

We appreciate the reviewer's comment and acknowledge that the metric highlighted aligns well with the current understanding of membrane protein insertion energetics. However, we are the first to develop a quantitative method to assess this and relate it to the experimental results. There are additional seemingly intuitive hypotheses that were rejected through this investigation, such as the association between length of protein or number of transmembrane helices with synthesis success. Length appeared to have limited predictive power on success unless the protein was below 25 kDa. It would also be assumed that proteins from the same organism would require similar reaction conditions. However, this was also shown not to be the case. We do, however, agree that the language could be toned down, so we have changed the main text and abstract to reflect that. (Abstract, Results Section 5 - paragraph 1, page 11)

(5) Fifth, the authors mention chaperone proteins as one of the important conditions, and they are among the four reaction components depicted in Figure 1. However, I could not find any results with chaperones in the paper. Did the authors add chaperones or vary their concentrations in any experimental conditions? If not, then please remove them from Figure 1 to avoid confusion. Thank you for highlighting this. We did not include any other chaperones in the data presented. We modified the figure to clarify that the only chaperone used was the Sec translocon (Figure 1A).

(6) Sixth, I am confused about the batch-to-batch benchmarking of whole-cell extracts in this study. The authors state in M2 that each extract was “benchmarked against the previously used extract across several different magnesium (8-16mM) and potassium (85-185mM). The coefficient of variance (standard deviation / mean) between the replicates of each extract at each unique salt concentration were calculated to assess their similarity.” How was this benchmarking done? Was it by expressing a subset of proteins and monitoring split-GFP fluorescence? What are the replicates the authors reference in this case? Do the authors mean variance in split-GFP fluorescence among identically prepared reaction mixes using extract N and extract N+1?

We appreciate the reviewer’s comment. The benchmarking was conducted by expressing deGFP in various different salt conditions using the new extract and the previous extract. The signals observed for each extract at each salt concentration were compared side-by-side to identify differences in the new extract’s performance compared to the previous one. All reactions were run in triplicate and used to calculate the variance between the extracts’ performances. The text in M2 has been modified accordingly (Method 2 paragraph 2 and Results Section 1 - paragraph 1, page 3-4).

(7) Seventh, the authors mention in M17 that, in the interest of increasing the number of data points available for training, replicate samples were treated as independent samples, but “All replicates for a given reaction condition were kept within the same split.” Could you please clarify this statement? We thank the reviewer for this comment. We agree that the text was confusing and have changed it to “All replicates of a single reaction composition were kept in either the training or the testing set when training the neural networks” (Method 17, paragraph 2). The motivation behind this method of data splitting was to avoid giving the training set an unfair advantage by showing it some of the replicate data that was in the test set.

(8) Eighth, M20 states that “A concave hull around the membrane spanning region was calculated for each protein.” How was it calculated?

We appreciate the reviewer identifying this omission. A surface to represent the closest position of the surrounding lipids was calculated using a three-dimensional alpha shape²³. This surface contours to the shape of the membrane protein by identifying the outermost amino acids based on their proximity to the surrounding amino acids. The outer amino acids are presumed to be in contact with the membrane. The alpha shape is dependent on the parameter α , which determines how tightly the surface fits to the protein. This value was scaled in relation to the size of the protein to ensure the surface matched the protein well. The bounding surface that was located outside of the membrane was extended far from the protein so that the distance between each amino acid and the surface always went to the sides of the protein instead of the top or bottom. This description has been added to the method section on structure analysis and feature extraction (Method 20, paragraph 1).

References

- (1) MacKenzie, K. R. Folding and Stability of α -Helical Integral Membrane Proteins. *ChemInform* **2006**, *37* (31). <https://doi.org/10.1002/chin.200631299>.
- (2) Cymer, F.; von Heijne, G.; White, S. H. Mechanisms of Integral Membrane Protein Insertion and Folding. *Journal of Molecular Biology* **2015**, *427* (5), 999–1022. <https://doi.org/10.1016/j.jmb.2014.09.014>.
- (3) Bruni, R.; Laguerre, A.; Kaminska, A.-M.; McSweeney, S.; Hendrickson, W. A.; Liu, Q. High-Throughput Cell-Free Screening of Eukaryotic Membrane Protein Expression in Lipidic Mimetics. *Protein Science* **2022**, *31* (3), 639–651. <https://doi.org/10.1002/pro.4259>.
- (4) Isaksson, L.; Enberg, J.; Neutze, R.; Göran Karlsson, B.; Pedersen, A. Expression Screening of Membrane Proteins with Cell-Free Protein Synthesis. *Protein Expression and Purification* **2012**, *82* (1), 218–225. <https://doi.org/10.1016/j.pep.2012.01.003>.
- (5) Savage, D. F.; Anderson, C. L.; Robles-Colmenares, Y.; Newby, Z. E.; Stroud, R. M. Cell-Free Complements in Vivo Expression of the E. Coli Membrane Proteome. *Protein Science* **2007**, *16* (5), 966–976. <https://doi.org/10.1110/ps.062696307>.
- (6) Sonnabend, A.; Spahn, V.; Stech, M.; Zemella, A.; Stein, C.; Kubick, S. Production of G Protein-Coupled Receptors in an Insect-Based Cell-Free System. *Biotechnology and Bioengineering* **2017**, *114* (10), 2328–2338. <https://doi.org/10.1002/bit.26346>.
- (7) Stech, M.; Quast, R. B.; Sachse, R.; Schulze, C.; Wüstenhagen, D. A.; Kubick, S. A Continuous-Exchange Cell-Free Protein Synthesis System Based on Extracts from Cultured Insect Cells. *PLOS ONE* **2014**, *9* (5), e96635. <https://doi.org/10.1371/journal.pone.0096635>.
- (8) Jacobs, M. L.; Boyd, M. A.; Kamat, N. P. Diblock Copolymers Enhance Folding of a Mechanosensitive Membrane Protein during Cell-Free Expression. *Proceedings of the National Academy of Sciences* **2019**, *116* (10), 4031–4036. <https://doi.org/10.1073/pnas.1814775116>.
- (9) Baumann, A.; Kerruth, S.; Fitter, J.; Büldt, G.; Heberle, J.; Schlesinger, R.; Ataka, K. In-Situ Observation of Membrane Protein Folding during Cell-Free Expression. *PLOS ONE* **2016**, *11* (3), e0151051. <https://doi.org/10.1371/journal.pone.0151051>.
- (10) Ataka, K.; Baumann, A.; Chen, J.-L.; Redlich, A.; Heberle, J.; Schlesinger, R. Monitoring the Progression of Cell-Free Expression of Microbial Rhodopsins by Surface Enhanced IR Spectroscopy: Resolving a Branch Point for Successful/Unsuccessful Folding. *Front. Mol. Biosci.* **2022**, *9*. <https://doi.org/10.3389/fmolb.2022.929285>.
- (11) Ernst, O. P.; Lodowski, D. T.; Elstner, M.; Hegemann, P.; Brown, L. S.; Kandori, H. Microbial and Animal Rhodopsins: Structures, Functions, and Molecular Mechanisms. *Chem. Rev.* **2014**, *114* (1), 126–163. <https://doi.org/10.1021/cr4003769>.
- (12) Gunasekaran, M.; Bansal, S.; Ravichandran, R.; Sharma, M.; Perincheri, S.; Rodriguez, F.; Hachem, R.; Fisher, C. E.; Limaye, A. P.; Omar, A.; Smith, M. A.; Bremner, R. M.; Mohanakumar, T. Respiratory Viral Infection in Lung Transplantation Induces Exosomes That Trigger Chronic Rejection. *The Journal of Heart and Lung Transplantation* **2020**, *39* (4), 379–388. <https://doi.org/10.1016/j.healun.2019.12.009>.
- (13) Zhang, X.; Borg, E. G. F.; Liaci, A. M.; Vos, H. R.; Stoorvogel, W. A Novel Three Step Protocol to Isolate Extracellular Vesicles from Plasma or Cell Culture Medium with Both High Yield and Purity. *Journal of Extracellular Vesicles* **2020**, *9* (1), 1791450. <https://doi.org/10.1080/20013078.2020.1791450>.
- (14) Hasselmann, J.; Coburn, M. A.; England, W.; Figueroa Velez, D. X.; Kiani Shabestari, S.; Tu, C. H.; McQuade, A.; Kolahdouzan, M.; Echeverria, K.; Claes, C.; Nakayama, T.; Azevedo, R.; Coufal, N. G.; Han, C. Z.; Cummings, B. J.; Davtyan, H.; Glass, C. K.; Healy, L. M.; Gandhi, S. P.; Spitale, R. C.; Blurton-Jones, M. Development of a Chimeric Model to Study and Manipulate

- Human Microglia *In Vivo*. *Neuron* **2019**, *103* (6), 1016-1033.e10.
<https://doi.org/10.1016/j.neuron.2019.07.002>.
- (15) Binder, H.; Zschörnig, O. The Effect of Metal Cations on the Phase Behavior and Hydration Characteristics of Phospholipid Membranes. *Chemistry and Physics of Lipids* **2002**, *115* (1), 39–61.
[https://doi.org/10.1016/S0009-3084\(02\)00005-1](https://doi.org/10.1016/S0009-3084(02)00005-1).
- (16) Altrichter, S.; Haase, M.; Loh, B.; Kuhn, A.; Leptihn, S. Mechanism of the Spontaneous and Directional Membrane Insertion of a 2-Transmembrane Ion Channel. *ACS Chem. Biol.* **2017**, *12* (2), 380–388. <https://doi.org/10.1021/acscchembio.6b01085>.
- (17) Tan, C.; Saurabh, S.; Bruchez, M. P.; Schwartz, R.; LeDuc, P. Molecular Crowding Shapes Gene Expression in Synthetic Cellular Nanosystems. *Nature Nanotech* **2013**, *8* (8), 602–608.
<https://doi.org/10.1038/nnano.2013.132>.
- (18) Mishima, K.; Satoh, K.; Suzuki, K. Increase in Molecular Order of Phospholipid Membranes Due to Osmotic Stress by Polyethylene Glycol. *Colloids and Surfaces B: Biointerfaces* **1997**, *10* (2), 113–117. [https://doi.org/10.1016/S0927-7765\(97\)00047-7](https://doi.org/10.1016/S0927-7765(97)00047-7).
- (19) Passmore, L. A.; Russo, C. J. Chapter Three - Specimen Preparation for High-Resolution Cryo-EM. In *Methods in Enzymology*; Crowther, R. A., Ed.; The Resolution Revolution: Recent Advances In cryoEM; Academic Press, 2016; Vol. 579, pp 51–86.
<https://doi.org/10.1016/bs.mie.2016.04.011>.
- (20) Borkowski, O.; Koch, M.; Zettor, A.; Pandi, A.; Batista, A. C.; Soudier, P.; Faulon, J.-L. Large Scale Active-Learning-Guided Exploration for in Vitro Protein Production Optimization. *Nat Commun* **2020**, *11* (1), 1872. <https://doi.org/10.1038/s41467-020-15798-5>.
- (21) Pandi, A.; Adam, D.; Zare, A.; Trinh, V. T.; Schaefer, S. L.; Burt, M.; Klabunde, B.; Bobkova, E.; Kushwaha, M.; Foroughijabbari, Y.; Braun, P.; Spahn, C.; Preußner, C.; Pogge von Strandmann, E.; Bode, H. B.; von Buttlar, H.; Bertrams, W.; Jung, A. L.; Abendroth, F.; Schmeck, B.; Hummer, G.; Vázquez, O.; Erb, T. J. Cell-Free Biosynthesis Combined with Deep Learning Accelerates de Novo-Development of Antimicrobial Peptides. *Nat Commun* **2023**, *14* (1), 7197. <https://doi.org/10.1038/s41467-023-42434-9>.
- (22) Matsubayashi, H.; Kuruma, Y.; Ueda, T. In Vitro Synthesis of the E. Coli Sec Translocon from DNA. *Angewandte Chemie International Edition* **2014**, *53* (29), 7535–7538.
<https://doi.org/10.1002/anie.201403929>.
- (23) Edelsbrunner, H.; Mücke, E. P. Three-Dimensional Alpha Shapes. *ACM Trans. Graph.* **1994**, *13* (1), 43–72. <https://doi.org/10.1145/174462.156635>.

Cheemeng Tan
Associate Professor, Department of Biomedical Engineering
One Shields Ave, Davis, CA 95616

Phone: 530-752-7849
Fax: 530-754-5739
email: cmtan@ucdavis.edu

Dec 4th, 2024

Dear Reviewers,

Please find attached the revised manuscript, “Designer artificial environments for membrane protein synthesis” for publication in Nature Communications.

We greatly appreciate the feedback with regards to our manuscript. We have addressed all reviewer comments, and all changes are highlighted in yellow. Below, we have included a detailed point-by-point response to comments by the three reviewers.

I hope that you will find our response adequate. Please let us know if you have any concerns regarding the changes that we made.

Reviewer comments

Reviewer #1:

Comment 1: The first challenging step in cell-free protein synthesis is screening of proper translation by modification of the initiation of translation. As shown in previous publications, this can bring membrane protein production from almost nothing to mg amounts per mL. If the basic production is already suboptimal, which can thus be assumed for many of the targets, the value of any further solubilization optimization by screening of compound concentrations is limited. Furthermore, membrane protein production and solubilization/membrane insertion must not be correlated. The co-translational membrane protein solubilization depends on the presence of a suitable hydrophobic environment in sufficient concentration. The screening strategy included a mix of compounds affecting yield (Mg²⁺, K⁺, PEG) or membrane insertion (lipid types). However, as only membrane insertion/association was monitored as output, the effect of the yield affecting compounds is not clearly evident.

The presented pipeline is thus to my opinion preliminary and would require significant modifications such as an initial production screening (i.e. the optimization of the pure protein yield e.g. by template design and others), for rather professional applications. The current protocol omits this first step and instantly starts with quality screening (i.e. membrane insertion and folding). Of course, this may still give valuable hints for the lipid-based solubilization of individual membrane proteins and their preferences for some of the tested compounds, but any messages of yields or efficiencies are relative to the implemented suboptimal conditions and thus less meaningful.

We thank the reviewer for making this point. We have broken the response to the multifaceted comment below:

The main novelty and significance of our work lies in using a high-throughput and machine learning approach to learn the artificial “chemical” condition for cell-free membrane-protein synthesis.

Our work intentionally focuses on learning the chemical environment of cell-free protein synthesis systems in optimizing membrane-protein insertion into liposomes. This focus differentiates our work from the majority of membrane-protein work in the literature. It is the largest and most systematic dataset collected thus far on different chemical conditions. It is also the first machine-learning work applied to such datasets. Furthermore, we find new insights into protein structure that predict successful synthesis in liposomes, now validated by additional experiments requested by other reviewers. This and other reviewer comments do not challenge this main notion.

We do not claim that our work represents the complete pipeline for optimizing membrane-protein synthesis. We also do not claim that our membrane-protein yield is the highest in the literature. Instead, our work contributes significantly to the conceptual idea of leveraging machine learning to learn how the chemical environment impacts cell-free protein synthesis, a research area that is scarce in the literature. MEMPLEX is also a novel module that can be incorporated into existing lab or industry workflows for synthesizing membrane proteins using *E. coli* cell-free protein synthesis. Notably, our approach has enabled the synthesis of new membrane proteins that were not possible by others (Main Figure 3C).

Overall, we believe it is beyond any scientific work's scope to claim a complete "professional" pipeline of synthesizing membrane proteins. Such a claim is not feasible and unfair to impose on our work, given the number of additional optimizations (including but not limited to translation rate, reactor design, lipids, etc.) required. We believe that our protocol already provides valuable data that can be plugged into the existing membrane-protein synthesis pipeline. While it is indeed an early-stage approach, the insights gained are highly relevant for understanding membrane protein behavior and can guide future optimization efforts. The findings from our study could be particularly useful for researchers interested in specific lipid environments or solubilization conditions, making our protocol a valuable tool in these contexts.

We have now clarified this point in the Discussion.

Importance of other parameters in optimizing membrane-protein insertion into liposomes

We agree with the reviewer that optimizing other parameters, including protein coding sequence, translation initiation, and chaperones, may enhance the synthesis of membrane proteins. As above, we do not claim that these optimizations are not important/relevant for membrane-protein synthesis. In addition, such work is abundant in the literature, as pointed out by the reviewer. Such well-established studies can be incorporated modularly into MEMPLEX, either before or after optimizing the chemical conditions. Incorporating these additional parameters directly into our machine-learning workflow is not trivial, but they would be interesting for future work. Such work requires additional and drastic investment because no computational approach has been able to predict the translation rate of protein in different chemical conditions – one of the best calculators (e.g. the translation calculator by the Salis Lab) to date only works for "standard" cellular conditions. We have added this point in Discussion.

We disagree with the reviewer's generalized argument, "Furthermore, membrane protein production and solubilization/membrane insertion must not be correlated."

The argument likely applies to certain protein synthesis systems that synthesize membrane proteins first, followed by purification, and then incorporation into liposomes. In such systems, total protein synthesis and subsequent solubilization are likely not correlated.

However, the argument "Furthermore, membrane protein production and solubilization/membrane insertion must not be correlated" is unlikely to apply to our system. In our one-shot cell-free membrane-protein synthesis workflow, protein insertion into the membrane

will depend on the synthesis rate of each protein. In the context of a rapidly folding or stable protein, increasing total protein production would increase the amount of protein available to interact with liposomes. It would, therefore, increase the likelihood of protein inserting into the liposome. However, if the protein is prone to aggregation, the increase in protein production can cause the new proteins to be more likely to interact with one another and form aggregates than it would be to interact and insert into a liposome. Indeed, our new analysis has shown a lack of correlation between the translation initiation rate and the synthesis yield of our membrane proteins (see details in our response below).

The critique of the mixed approach to using compounds that affect both yield and membrane insertion is well noted. However, our strategy was to take a holistic view of protein behavior in a common one-shot cell-free protein-synthesis system (where protein synthesis and insertion occur simultaneously), so as to improve our active learning model and broaden the scope of its applications. By including compounds that report both yield and membrane insertion, we aimed to capture a more comprehensive picture of how these factors interplay. The data on membrane insertion and folding under varying conditions still provide valuable information that can inform future optimization efforts.

Given this, screening for high-yield total-protein synthesis before screening for insertion could lead us down the wrong optimization path for hard-to-produce proteins. Therefore, we focused on optimizing our ability to rapidly screen many conditions for insertion, as that is the fundamental focus of this pursuit. Furthermore, yield optimization can be performed separately, and future work can be done once the optimal chemical cell-free environments are identified. Whether translation initiation should be optimized before or after MEMPLEX is a nuanced topic that is debatable (based on our new analysis) and beyond the scope of our work.

Impact of coding sequences on successful insertion of proteins

We agree that the initial stages of protein synthesis, such as translation initiation, are crucial, and that a properly designed DNA template is an important prerequisite for efficient cell-free membrane protein production. If the total synthesis level of a protein is already suboptimal, optimizing the chemical environment is likely less useful.

Here, we conducted an analysis of the transcription and translation rate of our genetic modules using the best calculators available^{1,2} (Figure 1A-B). The results show that in our system, there is no correlation between protein translation/transcription rate and the membrane insertion yield. The results support the rationale of optimizing the chemical environments of cell-free membrane-protein synthesis and that optimizing the translation rate or expression template is not strictly a prior requirement before executing our pipeline. Again, we agree that our membrane-protein yield may be further optimized by considering other factors, such as translation initiation rate, reaction volume, and lipid type. But, these optimizations are beyond the scope of our manuscript claim and can be incorporated by others or us in future work.

Figure 1: Lack of correlation between the observed membrane protein yield and the predicted transcription/translation rates. A) The calculation of the max transcription rate, conducted based on LaFleur et al.¹, shows that even the protein with the lowest calculated transcription rate can have a high protein yield. **B)** The calculation of the max translation initiation rate, conducted based on the Cetnar and Salis study², also shows no clear correlation between the observed yield and the predicted rate.

We have clarified these points in the Discussion.

Comment 2: I completely agree that GFP fluorescence would be great for htp screening. But unfortunately, it does not correlate with function. The authors speculate that split GFP may be a better folding monitor if compared with terminally attached full-length GFP. I rather think that it may be worse as the large GFP domain is added in already folded condition and the presentation of a small terminal domain attached to an unfolded or partly unfolded membrane protein appears to be even more likely. To show the correlation of split GFP fluorescence in a number of conditions with the activity of at least one target as proof of principle would thus still significantly improve the manuscript and support the value of the presented strategy. Suitable targets of the list are e.g. the adrenergic receptors, where well established and relatively easy to perform binding assays are available. The shown antibody approach is of limited value as it only shows the recognition of relatively short domains and not the function of a protein. The argument that in vivo the antibodies were shown to recognize functionally folded protein does not work for the cell-free system. As with FSEC, where in vivo the folding of GFP can be correlated with folded membrane protein, this cannot be assumed for the cell-free system as any proof-reading systems removing non-functional proteins from the membrane are missing. The mentioned liposome control does not consider improperly inserted, unfolded and partially unfolded membrane proteins which are then in the soluble fraction in presence of liposomes. Again, the pure solubilization of a cell-free synthesized membrane protein by supplied lipids does not indicate its functional folding. The authors also ignore previous publications showing that frequently only fractions of cell-free produced and membrane solubilized membrane proteins are functionally folded.

The authors mention that in previous htp reports physical methods such as solubilization, labelling, SEC profiling, thermophoresis or CD spectroscopy have been used to speculate about membrane protein functionality and they now combined a higher number of these techniques in their manuscript. I agree with the authors that these techniques are not proof of functionality, but also

a more comprehensive combination of these techniques cannot replace a quantitative functional assay. In addition, the presented production of bacteriorhodopsin shows that their cell-free system is able to produce functional protein, which was not doubted. But it has nothing to do with the split GFP approach and thus does not support it.

Thank you for this excellent observation. We once again break our response up below:

Full-length versus Split-GFP reporter

The hypothesis that tagging proteins with split GFP may be worse compared to terminally attached full-length GFP is a valid one that we considered as well. As mentioned in our previous response, when synthesizing proteins with terminally attached full-length GFP and no liposomes, we observed significantly higher background fluorescence than when using the GFP complementation assay. This observation aligns with the results shown by Jacobs et al³.

Our data highlights the difference between using the full-length GFP and using the split GFP-based folding reporter.

Figure 2: Full-length GFP reporter causes higher background signal than split-GFP reporter

The key difference between the two reporters is the signal seen when DNA is added, but liposomes are omitted. This represents the background signal seen when the protein should be produced, but all of it should aggregate as there is no membrane to insert into. The high background of the full-length GFP reporter motivated us to switch to the split GFP-based reporter. Comparing the fold change between the reactions with the most DNA and most liposomes to the corresponding reaction without liposomes, the fold change for the full-length GFP reporter is 3.5 versus the split GFP reporter, which is 14.7.

However, we do agree that there is a possibility of producing a false positive signal with any reporter and that the signal could depend on protein and reaction conditions. To address this, we doubled the number of reactions that we ran. Specifically, we tested no-liposome conditions for every unique reaction composition to quantify the maximal possible fluorescent signal obtained by the complementation assay from aggregated, partially folded, or unfolded membrane proteins. This approach effectively allows us to measure the ‘false positive’ signal—the amount of small GFP fragments still capable of binding the large GFP fragment even when the membrane protein is not properly incorporated into the membrane.

By subtracting this signal from the total fluorescence observed in the liposome conditions, we aim to isolate the signal specifically related to proper membrane insertion and folding. This method reduces the contribution of non-specific binding or aggregation, which in turn likely

underestimates the actual yield but provides a more accurate representation of properly folded, membrane-incorporated proteins.

We highlight some additional advantages of the split GFP system:

- **Contextual Sensitivity:** While the small GFP domain could be exposed in a partially folded or unfolded protein, there is a bigger concern that the membrane protein's tendency to aggregate may not be strong enough to fully destabilize and unfold the highly stable full-length GFP. This is highly evident in our high background samples with the full-length GFP.
- **Minimized Disruption:** The smaller GFP fragments of the split GFP system are less likely to interfere with the membrane protein's folding and insertion processes compared to the larger full-length GFP. For some proteins, adding full-length GFP doubles the total length of the protein. The split-GFP system reduces the risk of introducing artifacts related to the synthesis, folding, or function of the original membrane protein.

Split-GFP signal and protein function correlation

While we believe the split-GFP assay is a better metric for yield assessment than terminally attaching the full-length GFP protein, we agree with the reviewer's comment that the split GFP assay does not inherently correlate with protein function. It would be ideal to use an assay based on the binding of the adrenergic receptors, but unfortunately, we did not identify any successful conditions for the Beta-2 or Beta-3 adrenergic receptors. The best binding assay with off-the-shelf components we could find was the antibody-based assay. We do agree that these assays still do not represent the actual function of the protein. To demonstrate actual function, we implemented an assay to assess the light-dependent proton pumping activity of bacteriorhodopsin. This assay allows us to correlate the split-GFP signal with the functional activity of the target protein.

To show that the split GFP metric correlates with the activity of a protein across a range of conditions, we screened different reaction compositions for bacteriorhodopsin (bR) synthesis with three split GFP values (one low, one medium, one high) and assessed bR function via the ACMA assay⁴. This assay monitors the light-dependent proton-pumping activity of bacteriorhodopsin.

The plot below relates bR yield as quantified by split GFP with bR activity (shifts in signal between light and dark conditions) as a function of the split GFP signal.

We have modified the main text accordingly (Results Section 1, paragraph 4, and Discussion paragraph 3), and included these data in Supplementary Figure 6. A detailed report of how the ACMA assay was conducted and quantified is described in Method 24.

Figure 3: Protein yield correlates with protein function

The functional response of bR was tested using the ACMA assay, allowing quantitative measurements of ion flux across lipid membranes. Ion flux through a specific ion channel is measured via the varying fluorescence of the ACMA dye inside and outside the vesicles. In the case of bR, ion flux is caused by response to light. (A) The functional metric plotted here is calculated as the change between the slope of the curve during the light response and the dark response (i.e., slope recovery). All plotted samples were normalized to the negative control (blank liposomes excluded from cell-free mixture) ($n=3$, pairwise t-test). (B) Correlation between split-GFP obtained yield and slope recovery.

Highlighting other high throughput papers

The primary motivation for providing the context of other HTP membrane protein papers is not to say that those assays can replace a full functional assay. Instead, the discussion highlights the current state of the art. Furthermore, these articles have provided a similar, if not more thorough, investigation to characterize the folded state of our proteins compared to other related papers.

Reviewer #3:

The authors have tried to respond to the concerns as directly as possible, including my original comments. I think this has been reasonable. Given the limited number of activity assays (which are described in response to Reviewer 1), it is still a bit unclear how much this really moves the needle on membrane protein expression, which of course requires activity. The work does catalog many conditions for membrane protein synthesis that the abstract now better highlights.

We thank the reviewer for the thoughtful follow-up comments and for recognizing our efforts to address the concerns raised. We appreciate the understanding of the scope of our work and the context in which it was conducted.

Reviewer #4:

The authors have addressed some, but not all, of my concerns. I appreciate the bolstered analysis of the effects of active learning vs. random reaction choice. However, I continue to have concerns about how the authors assess the success or failure of their method in improving or rescuing protein expression.

Comment 1: My best understanding of the authors' method as presented is that, for any given

protein, it takes only one expression condition with improved yield at $p < 0.05$ for the authors to declare that they have successfully improved liposome-bound expression of that protein. Since the authors screen hundreds of conditions per protein, they are, in fact, testing hundreds of hypotheses and declaring success if any one of those hundreds of hypotheses appears true with $p < 0.05$. Unless I am really missing something, there must be a multiple-hypotheses correction when drawing conclusions like this. Otherwise, the authors could simply set up 20 identical conditions in quadruplicate and expect that one of those conditions might by chance appear to have improved expression and therefore declare “success.” If my understanding is correct, then please carry out the multiple hypotheses correction. If my understanding is not correct, and the authors use some other definition of success, then please make that definition explicit in the paper.

We appreciate the reviewer’s detailed explanation of their concern and apologize for not properly understanding the original comment about multiple hypothesis testing. We have now conducted multiple hypothesis corrections on all the data and reassessed the findings. With the Benjamini-Hochberg correction, we found that only CRCM did not pass the 0.05 threshold. We have updated our results accordingly.

Comment 2: The authors have not addressed my question. To clarify what I mean, Figure 3C lists six of the proteins whose expression previously failed in other studies: Glut, Neu, Vol, Aux, InP, and Dia. In their revised text, the authors state that all six were successfully expressed by MEMPLEX. Have the authors validated successful expression of any of those six proteins by low-throughput methods? In other words, did MEMPLEX actually find conditions that successfully express transmembrane proteins that other researchers had previously failed to express? That should be a straightforward and very important assessment of the new method’s real-world success rate, yet I can’t find that assessment in the paper.

We thank the reviewer for the added clarifications. We should have explicitly stated the proteins that were successfully expressed. The proteins that were previously unsuccessful but found to be successful in our work were: Glut, Vol, Aux, InP, Dia, and CaM. All these proteins had at least one condition yielding $>1\text{pmol}$ of protein and passed the multiple hypothesis corrected p -value threshold of 0.05, which was the calculated benchmark for a protein to be considered successfully synthesized. Regarding low throughput methods to characterize the production of the proteins, we conducted SEC and flow cytometry on several additional proteins. These tests demonstrate that the protein is strongly colocalized with the membrane. The tested proteins include Dia from the previously unsynthesized group and OR1A1, CD47, and CD9 from the previously not attempted group. The results of the low throughput analysis are shown in Supplementary Figure 15. We also conducted a western blot analysis on CD47 with a readily available antibody.

References:

- (1) LaFleur, T. L.; Hossain, A.; Salis, H. M. Automated Model-Predictive Design of Synthetic Promoters to Control Transcriptional Profiles in Bacteria. *Nat Commun* **2022**, *13* (1), 5159. <https://doi.org/10.1038/s41467-022-32829-5>.
- (2) Cetnar, D. P.; Salis, H. M. Systematic Quantification of Sequence and Structural Determinants Controlling mRNA Stability in Bacterial Operons. *ACS Synthetic Biology* **2021**. <https://doi.org/10.1021/acssynbio.0c00471>.
- (3) Jacobs, M. L.; Boyd, M. A.; Kamat, N. P. Diblock Copolymers Enhance Folding of a Mechanosensitive Membrane Protein during Cell-Free Expression. *Proceedings of the National Academy of Sciences* **2019**, *116* (10), 4031–4036. <https://doi.org/10.1073/pnas.1814775116>.
- (4) Islam, Md. S.; Gaston, J. P.; Baker, M. A. B. Fluorescence Approaches for Characterizing Ion Channels in Synthetic Bilayers. *Membranes (Basel)* **2021**, *11* (11), 857. <https://doi.org/10.3390/membranes11110857>.

Dear Reviewers,

Please find attached the revised manuscript, “Designer artificial environments for membrane protein synthesis” for publication in Nature Communications.

We greatly appreciate the feedback with regard to our manuscript. We have addressed all reviewer comments, and all changes are highlighted in yellow. Below, we have included a detailed point-by-point response to the reviewer’s comments.

I hope that you will find our response adequate. Please let us know if you have any concerns regarding the changes that we made.

Reviewer comments

Reviewer #1:

Comment 1: To my opinion, the response to my concerns is still not adequate as specified below. Nevertheless, the authors invested a remarkable workload and their presented MEMPLEX approach goes into the right direction. However, interpretation of the current results still appears sometimes to be overestimated, and I would appreciate if rather preliminary and misleading statements regarding the below mentioned topics would be clearly modified as exemplified below. I would find it more valuable to present the pipeline not as a high-end tool but rather as a first case study with an initial basic platform that may become individually adjusted in future to the requirements of certain membrane protein types or families.

We thank the reviewer for recognizing the added efforts to address their concerns.

Overall, we have updated the manuscript, following reviewer suggestions, to clarify the fluorescent solubilization reporter and its associated claims. We have also updated the method and supplementary sections with additional details.

Comment 2: The high impact of proper initiation of translation was shown in several previous reports and is well recognized. The shown calculation of translation initiation strength of their targets is hard to evaluate as method data are missing. I assume that just the promoter and nearby regions were considered. The algorithms of the cited references are to my opinion not suitable, as they do not consider sequences and potential secondary structure formations of the individual codon regions. Secondary structure formation of the initiation area with further downstream sequences can protect proper interaction with the ribosome and thus prevent translation. Programs are available to predict potential secondary structure formations, but they have to be applied to each individual codon region and usually result into too many possibilities. Thus, empirical approaches as e.g. published in Cho et al 2023 are more straightforward and highly successful. It would have been interesting to perform that with some of the low expressing targets.

We appreciate the reviewer’s comment regarding the complexity of translation initiation and the role of downstream secondary structures. Our inclusion of the *de novo* DNA calculations, which draw on the Salis lab algorithms, was intended to provide a baseline prediction of each template’s translational feasibility around the ribosome-binding site and start codon. We recognize that these models focus primarily on the 5’ UTR/initiation region and may not fully capture all possible secondary structures that could form further downstream.

In agreement with the reviewer, empirical methods—such as the approach reported in Cho *et al.* (2023)—are indeed valuable and often yield more definitive results by experimentally probing sequence variants and their expression outcomes. In future work, we plan to incorporate these or

related empirical optimization strategies for low-expressing targets, as a complementary approach to *in silico* predictions. Nonetheless, we included these calculations largely to show that significant translational impediments at the *start* site were unlikely to be the sole cause of poor expression. The future plan to investigate the optimization of DNA templates is included in the Discussion section of the paper.

Comment 3: For the usefulness of the split-GFP assay, the manuscript is lacking a clear proof of concept. In order to provide this, the authors now extended their previous analysis on the CF synthesized bR. In fact, bR and the related proteorhodopsin are frequently taken as “proof of principle” membrane protein targets in cell-free expression studies. The reason is that they are very well translated, and they rapidly integrate and fold in a large diversity of membranes and other hydrophobic environments. In short, they are “easy” membrane proteins present in almost 100% folded conformation. Consequently, if GFP or a split GFP moiety is attached, their fluorescence linearly corresponds to the synthesis of active bR, as it was shown in the new Fig. 2/supp Fig. 6. bR is thus not suitable to show that split-GFP only complements if attached to folded protein as there is simply no inactive bR present. The Supp Fig. 6 just shows that activity correlates with increasing protein yield, which is certainly expected.

This exceptional perfect situation with bR is unfortunately different with most other, less easy, membrane proteins, such as most GPCRs, transporters or channels. In these cases, similar to bR most of the CF synthesized protein associates with the membrane. However, as remarkable difference, only a fraction folds into functional conformations. A key problem of CF synthesis is that these unfolded fractions remain in the sample and are not removed by proofreading systems like in living cells. That is why FSEC does not work with CF systems. As an example, only 50% of totally membrane solubilized GPCR may be folded. The still unanswered question is now: Does the fluorescence of the complemented split-GFP correlates with the determined functional fraction of membrane protein (50%), or is it significantly higher and rather correlates with the total amount of synthesized membrane protein (100%). With full-length GFP fusions, it was shown that fluorescence clearly correlated with the total amount of synthesized protein independent of folded conformations. With split-GFP, it still would be very interesting to know it.

The message “we could not find successful conditions for beta2 and beta3” is not clear to me. As the proteins are synthesized, do they mean conditions for ligand binding? In that case it would be a strong hint that split-GFP fluorescence does not correlate with protein folding. Further evidence in that direction is the mentioned measured fluorescence of complemented split GFP after bR expression without liposomes. Can it be assumed that bR also folds without any hydrophobic environment?

To subtract the GFP fluorescence obtained after expression without liposomes is not a proper control for the expression with liposomes. These are two completely different expression modes and the fact that membrane proteins also incompletely insert into membranes or only associate with them is again ignored. In that context, supp Fig. 2 is too simplified and misleading. Unfolded proteins have random structures and consequently the exposure of the GFP11 moiety is random, meaning sometimes exposed and sometimes not. The random fluorescence of a protein precipitate has therefore nothing to do with the fluorescence of a mixture of membrane attached unfolded protein and membrane inserted folded protein.

Thank you for clarifying this point in greater detail.

We fully agree with the reviewer that the split GFP reporter described here does not necessarily report the yield of functional folded protein, and only reliably reports the fraction of membrane-

solubilized protein. The bR data is, therefore, only a proof-of-concept to demonstrate that the reported split-GFP yield can correlate with functional and fully folded protein, should the protein function be tested in parallel. Such investigation would need to be systematically implemented for every newly synthesized membrane protein within MEMPLEX. We deemed this to be outside the scope of this work, but we agree with the reviewer that this element needs further clarification in our manuscript. We have thoroughly modified the text to refer to this reporter as a “solubilization reporter” and not a “folding reporter”. We have added a supplementary note (Supplementary Note 2) and modified Supplementary Figure 2 to explicitly clarify the limitations of our reporter.

As for our previous statement: “we could not find successful conditions for beta2 and beta3”, we meant that there were no tested reaction compositions that yielded a fluorescent value significantly above the corresponding background, which indicates that it was not successfully synthesized/solubilized into the membrane. Therefore, we did not find it appropriate to attempt functional tests on proteins that were not successfully synthesized.

We agree with the reviewer that the subtraction of the no-liposome condition is not a perfect representation of the false positive signal as it does not account for the amount of protein solubilized in the membrane but is not correctly folded. We added a note in Supp Fig 2 when describing this control.

Comment 4: L. 443: The message is misleading. Structural approaches need pure protein transferred into micelles, polymers or nanoparticles. Considering the expected high losses during purification and processing, a basic expression yield of max 100 µg/mL is far too low.

We have modified the text to mitigate the confusion.

Comment 5: L. 46: Ref 10 is not about CF membrane protein synthesis.

The reference for this part of the statement was corrected.

Comment 6:

- L. 86: not folding but fluorescent protein “solubilization” monitor
- L 152: general membrane association.

We have modified this term throughout the manuscript and added further explanations on this in Supplementary Note 2.

Comment 7:

- L. 123/124: GFP-11 is inaccessible when the attached protein misfolded. This interpretation is not correct and misleading. The authors use the cited references as proof for their statement that split-GFP complementation only happens with functional proteins. In the cited references, fluorescence of complemented split GFP was used to distinguish precipitate from soluble protein. Precipitated protein is certainly misfolded; However, it was not shown that the soluble proteins in addition contain a misfolded fraction that may give fluorescence. Solubilized CF generated membrane proteins frequently contain a mixture of folded and misfolded protein. Thus, solubilization is not identical to functional folding.
- L 127/128: Based on the current data, this statement is pure speculation.

Thank you for these excellent points. We have corrected the text accordingly, added a Supplementary Note (2), and modified Supplementary Figure 2 to clarify these points in greater detail.